

# Comparing Airborne and Satellite Retrievals of Optical and Microphysical Properties of Cirrus and Deep Convective Clouds using a Radiance Ratio Technique

Trismono C. Krisna[1], Manfred Wendisch[1], André Ehrlich[1], Evelyn Jäkel[1], Frank Werner[1,*],
Ralf Weigel[3,4], Stephan Borrmann[3,4], Christoph Mahnke[3], Ulrich Pöschl[4], Meinrat O. Andreae[4,6],
Christiane Voigt[2,3], and Luiz A. T. Machado[5]

[1]Leipziger Institut für Meteorologie (LIM), Universität Leipzig, Leipzig, Germany
[2]Institut für Physik der Atmosphäre, Deutsches Zentrum für Luft und Raumfahrt (DLR), Oberpfaffenhofen, Germany
[3]Institut für Physik der Atmosphäre, Johannes Gutenberg-Universität Mainz, Mainz, Germany
[4]Biogeochemistry, Multiphase Chemistry, and Particle Chemistry Departments, Max Planck Institute for Chemistry (MPIC), Mainz, Germany
[5]Center of Weather Forecast and Climates Studies, National Institute for Space Research, Sao Jose Dos Campos, Brazil
[6]Scripps Institution of Oceanography, University of California San Diego, La Jolla, California, USA
[*]now at : Joint Center for Earth Systems Technology, University of Maryland, Baltimore, MD, USA

*Correspondence to:* Trismono Candra Krisna
(trismono_candra.krisna@uni-leipzig.de)

**Abstract.** Solar radiation reflected by cirrus and deep convective clouds (DCCs) was measured by the Spectral Modular Airborne Radiation Measurement System (SMART) installed on the German HALO (High Altitude and Long Range Research Aircraft) during the ML-CIRRUS and the ACRIDICON-CHUVA campaigns. In particular flights, HALO performed closely collocated measurements with overpasses of the Moderate Resolution Imaging Spectroradiometer (MODIS) on board of Aqua

satellite. Based on the nadir upward radiance, the optical thickness $\tau$ and bulk particle effective radius $r_{\mathrm{eff}}$ of cirrus and DCC are retrieved using a radiance ratio algorithm which considers the cloud thermodynamic phase, the cloud vertical profile, multi layer clouds, and heterogeneity of the surface albedo. For the cirrus case, the comparison of $\tau_{\mathrm{ci}}$ and $r_{\mathrm{eff,ci}}$ retrieved on the basis of SMART and MODIS upward radiances yields a normalized mean absolute deviation of $0.5\,\%$ for $\tau_{\mathrm{ci}}$ and $2.5\,\%$ for $r_{\mathrm{eff,ci}}$. While for the DCC case, the respective deviation is $5.9\,\%$ for $\tau_{\mathrm{dcc}}$ and $13.2\,\%$ for $r_{\mathrm{eff,dcc}}$. The larger deviations in case

of DCC are mainly attributed to the fast cloud evolution and three-dimensional radiative effects. Measurements of spectral radiance at near-infrared wavelengths with different absorption by cloud particles are employed to investigate the vertical profile of cirrus effective radius. The retrieved values of cirrus effective radius are further compared with corresponding in situ measurements using a vertical weighting method. Compared to the MODIS observation, spectral measurements of SMART provide an increased amount of information on the vertical distribution of particle sizes at cloud top, and therefore allow

to reconstruct the profile of effective radius at cloud top. The retrieved effective radius differs to in situ measurements with a normalized mean absolute deviation between $4 - 19\,\%$, depending on the wavelength chosen in the retrieval algorithm. While, the MODIS cloud product underestimates the in situ measurements by $48\,\%$. The presence of liquid water clouds below



the cirrus, the variability of particle size distributions, and the simplification in the retrieval algorithm assuming vertically homogeneous cloud are identified as the potential error contributors.

## 1 Introduction

Clouds constitute an important component of the global climate system. Covering about $75\%$ of the Earth, their high albedo essentially affects to the Earth's energy budget (Wylie et al., 2005; Kim and Ramanathan, 2008; Stubenrauch et al., 2013). In particular, cirrus clouds are not adequately represented in general circulation models. They pose large challenges in predicting future climate changes (Heymsfield et al., 2017) because their coverage regionally can be as high as about $50\%$ in the tropics and $30\%$ over Europe. Cirrus clouds reduce the loss of radative energy to space due to absorption of terrestrial radiation and re-emission at a lower temperature (greenhouse effect). Optically thin cirrus is expected to contribute to a warming of the atmosphere below the cloud, while thick cirrus may cool (e.g., Liou, 1986; Wendisch et al., 2005, 2007; Voigt et al., 2017).

On the other hand, deep convective clouds (DCCs) alter the radiative energy distribution in the atmosphere by reflection of solar and absorption or emission of terrestrial radiation, as well as by changes of liquid and ice water and hydrometeor profiles (Jensen and Del Genio, 2003; Sherwood et al., 2004; Sohn et al., 2015). Their life cycle is determined by complex microphysical processes including changes of the thermodynamic phase and the development of precipitation. DCCs are typically optically thick and often associated with heavy precipitation and severe weather events. In addition, DCCs are related to strong turbulence, vertical motion, lightning, hail formation and icing (Mecikalski et al., 2007; Lane and Sharman, 2014).

Two important cloud parameters which quantify cloud radiative properties are the cloud optical thickness $\tau$ and effective radius $r_{\mathrm{eff}}$ (King et al., 2013). Changes in $\tau$ and $r_{\mathrm{eff}}$ can lead to a cooling or warming effect (Slingo, 1990; Shupe and Intrieri, 2004). Passive remote sensing using reflected solar or emitted thermal infrared radiance measured either by satellite or airborne platform is a well–established technique to retrieve cloud properties such as $\tau$ and $r_{\mathrm{eff}}$ (Stephens and Kummerow, 2007). Cloud properties are retrieved by inversion of radiative transfer model simulations, which is often realized by pre–calculated lookup tables (Nakajima and King, 1990; Platnick et al., 2017). Airborne remote sensing of cirrus and DCCs properties gives a snapshot of the cloud field only, whereas satellite remote sensing (e.g., MODIS) may provide statistical data on a global scale and record long time series to determine temporal changes of cloud properties (Rosenfeld and Lensky, 1998; Lindsey et al., 2006; Berendes et al., 2008).

The performance of post-launch validation activities is crucial to verify the quality of satellite measurement systems. It is essential to address all components of the measurement system, i.e., sensors, algorithms, along with the originally measured radiances and derived data products, and continue validation activities throughout the satellite life (Larar et al., 2010). Radiance measurements above highly reflecting surfaces such as salt lake, desert, snow/ice (Wan, 2014) and clouds (Mu et al., 2017) are usually evaluated in order to monitor the long term stability of the satellite sensors. An estimated uncertainty of about $1 - 5\%$ in case of MODIS reflective solar bands (RSBs) was reported by Xiong et al. (2003). This measurement uncertainty propagates into the retrieval results (King and Vaughan, 2012). Additionally, uncertainties in the retrieval may arise from errors in the assumed surface albedo and three-dimensional (3-D) radiative effects. An inaccurate assumption of the surface albedo can lead



to an uncertainty of up to $83\%$ for $\tau$ and $62\%$ for $r_{\text{eff}}$ (Rolland and Liou, 2001; Fricke et al., 2014; Ehrlich et al., 2017). King et al. (2013) showed that three-dimensional (3-D) radiative effects can enhance the retrieval uncertainty and should be considered when interpreting the comparison of retrieved cloud properties from different instruments. In order to reduce these uncertainties, collocated measurements i.e., airborne and satellite remote sensing accompanied with in situ observations are

necessary. The similar geometry of airborne and satellite radiation sensors allows for a direct comparison of upward radiance and a stringent validation of methodologies and retrieval algorithms.

Platnick (2000), King et al. (2013), Nagao et al. (2013), Miller et al. (2016), and van Diedenhoven et al. (2016) discussed that $r_{\text{eff}}$ retrieved from reflected radiation measurements depends on the vertical penetration of reflected photons into the cloud. At a wavelength with higher absorption by cloud particles, the probability of photons being scattered back out of the cloud

without being absorbed decreases. Therefore, using different near-infrared wavelengths with different absorption by cloud particles in the retrieval algorithm will result in $r_{\text{eff}}$ related to different cloud altitudes. Chang and Li (2002), Chang and Li (2003), and King and Vaughan (2012) showed that airborne-satellite retrievals assuming a vertically homogeneous cloud result in a single bulk value of effective radius $r_{\text{eff}}$ representing the entire cloud layer where the contribution of each individual layer to the absorption is a function the cloud profile itself and the wavelength chosen in the retrieval. Thus, it should be noted

that the effective radius retrieved by this technique does not represent an effective radius at a single layer only, and therefore does not represent the real profile of effective radius in the cloud. In reality, as measured by in situ instruments, the cloud particle effective radius is sampled at a specific cloud altitude $z$ and it considerably varies as a function of altitude $r_{\text{eff,z}}$. These different definitions make difficulties to compare remote sensing and in situ observations and can lead to large discrepancies. A direct comparison at a single cloud layer is problematic because it is unclear to what level the remote sensing retrieved

$r_{\text{eff}}$ corresponds to the in situ $r_{\text{eff,z}}$. Consequently, this also can reveal significant discrepancies between retrieved an in situ measured cloud water path, as reported in Chang and Li (2003), Chen et al. (2007), and King and Vaughan (2012).

A useful comparison between remote sensing results and in situ measurements can only be made when the full vertical extent of the cloud is measured by an aircraft profiling throughout the cloud. Studies by Painemal and Zuidema (2011) and King et al. (2013), who compared the effective radius retrieved from MODIS observations with the average value of effective radius

measured in the near cloud top in several cases of in situ profile measurements revealed absolute deviations of up to $20\%$. King et al. (2013) found that there is no apparent link between the variation of the effective radius retrieved using different near-infrared wavelengths of MODIS and the vertical structure of effective radius measured by in situ. Painemal and Zuidema (2011) identified four potential error sources such as the variability of droplet size distributions, forming of precipitation, above cloud water vapour absorption, and viewing geometry dependent biases as potential contributors to the deviation. While,

studies by Zhang et al. (2010) and Nagao et al. (2013) argued that the discrepancy between passive remote sensing and in situ measurements is due to simplification in the retrieval algorithm which assumes in-cloud vertical homogeneity.

Measurements of spectral solar radiation using SMART installed on board of HALO during the Mid-Latitude Cirrus (ML-CIRRUS) and the Aerosol, Cloud, Precipitation, and Radiation Interaction and Dynamic of Convective Clouds System - Cloud Processes of the Main Precipitation Systems in Brazil : A Contribution to Cloud Resolving Modelling and to the Global Pre-

cipitation Measurement (ACRIDICON-CHUVA) campaign are analyzed. HALO with its long endurance of up to 8 hours and



high ceiling of up to 15 km altitude is optimally suited to fly above cirrus and DCCs (Wendisch et al., 2016; Voigt et al., 2017). In high altitude, measurements of upward radiance (cloud-reflected) are only marginally affected by atmospheric interferences due to scattering and absorption by gas molecules and aerosol particles. For the purpose of airborne-satellite validation, designated flights above clouds were carried out during the ML-CIRRUS and ACRIDICON-CHUVA campaign (Wendisch et al.,

2016; Voigt et al., 2017), which were closely collocated with overpasses of the A–Train constellation (Savtchenko et al., 2008). Two airborne campaigns are introduced in Section 2 followed by instrumentation in Section 3. In Section 4, comparison techniques, data filters, and results of upward radiance comparison are presented. The radiance ratio algorithm, forward simulation of vertically inhomogeneous cloud, vertical weighting function, heterogeneity of the surface albedo, impact of underlying liquid water cloud, and results of $\tau$ and $r_{\text{eff}}$ comparison are discussed in Section 5. In Section 6, methods and results of the

comparison between in situ and retrieved effective radius are presented. Finally, conclusions are given in Section 7.

## 2 Airborne campaigns

Data from two airborne campaigns with HALO are used in this study. Between 21 March 2014 and 15 April 2014, the ML-CIRRUS campaign performed 16 research flights over Europe and the Atlantic ocean to study nucleation, life-cycle, and climate impact of natural cirrus and aircraft induced contrail cirrus (Voigt et al., 2017; Schumann et al., 2017). Between 1 September

2014 and 4 October 2014, the ACRIDICON-CHUVA campaign performed 14 research flights combined with satellite and ground-based observations over the Brazilian Amazon rainforest to quantify aerosol-cloud-precipitation interactions and their thermodynamic, dynamic, and radiative effects of tropical deep convective clouds (DCCs) over Amazon rainforest (Wendisch et al., 2016).

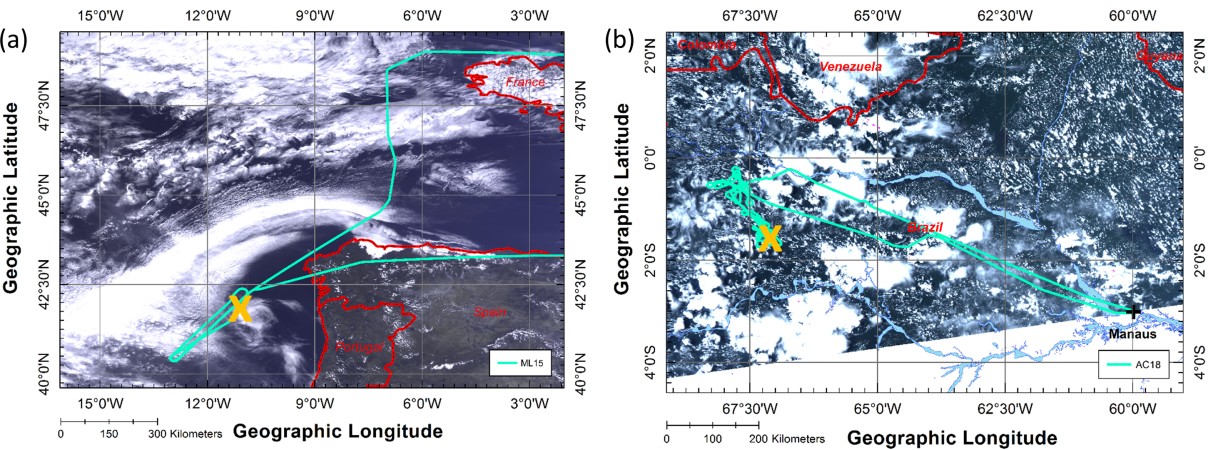

**Figure 1.** The flight trajectory of ML-15 (a) and AC-18 (b) overlayed with MODIS true color image. The yellow cross indicates the flight section which is selected for the analysis.





One common objective of ML-CIRRUS and ACRIDICON-CHUVA was the validation of satellite measurements and products. Closely collocated measurements with the A-Train during satellite overpasses were performed in order to evaluate optical and microphysical properties of cirrus and DCCs. One flight from the ML-CIRRUS (ML-15, 13 April 2014) and another one from the ACRIDICON-CHUVA (AC-18, 28 September 2014) fulfill the requirements of a reliable satellite comparison. The flight

trajectory of ML-15 is shown in Fig. 1a. During the MODIS overpass at 13:55:00 UTC, HALO flew west of Portugal over the North Atlantic. In this area, a wide field of cirrus was located above a low liquid water cloud (stratocumulus). Fig. 1b shows the flight trajectory of AC-18. HALO was flying in the north-west of Brazil over Amazonian rainforest during MODIS overpass at 17:55:00 UTC, where a DCC topped by an anvil cirrus was observed.

## 3    Instrumentation

### 3.1    Airborne

During the ML-CIRRUS and ACRIDICON-CHUVA campaign, a comprehensive set of in situ and remote sensing instruments were operated on board of HALO (Wendisch et al., 2016; Voigt et al., 2017). SMART measured spectral upward radiance $I_{s,\lambda}^{\uparrow}$, as well as spectral upward $F_{s,\lambda}^{\uparrow}$ and downward irradiace $F_{s,\lambda}^{\downarrow}$. The index "s" refers to measurements of SMART, while $\lambda$ indicates spectral quantities in units of $\mathrm{nm}^{-1}$. The irradiance data can be used to determine the spectral surface albedo

(Wendisch et al., 2001; Wendisch and Mayer, 2003; Wendisch et al., 2004). An active stabilization system keeps the optical inlets in a horizontal position during aircraft movements of up to $\pm 6°$ from the horizontal plane (Wendisch et al., 2001). SMART has two types of separate spectrometers, which measure in the solar spectrum. The Visible to Near Infrared (VNIR) spectrometer ranges from 300 - 1000 nm and the Shortwave-Infrared (SWIR) spectrometer ranges from 1000 - 2200 nm. Combination of both spectrometers cover approximately $97\%$ of the entire solar spectrum Bierwirth (2008).

The spectral resolution defined by the full width at half maximum (FWHM) is 2 - 3 nm for the VNIR spectrometer and 8 - 10 nm for the SWIR spectrometer (Werner et al., 2013). For the purpose of this study, we focus on the upward radiance. The radiance optical inlet has a field of view (FOV) of $2°$ looking at nadir (Wolf et al., 2017). The nadir radiance measured by SMART is comparable to measurements of MODIS reflective solar bands (RSBs) in the band number 1 - 19, and 26 ranging between 410 - 2130 µm (Xiong and Barnes, 2006). Primarily, SMART is calibrated radiometrically before, during, and after

each campaign using certified calibration standards traceable to NIST (National Institute of Standards and Technology) and by secondary calibration using a travelling standard. The measurement uncertainty of $I_{s,\lambda}^{\uparrow}$ is comprised of spectral calibration, spectrometer noise and dark current, radiometric calibration, and transfer calibration (Eichler et al., 2009; Brückner et al., 2014; Wolf et al., 2017). The main uncertainty results from the Signal-to-Noise-Ratio (SNR) and the calibration standard, while spectral and transfer calibration errors are almost negligible (Wolf et al., 2017). The resulting total uncertainty is about

$4\%$ for the VNIR and $10\%$ for the SWIR.

The Cloud Combination Probe (CCP) incorporates two separate instruments, the Cloud Droplet Probe (CDP) and the greyscale Cloud Imaging Probe (CIPgs) (Weigel et al., 2016). This way the CCP overall covers a size diameter range from 2 µm to 960 µm, including large aerosol particles, liquid cloud droplets and small frozen hydrometeors (Klingebiel et al., 2015). The



CDP part detects the forward scattered laser light when cloud particles cross the CDP laser (Lance et al., 2010). Thus, the CDP provides an improved replacement for the Forward Scattering Spectrometer Probe (FSSP) (Dye and Baumgardner, 1984; Baumgardner et al., 1985). Molleker et al. (2014) characterized the CDP in detail, revealing that the instrument exhibits a nominal limit for cloud particle diameters from 3 µm up to 50 µm. The CIPgs records two-dimensional shadow images of

cloud particles in a size range from 15 µm up to 960 µm with an optical resolution of 15 µm (Klingebiel et al., 2015; Weigel et al., 2016).

Specialized algorithms are used to process and analyze the captured images in order to estimate particle number concentrations, particle size distributions, and to differentiate particle shapes (Korolev, 2007). The accuracy of the cloud particle sizing is estimated to be about $10\%$ for spherical particles and correctly assumed refractive indices (Molleker et al., 2014). The sizing

uncertainty increases as a function of particles size. The size bin limits of the CCP cloud particle data are adapted to reduce ambiguities due to the Mie curve, particularly for cloud particles with small sizes less than 5 µm. The instrument sample volume is calculated as a product of the probe air speed (measurement condition) and the instrument specific effective detection area. All concentration data are corrected concerning the air compression upstream of the underwing cloud probe at the high flight speeds inherent with airborne measurements on board of HALO (Weigel et al., 2016). The robust performance of the specific

CCP instrument used in this study was demonstrated by Frey et al. (2011) for tropical convective outflow, Molleker et al. (2014) for polar stratospheric clouds (PSC), Klingebiel et al. (2015) for low level mixed-phase clouds in the Arctic, as well as by (Braga et al., 2017) and Cecchini et al. (2017) for tropical convective clouds.

## 3.2  Satellite

Satellite data used in this sudy stem from the Moderate Resolution Imaging Spectroradiometer (MODIS) - Aqua calibrated

products MYD021KM. Detailed instrument specifications and features of MODIS have been described by Platnick et al. (2003), Xiong and Barnes (2006), and others. The data contain calibrated and geolocated radiances and reflectances for 36 discrete spectral bands distributed between 0.41 µm and 14.2 µm, including 20 RSBs and 16 thermal emissive bands (TEBs) (Platnick et al., 2003; Xiong and Barnes, 2006), with nadir horizontal resolutions of about 1 km. The radiances are generated from MODIS Level 1A scans of raw radiance and in the process converted to geophysical units. The solar reflectance values

are based on a solar diffuser panel for reflectance calibration up through the RSBs and an accompanying diffuser stability monitor for assessing the stability of the diffuser up to 1 µm (Platnick et al., 2003). The spectral response is determined by an interference filter overlying a detector array imaging a 10-km along track scene for each scan (40, 20, and 10 elements arrays for the 250 m, 500 m, and 1 km bands, respectively) (Platnick et al., 2003). Onboard instruments used for in-orbit radiometric calibration were discussed by Xiong et al. (2003) and Sun et al. (2007).



## 4 Comparison of upward radiance

### 4.1 Spectral and spatial resolution adjustment

SMART and MODIS have different different spectral resolutions. MODIS measures in broad spectral bands, while SMART measures a continuous spectrum with FWHM between 2 - 10 nm. Therefore, to compare both measurements the spectral

upward radiance of SMART $I_{s,\lambda}^{\uparrow}$ must be convoluted with the MODIS relative spectral response $R(\lambda)$. The convoluted radiance of SMART $I_{S,\lambda}^{\uparrow}$ is calculated by:

$$I_{S,\lambda}^{\uparrow} = \frac{\int_{\lambda_1}^{\lambda_2} I_{s,\lambda}^{\uparrow} \cdot R(\lambda) \, \mathrm{d}\lambda}{\int_{\lambda_1}^{\lambda_2} R(\lambda) \, \mathrm{d}\lambda}, \tag{1}$$

In this study, upward radiances at the MODIS band 1 ($\lambda = 645$ nm), band 5 ($\lambda = 1240$ nm), and band 6 ($\lambda = 1640$ nm) will be primarily used to retrieve $\tau$ and $r_{\mathrm{eff}}$. However, it is known that 15 of the 20 detectors in the MODIS-Aqua band 6 are either

nonfunctional or noisy. According to Wang et al. (2006), the MODIS radiance band 6 $I_{\mathrm{M,B6}}$ can be restored using the MODIS radiance band 7 $I_{\mathrm{M,B7}}$ ($\lambda = 2130$ nm) by:

$$I_{\mathrm{M,B6}} = 1.6032 \cdot I_{\mathrm{M,B7}}^3 - 1.9458 \cdot I_{\mathrm{M,B7}}^2 + 1.7948 \cdot I_{\mathrm{M,B7}} + 0.012396, \tag{2}$$

MODIS data used in this study are delivered at a horizontal resolution of 1 km at nadir, whereas the spatial resolution of SMART varies depending on the flight altitude and temporal resolution. At a flight altitude of 10 km, SMART has a swath of

approximately 349 m at the Earth surface. During the two campaigns, the temporal resolution of SMART was between 0.2 - 0.5 s, depending on the measurement conditions. Therefore, this has to be considered in the data analysis. In order to decrease biases resulting from comparisons of individual measurements, SMART measurements are averaged over 1 s resolution using a binning method.

### 4.2 Data filter

Only clouds with a top altitude higher than 8 km are selected. The higher proximity to TOA reduces the influence of scattering and absorbtion by atmospheric molecules and aerosol particles. Consequently, no correction of the influence of the atmospheric layer above HALO is needed. To assure a similar viewing zenith angle of SMART and MODIS, only nadir observations in the center of MODIS swath were selected for the comparison. Werner et al. (2013) discussed that off-nadir measurements of less than 5° may lead to a bias in the retrieved $\tau$ and $r_{\mathrm{eff}}$ of up to $1\,\%$ and $5\,\%$, respectively. To minimize this bias, SMART

measurements with roll and pitch angles larger than 3° are discarded and only straight flight legs with altitude changes of less than 50 m are analyzed. Cloud edges are associated with sharp changes in $I_{\lambda}^{\uparrow}$ and 3-D radiative effects. Fisher (2014) discussed variations in cloud height and surface orology to find an offset distance assigned to an uncertainty of $\pm$ 40 m. Therefore, the first and the last pixel of MODIS cloudy pixels are masked. For this purpose, the cloud mask algorithm by Ackerman et al. (1998) is employed to discriminate clear and cloudy pixels.





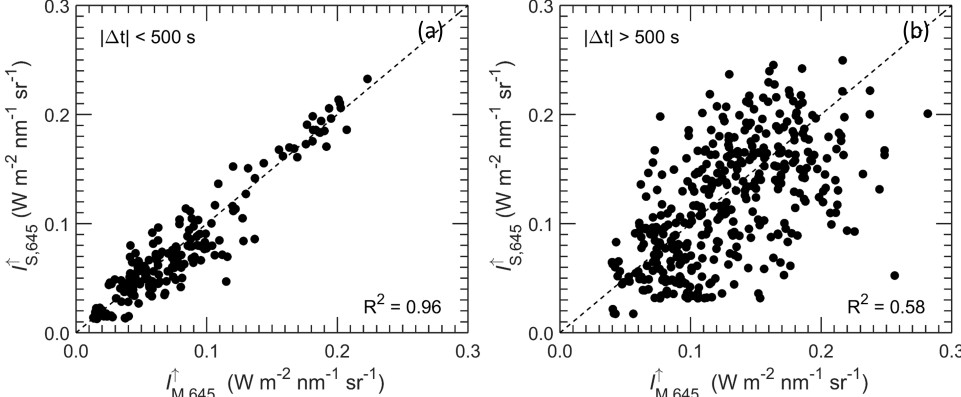

**Figure 2.** Comparison of $I_{645}^{\uparrow}$ measurements between SMART and MODIS for an absolute time difference $|\Delta t|$ of $\leq 500$ s (a) and $\geq 500$ s (b). Data were taken from ML-15 during 13:40:00 - 14:20:00 UTC.

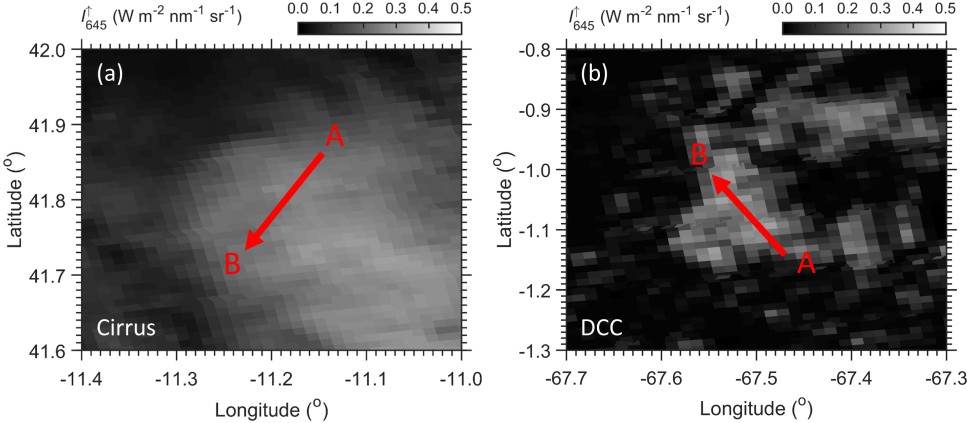

**Figure 3.** MODIS radiance band 1 ($\lambda = 645$ nm) for the cirrus case (a) and the DCC case (b) overlayed with the selected flight legs of HALO during cloud measurements (red line). The flight direction is from point A to B.

MODIS flies much faster than the aircraft. Therefore, it is impossible that SMART and MODIS always measure exactly above each other along the joint flight track. To analyze effects caused by time shifts between SMART and MODIS, data from ML-CIRRUS are divided into groups within and without a threshold of 500 s time delay. Fig. 2a yields that the comparison between $I_{S,645}^{\uparrow}$ and $I_{M,645}^{\uparrow}$ for $|\Delta t| \leq 500$ s shows a better agreement, while for $|\Delta t| \geq 500$ s reveals a scatter as shown in Fig. 2b with the respective correlation coefficient $R^2 = 0.96$ for $|\Delta t| \leq 500$ s and $R^2 = 0.58$ for $|\Delta t| \geq 500$ s. The large scatter for $|\Delta t| \geq 500$ s is mainly attributed to the fast horizontal wind speed during cirrus measurements which was $21 \text{ ms}^{-1}$ on average. In addition, the wind direction is also a key factor causing a significant cloud drift within time delay. In case of DCC, the horizontal wind speed was smaller with an average of $9 \text{ ms}^{-1}$. However, the fast cloud evolution is the major issue for DCC. Therefore, all comparisons are restricted to $|\Delta t| \leq 500$ s. After the filtering, only two suitable cases are left which fulfill all





**Table 1.** Flight descriptions and atmospheric conditions during cloud measurements. Horizontal wind speed HWS and solar zenith angle $\theta_0$ are averaged during the selected time series.

| Flight | Date | Cloud Type | Appearance | $z_t$ | Time - UTC | HWS | $\theta_0$ |
|---|---|---|---|---|---|---|---|
| | | | | (km) | (HH:MM:SS) | $(\mathrm{m\,s^{-1}})$ | (°) |
| ML-15 | 04/13/2014 | Cirrus | Homogeneous | 12 | 13:56:20 - 13:57:35 | 21 | 37 |
| AC-18 | 09/28/2014 | DCC | Inhomogeneous | 8 | 17:56:00 - 17:57:30 | 9 | 26 |

the requirements of the analysis. The first case, a cirrus cloud located above low liquid water clouds (stratocumulus) is selected from ML-15 between 13:56:20 - 13:57:35 UTC as shown in Fig. 3a. The cloud top altitude $z_t$ of cirrus was about 12 km while HALO flew at about 12.2 km altitude. The second case, a DCC topped by an anvil cirrus is selected from AC-18 between 17:56:00 - 17:57:30 UTC as presented in Fig. 3b. The cloud top altitude $z_t$ of the selected DCC was about 8 km while HALO

flew at 8.3 km altitude. Flight descriptions and atmospheric conditions during cloud measurements are summarized in Table 1. Each case is comprised of a 75 s for the cirrus and 90 s for the DCC case. For HALO flight path at constant altitudes, those correspond to a horizontal distance of about 15 km and 18 km, respectively.

### 4.3   Result of upward radiance comparison

Upward radiances measured by SMART and MODIS are compared for the two cloud cases, cirrus and DCC. Fig. 4a shows

time series of upward radiance centered at $\lambda = 1240$ nm measured by SMART $I^{\uparrow}_{\mathrm{S},1240}$ and MODIS $I^{\uparrow}_{\mathrm{M},1240}$, while Fig. 4b is the scatter plot of the respective measurements. Although the cirrus was located above liquid water clouds, this multi-layer cloud structure does not affect the comparison as it is independent on the target observed by both instruments. It is found, that $I^{\uparrow}_{\mathrm{M},1240}$ measured by MODIS are systematically higher than those measured by SMART $I^{\uparrow}_{\mathrm{S},1240}$. This wavelength is characterized by low absorption by cloud particles, which is useful to retrieve $r_{\mathrm{eff}}$ from the lower cloud layer. Consequently, the retrieval of $r_{\mathrm{eff}}$

is highly sensitive to small changes in the measurements. At a given $6\%$ measurement uncertainty, the observed differences of $I^{\uparrow}_{\mathrm{M},1240}$ can result in an uncertainty of up to $50\%$ in the retrieved $r_{\mathrm{eff}}$. To gain meaningful cloud properties from the retrieval using measurements at this wavelength, $I^{\uparrow}_{\mathrm{M},1240}$ has to be corrected. According to Lyapustin et al. (2014), a correction factor $g$ is calculated by the slope of the linear regression between $I^{\uparrow}_{\mathrm{M},1240}$ and $I^{\uparrow}_{\mathrm{S},1240}$. The resulting $g$ yields a value of 0.86. Thus, the $I^{\uparrow}_{\mathrm{M},1240}$ is corrected by the following equation:

$$I^{\uparrow}_{\mathrm{M},1240,\mathrm{cor}} = g \cdot I^{\uparrow}_{\mathrm{M},1240}, \tag{3}$$

In the following, $I^{\uparrow}_{\mathrm{M},1240}$ always refers to $I^{\uparrow}_{\mathrm{M},1240,\mathrm{cor}}$. Fig. 5 shows time series of upward radiance measured by SMART $I^{\uparrow}_{\mathrm{S},\lambda}$ and MODIS $I^{\uparrow}_{\mathrm{S},\lambda}$ centered at $\lambda = 645$ nm (a), 1240 nm (b), and 2130 nm (c) for the cirrus case, while Fig. 6 is for the DCC case. Additionally, the scatter plots of the respective measurements are shown in Fig. 7. The results show, that the radiance





measurements of SMART and MODIS agree better for the cirrus case than for the DCC case. The larger discrepancies in case of DCC are mainly attributed to the fast cloud evolution.

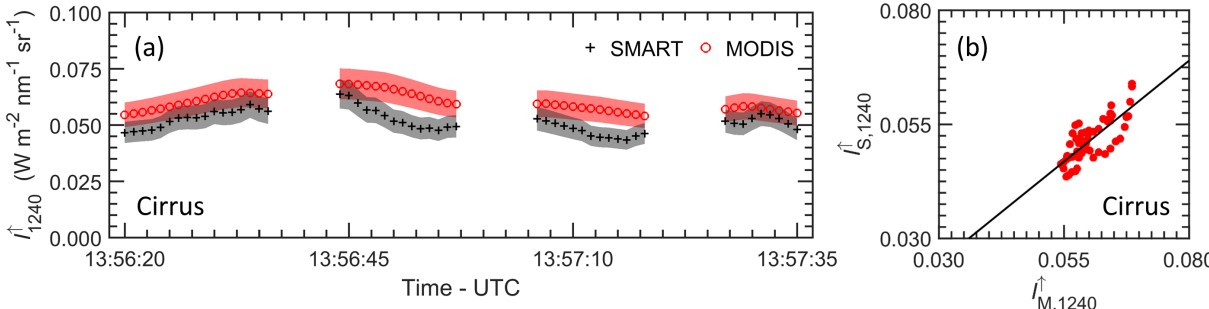

**Figure 4.** (a) Time series of SMART $I^{\uparrow}_{S,1240}$ (black) and MODIS uncorrected $I^{\uparrow}_{M,1240}$ (red) for the cirrus case. Shaded areas illustrate measurement uncertainties. Gaps on the time series indicate when the shutter of SMART closed for dark current measurements. (b) The scatter plot of the respective measurements. The black line represents a linear regression line.

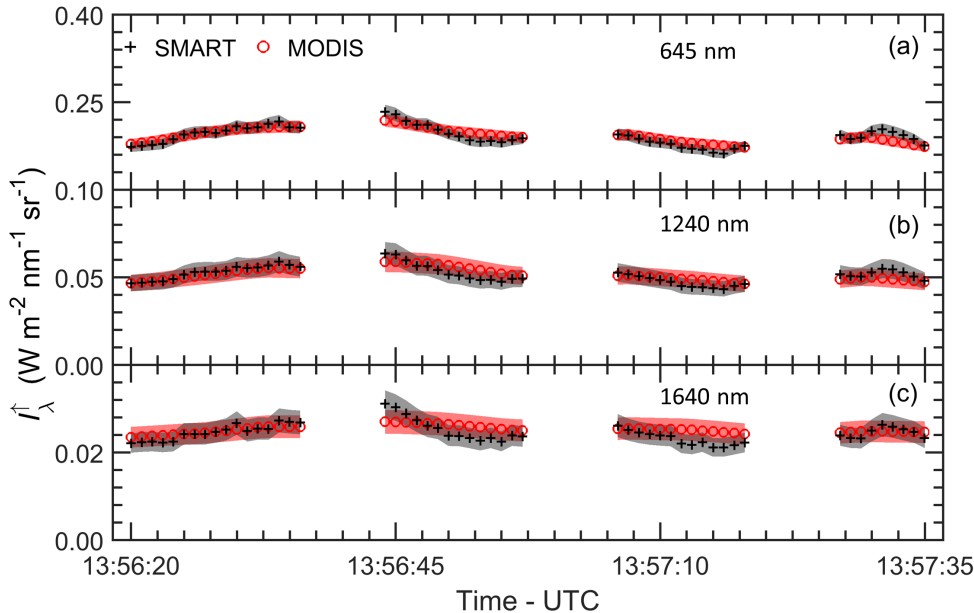

**Figure 5.** Time series of $I^{\uparrow}_{\lambda}$ centered at $\lambda = 645$ nm (a), 1240 nm (b), and 1640 nm (d) measured by SMART (black) and MODIS (red) for the cirrus case. $I^{\uparrow}_{M,1240}$ has been corrected using Eq. 3.

Fig. 8 shows the comparison of upward radiance measured by SMART and MODIS at reflective solar bands (RSBs) for the cirrus (a) and DCC case (b). Radiance measurements are averaged along the selected time series. The solid line represents





spectral radiance measured by SMART $I_{s,\lambda}^{\uparrow}$. $I_{S,\lambda}^{\uparrow}$ represents the convoluted radiance of SMART using Eq. 1, while $I_{M,\lambda}^{\uparrow}$ is the radiance measured by MODIS. The resulting mean $\pm$ standard deviations $\eta$ are summarized in Table 2. To quantify the agreement between SMART and MODIS measurements, the normalized mean absolute deviation $\zeta$ is calculated by:

$$\zeta = \frac{1}{n} \sum_{i=1}^{n} \left| \frac{x_i - \overline{x}}{\overline{x}} \right|, \tag{4}$$

where $n$ is the number of the observed values, $x_i$ are the individual values, and $\overline{x}$ is the mean of observed values. In this case, $x_i$ are the mean value of SMART and MODIS measurements along the selected time series. For the purpose of this study, only $I_\lambda^{\uparrow}$ centered at $\lambda$ = 645 nm, 1240 nm, and 1640 nm used in the retrieval of cloud properties are analyzed.

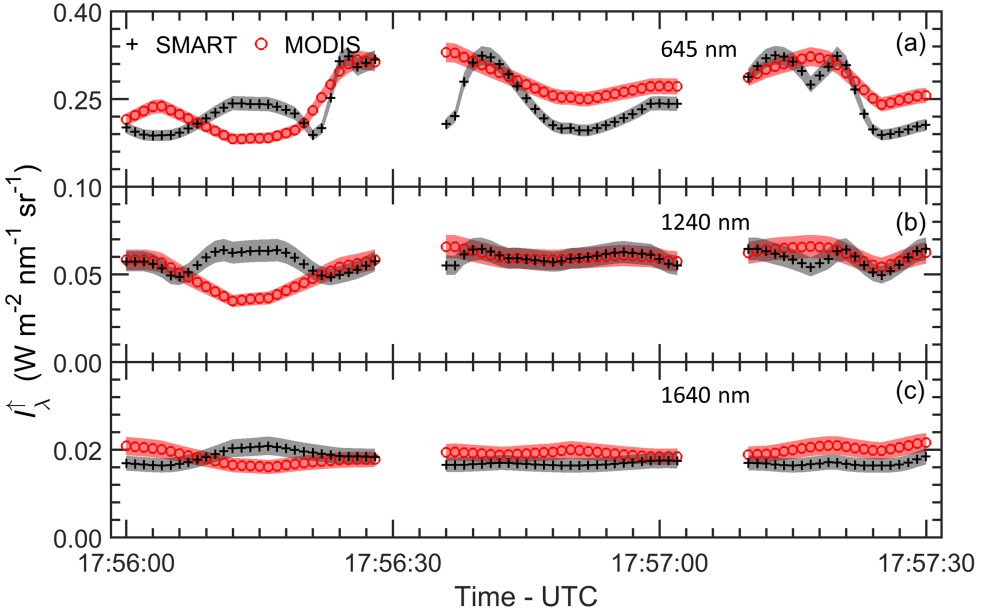

**Figure 6.** Same as Fig. 5 but for the DCC case.

For the cirrus case, $\zeta_{645}$ is found to be $0.04\,\%$, while $\zeta_{1240}$ and $\zeta_{1640}$ are $0.42\,\%$ and $1.36\,\%$, respectively. For the DCC case, $\zeta_{645}$ yields a value of $4.25\,\%$, while $\zeta_{1240}$ and $\zeta_{1640}$ are $6.72\,\%$ and $5.61\,\%$, respectively. All values of $\zeta$ are within the measurement

uncertainties. In case of cirrus, better agreements between SMART and MODIS measurements reveal that the cloud evolution is not dominant. The deviations between SMART and MODIS measurements are mainly affected by the horizontal wind speed and wind direction. For the DCC case, in addition of the fast cloud evolution, 3-D radiative effects are larger. Zhang et al. (2011) and King et al. (2013) analyzed the influence of 3-D radiative effects using the cloud heterogeneity index $\sigma_{sub}$, which is calculated by the ratio of the standard deviation and the mean value of MODIS reflectance band 2. The resulting values of $\sigma_{sub}$

range about 0.1 for the cirrus case and 0.4 for the DCC case. The higher values in case of DCC indicates that 3-D radiative effects are obviously larger and have to be considered in the analysis of retrieval results.



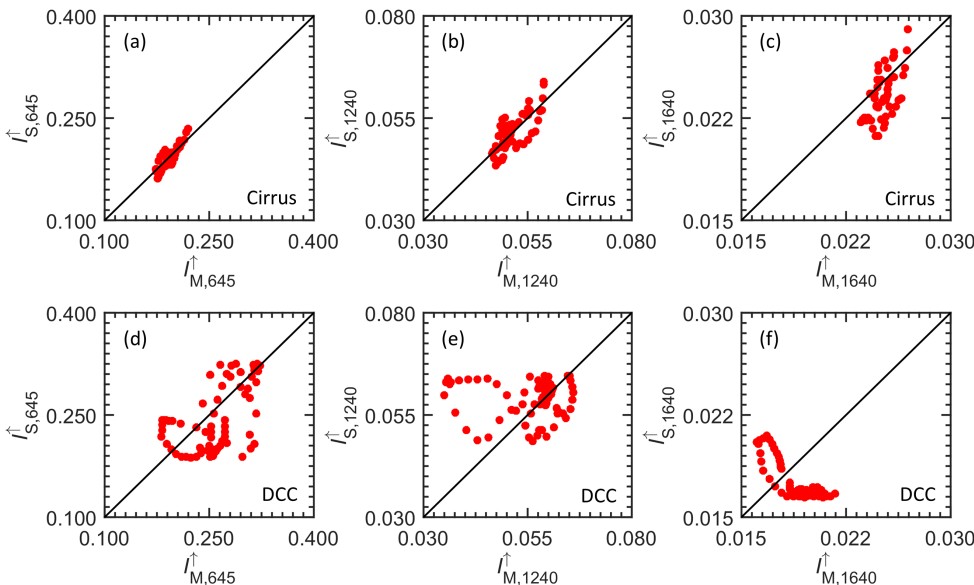

**Figure 7.** Scatter plots of upward radiances centered at $\lambda$ = 645 nm, 1240 nm, and 1640 nm measured by SMART $I_{S,\lambda}^{\uparrow}$ and MODIS $I_{M,\lambda}^{\uparrow}$. (a), (b), (c) are for the cirrus case, while (d), (e), (f) are for the DCC case. The black line represents the one-to-one line.

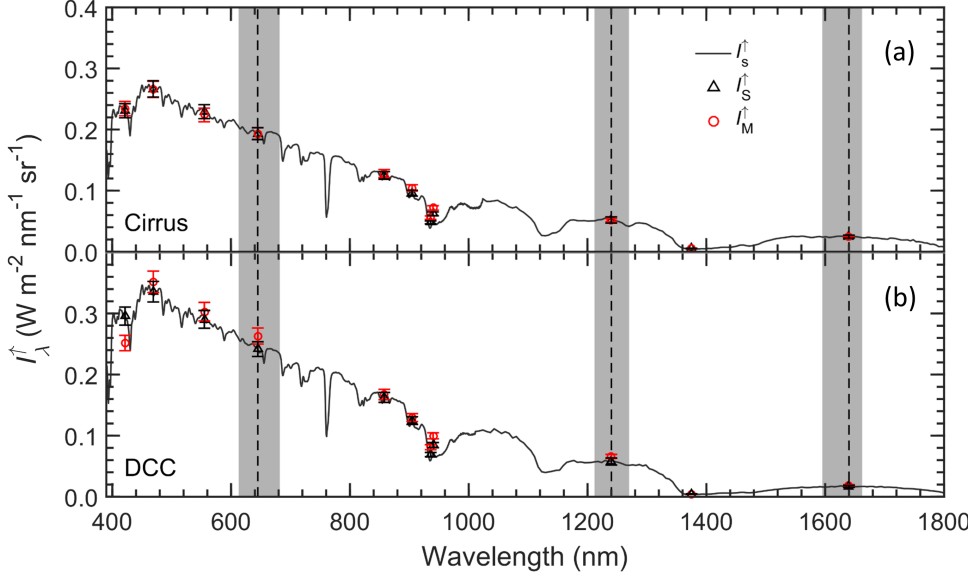

**Figure 8.** Comparison of mean $I_{\lambda}^{\uparrow}$ measured by SMART and MODIS for the cirrus case (a) and the DCC case (b) at RSBs window. Error bars represent measurement uncertainties. Wavelengths centered at $\lambda$ = 645 nm, 1240 nm, and 1640 nm are indicated by dashed lines while grey band correspond to the interval of MODIS relative spectral response $R(\lambda)$ for the respective wavelengths.





**Table 2.** Comparison of SMART $I_{\mathrm{S},\lambda}^{\uparrow}$ and MODIS $I_{\mathrm{M},\lambda}^{\uparrow}$ for the cirrus (ci) and DCC (dcc) case. $\eta$ is the mean $\pm$ standard deviation with a subscript of "S" for SMART and "M" for MODIS. $\zeta$ is the normalized mean absolute deviation between SMART and MODIS measurements.

| $\lambda$ (nm) | $\eta_{\mathrm{S,ci}}$ | $\eta_{\mathrm{M,ci}}$ | $\zeta_{\mathrm{ci}}$ (%) | $\eta_{\mathrm{S,dcc}}$ | $\eta_{\mathrm{M,dcc}}$ | $\zeta_{\mathrm{dcc}}$ (%) |
|---|---|---|---|---|---|---|
| 421 | $0.231 \pm 0.014$ | $0.234 \pm 0.011$ | 0.81 | $0.295 \pm 0.122$ | $0.251 \pm 0.013$ | 8.06 |
| 469 | $0.266 \pm 0.018$ | $0.265 \pm 0.014$ | 0.20 | $0.335 \pm 0.149$ | $0.351 \pm 0.050$ | 2.34 |
| 555 | $0.229 \pm 0.018$ | $0.224 \pm 0.013$ | 1.19 | $0.290 \pm 0.135$ | $0.303 \pm 0.047$ | 2.12 |
| 645 | $0.193 \pm 0.016$ | $0.193 \pm 0.012$ | 0.04 | $0.241 \pm 0.117$ | $0.263 \pm 0.042$ | 4.25 |
| 858 | $0.125 \pm 0.011$ | $0.128 \pm 0.008$ | 1.29 | $0.162 \pm 0.069$ | $0.167 \pm 0.018$ | 1.47 |
| 905 | $0.096 \pm 0.008$ | $0.104 \pm 0.007$ | 4.36 | $0.124 \pm 0.059$ | $0.129 \pm 0.016$ | 1.96 |
| 936 | $0.048 \pm 0.005$ | $0.056 \pm 0.005$ | 7.49 | $0.069 \pm 0.043$ | $0.080 \pm 0.018$ | 7.95 |
| 940 | $0.062 \pm 0.006$ | $0.071 \pm 0.005$ | 7.18 | $0.084 \pm 0.047$ | $0.099 \pm 0.018$ | 8.26 |
| 1240 | $0.052 \pm 0.004$ | $0.051 \pm 0.003$ | 0.42 | $0.057 \pm 0.029$ | $0.065 \pm 0.009$ | 6.72 |
| 1375 | $0.005 \pm 0.001$ | $0.005 \pm 0.001$ | 3.24 | $0.004 \pm 0.004$ | $0.004 \pm 0.003$ | 6.17 |
| 1640 | $0.024 \pm 0.002$ | $0.025 \pm 0.001$ | 1.36 | $0.016 \pm 0.010$ | $0.018 \pm 0.001$ | 5.61 |

## 5   Comparison of optical thickness and effective radius

### 5.1   Radiance ratio retrieval and uncertainty estimation

A radiance ratio technique adapted from Werner et al. (2013), Brückner et al. (2014), LeBlanc et al. (2015), Carlsen et al. (2017), and Ehrlich et al. (2017) is applied to retrieve $\tau$ and $r_{\mathrm{eff}}$ based on the nadir upward radiance measured by SMART and MODIS. The measurement uncertainties of SMART mostly originate from the radiometric calibration given by the uncertainty of the applied radiation source and the SNR. In case radiance ratios are applied, the uncertainties are reduced because the uncertainties of the radiation source identically influence all measured radiances, and therefore do not contribute to the uncertainty of the ratio. In the radiance ratio algorithm, the upward radiance at the MODIS bands centered at $\lambda_0 = 645$ nm (band 1), $\lambda_1 = 1240$ nm (band 5), and $\lambda_2 = 1640$ nm (band 6) are employed to calculate the following radiance ratios, $\Re_{1240} = I_{\lambda_1}^{\uparrow} / I_{\lambda_0}^{\uparrow}$ and $\Re_{1640} = I_{\lambda_2}^{\uparrow} / I_{\lambda_0}^{\uparrow}$.

In the retrieval algorithm, a decision tree is applied to decide the retrieval mode. The retrieval can be performed either in the liquid water or ice mode. To decide which mode is used, a cloud phase index $I_{\mathrm{p}}$ is determined by spectral slope method (Ehrlich et al., 2008; Jäkel et al., 2013). $I_{\mathrm{p}}$ is typically higher than zero for ice clouds. A threshold of 0.2 is used to discriminate between ice and liquid water clouds. For the cirrus case, time series of $I_{\mathrm{p}}$ calculated with the SMART observation yield values larger than 0.4 indicating ice clouds. This reveals, that for the cirrus case the underlying liquid water clouds did not significantly

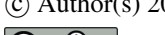



influence $I_p$. Additionally, the high values of $I_p$ show that $I_p$ is mostly sensitive to the thermodynamic phase of the top cloud layer (cirrus), while the underlying liquid water clouds below the cirrus have a limited influence on the radiances within the wavelength range analyzed for the $I_p$. For the DCC case, $I_p$ varies between 0.2 - 0.4 along the time series with a mean value of 0.25. Based on the high $I_p$, the retrievals in both analyzed cloud cases are performed in the ice mode. However in the

5  cirrus case, to retrieve $\tau$ and $r_{eff}$ of the cirrus layer, the underlying liquid water cloud have to be considered in the forward simulations.

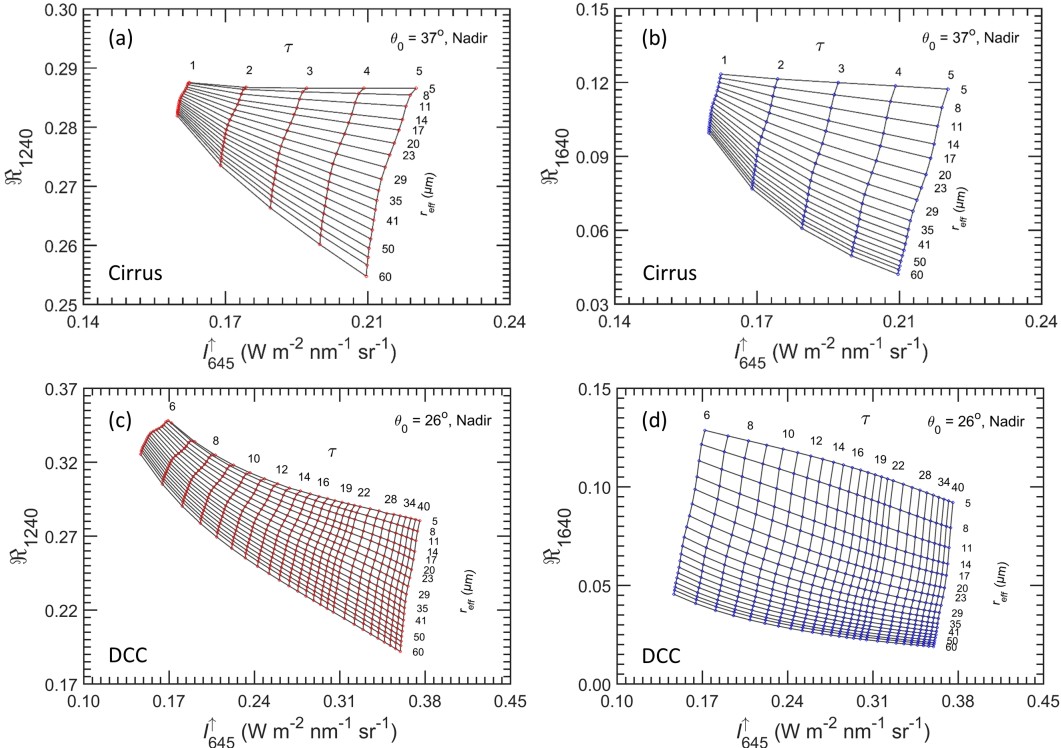

**Figure 9.** Lookup tables for the cirrus case (a,b) and DCC case (c,d). (a) and (c) are using combination 1 ($I_{645}^{\uparrow}$ and $\Re_{1240}$), while (b) and (d) are using combination 2 ($I_{645}^{\uparrow}$ and $\Re_{1640}$). Simulations are performed with solar zenith angle $\theta_0 = 37°$ for the cirrus and $\theta_0 = 26°$ for the DCC case. Ice crystal shape of GHM (Baum et al., 2007) is assumed in the forward simulation.

To calculate the lookup table, 1-D radiative transfer simulations are performed using the radiative transfer package LibRadtran 2.0 (Mayer, 2005; Emde et al., 2016), the discrete ordinate radiative transfer solver (DISORT) version 2 (Stamnes et al., 2000), and assuming vertically homogeneous clouds. The atmospheric profiles of gases and constituents are adapted from

10  the standard profile (Anderson et al., 1986) "mid-latitude" for ML-CIRRUS and "tropical" for ACRIDICON-CHUVA, and are adjusted to radio soundings close to the measurement area. Extraterrestrial spectral irradiance is taken from Gueymard (2004). The standard aerosol particle profile for "spring/summer condition" of "maritime aerosol type" is applied (Shettle, 1989). Droplet optical properties are derived from Mie calculation (Wiscombe, 1980), while ice properties of general habit





mixture (GHM) by Baum et al. (2007) are applied corresponding to ice crystal shapes measured by in situ probes during the campaigns.

The lookup tables for the DCC case are shown in Fig. 9a and 9b, whereas Fig. 9c and 9d display the lookup tables for the cirrus case. The upward radiance at a non-absorbing wavelength $I_{645}^{\uparrow}$ is combined with $\Re_{1240}$ (C1 - combination 1) and with $\Re_{1640}$ (C2 - combination 2). $I_{645}^{\uparrow}$ is most sensitive to $\tau$, while ratios $\Re_{1240}$ and $\Re_{1640}$ are most sensitive to $r_{\text{eff}}$. For the DCC case, the lookup tables cover $\tau$ between 6 - 40 with steps of 1 for $\tau$ between 6 - 22 and steps of 2 for $\tau$ between 24 - 40, while $r_{\text{eff}}$ ranges between 5 - 60 µm with steps of 3 µm. For the cirrus case, the lookup tables cover $\tau$ between 1 - 5 with steps of 1 and $r_{\text{eff}}$ between 5 - 60 µm with steps of 3 µm. Due to the underlying liquid water cloud during the cirrus case, the simulations are performed with two cloud layers. Homogeneous liquid water clouds with fixed $\tau = 8$ and $r_{\text{eff}} = 10$ µm located between 1.5 - 2 km are applied in the radiative transfer simulations. The properties of liquid water cloud are estimated by comparing simulated and measured spectral radiance averaged over the selected time series, where the $r_{\text{eff}}$ of liquid water cloud agrees with values of in situ climatological data reported in e.g., Miles et al. (2000). The lookup tables for the cirrus case are shifted to higher $I_{645}^{\uparrow}$ because the underlying liquid water cloud enhances the reflected radiation at this wavelength which is dominated by scattering processes. Similar procedures are applied to run the retrievals for the MODIS data. However, the sensor altitude is set fixed at the top of atmosphere (TOA).

In the radiance ratio method, estimated measurement uncertainties of $4\,\%$ for $I_{645}^{\uparrow}$ and $6\,\%$ for $\Re_{1240}$ and $\Re_{1640}$ are considered. Retrieval uncertainties are estimated by considering the measurement uncertainties expressed by its double standard deviation $2\sigma$. The retrieval is performed by varying each measurements separately by adding and subtracting $2\sigma$ which resulted in four solutions. The median of the four solutions is used as the retrieval result of $\tau$ and $r_{\text{eff}}$ while the standard deviation is used to represent the retrieval uncertainty, $\Delta\tau$ and $\Delta r_{\text{eff}}$. Note that the retrieval of $r_{\text{eff}}$ using C1 will result in a larger uncertainty compared to by using C2 due to smaller absorption by cloud particles at $\lambda = 1240$ nm. Consequently, the lookup tables of $r_{\text{eff}}$ for C1 are more narrow. For the cirrus case the uncertainties of the retrieved cirrus properties are higher due to the additional uncertainties of the properties of the underlying liquid water clouds.

## 5.2 Forward simulation of vertically inhomogeneous clouds

It is known from measurements, that the cloud particle sizes can significantly vary with altitudes. For non precipitating ice clouds, ice crystal sizes typically decrease as a function of altitude (van Diedenhoven et al., 2016; Heymsfield et al., 2017). However, to simplify the retrieval algorithm vertically homogeneous clouds are commonly assumed in the forward radiative transfer simulations. To quantify the effects of such simplifications, simulations with vertically inhomogeneous ice clouds are performed. Analytical profiles of effective radius as a function of geometrical height are developed according to a formulae proposed by Platnick (2000):

$$r_{\text{eff}}(z, h) = \left(a_0 - a_1 \cdot \frac{z}{h}\right)^{1/k}, \tag{5}$$

where the altitude $z$ ranges from 0 at the cloud base to $h$ at the cloud top. Constant $a_0 = r_{\text{eff,t}}^k$ and $a_1 = r_{\text{eff,t}}^k - r_{\text{eff,b}}^k$ are determined from prescribed boundary condition of the cloud top effective radius $r_{\text{eff,t}}$ and the cloud base effective radius $r_{\text{eff,b}}$.





To represent a typical vertical structure of ice clouds, $k = 3$ is chosen. The profiles of effective radius are coupled with the profiles of ice water content, which considerably decrease as a function of altitude for ice clouds (Heymsfield et al., 2017).

**Table 3.** Total optical thickness $\tau_c$, effective radius at cloud top $r_{\mathrm{eff,t}}$ and cloud base $r_{\mathrm{eff,b}}$, ice water content (IWC) from cloud base (CB) to cloud top (CT), with the boundary of geometrical hight $z$ and thickness $h$. Retrieved effective radius $r_{\mathrm{eff,ret}}$ is compared to the weighting-estimate $r^*_{\mathrm{eff,w}}$ for two near-infrared wavelengths at $\lambda = 1240$ nm and 1640 nm.

| | | | | | Specification | | | | Validation | | | |
|---|---|---|---|---|---|---|---|---|---|---|---|---|
| Cloud | $\tau_c$ | $r_{\mathrm{eff,b}}$ | $r_{\mathrm{eff,t}}$ | $k$ | IWC | $z_b$ | $z_t$ | $r^*_{\mathrm{eff,w}}$ (µm) | | $r_{\mathrm{eff,ret}}$ (µm) | |
| | | (µm) | (µm) | | (g m$^{-3}$) | (km) | (km) | 1240 nm | 1640 nm | 1240 nm | 1640 nm |
| A | 3 | 40 | 10 | 3 | 0.1 - 0.04 | 10 | 12 | 18.4 | 17.7 | 17.9 | 17.3 |
| B | 15 | 50 | 20 | 3 | 0.2 - 0.1 | 6 | 8 | 26.6 | 24.1 | 26.1 | 24.0 |

Fig. 10a and 10b show the profile of effective radius for a representative cirrus (cloud A) and a DCC composed of ice particles only (cloud B). Both profiles are divided into 30 layers for the implementation in the radiative transfer simulation. Parameters used to set up both cloud A and B are summarized in Table 3. Forward radiative transfer simulations are performed to calculate spectral upward radiance above the cloud using an adding/superposition technique from the cloud top (CT) to the cloud base (CB) as described by Platnick (2000). Solar zenith angle $\theta_0$ of 37° for cloud A and 26° for cloud B is used according to the solar position during the measurements.

### 5.3 Vertical weighting function

The vertical photon transport depends on the absorption characteristics at the considered wavelengths. With increasing absorption the probability of a photon being scattered back out of the cloud without being absorbed decreases. Thus, utilizing different near-infrared wavelengths with different absorption characteristics in the retrieval will result $r_{\mathrm{eff}}$ from different altitudes in the cloud (King et al., 2013). To quantify this effect, the vertical weighting function $w_{\mathrm{m}}$ is investigated. $w_{\mathrm{m}}$ describes the contribution of each cloud layer to the absorption considering multiple scattering (Platnick, 2000). Therefore, it can be used to characterize the cloud level where the retrieved $r_{\mathrm{eff}}$ is most representative. $w_{\mathrm{m}}$ for nadir observation as a function of optical thickness $\tau$ is expressed by:

$$w_{\mathrm{m}}(\lambda, \tau, \tau_c, \mu_0, r_{\mathrm{eff}}) = \left| \frac{\mathrm{d}I(\lambda, \tau, \mu_0, r_{\mathrm{eff}})}{\mathrm{d}\tau} \right| \cdot \frac{1}{\int_0^{\tau_c} \left| \frac{\mathrm{d}I(\lambda, \tau, \mu_0, r_{\mathrm{eff}})}{\mathrm{d}\tau} \right|} , \qquad (6)$$

$I$ is the radiance above the cloud and $\tau_c$ is the total cloud optical thickness. Platnick (2000) showed that $w_{\mathrm{m}}$ can be used to estimate the retrieved value of effective radius $r^*_{\mathrm{eff,w}}$ (so-called weighting-estimate) from a given profile of $r_{\mathrm{eff}}(\tau)$ by:

$$r^*_{\mathrm{eff,w}}(\lambda, \tau_c, \mu_0, r_{\mathrm{eff}}) = \int_0^{\tau_c} w_{\mathrm{m}}(\lambda, \tau, \tau_c, \mu_0, r_{\mathrm{eff}})\, r_{\mathrm{eff}}(\tau)\, \mathrm{d}\tau , \qquad (7)$$





$w_m$ calculated for of cloud A and B are shown in Fig. 10c and 10d, respectively. For cloud A with $\tau_c = 3$, it is found that $w_m$ for $\lambda = 1240$ nm and 1640 nm are almost homogeneously distributed along the entire profile. Each cloud layer has nearly equal contribution to the absorption, and therefore to the retrieved $r_{eff}$. Whereas for cloud B with $\tau_c = 15$, the upper cloud layers contribute most to the absorption. Consequently, they strongly influence the retrieved $r_{eff}$. For both $w_m$ profiles, the peak for

5  $\lambda = 1640$ nm is found closer to the cloud top, while for $\lambda = 1240$ nm the peak lies deeper in the cloud. This illustrates, that a retrieval of $r_{eff}$ using $\lambda = 1640$ nm results in a $r_{eff}$ which represents particle sizes located in higher altitude compared to $\lambda = 1240$ nm. For the two idealized cloud cases (cloud A and B), this would in general lead to $r_{eff,1640} < r_{eff,1240}$. Additionally, the results from both cases show clearly that each cloud layer has a contribution to the absorption. Therefore, it should be noted that retrieved $r_{eff}$ from this remote sensing technique does not represent an effective radius at a single cloud layer only.

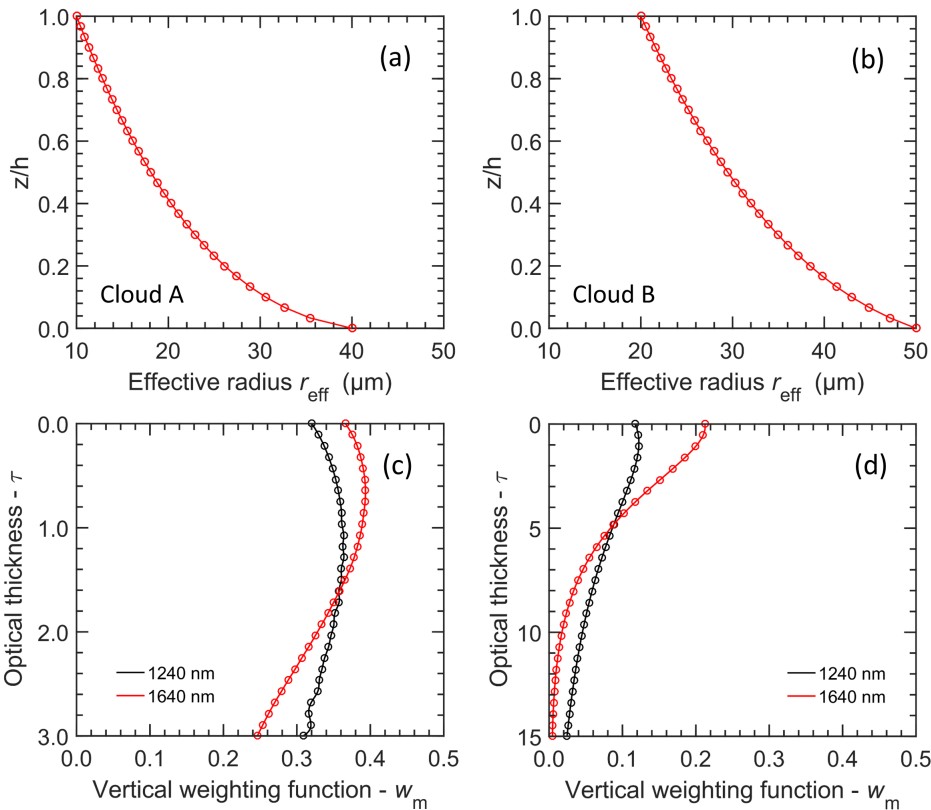

**Figure 10.** (a) is an analytic effective radius profile of a cirrus (cloud A) while (b) is for a DCC composed of ice particles only (cloud B). Detailed specifications of the two analytic profiles are summarized in Table 3. (c) is $w_m$ calculated for cloud A while (d) is for cloud B.

10  Fig. 11a shows the spectral $w_m$ calculated for cloud A (cirrus) in the wavelength range between 1000 - 2000 nm, while Fig. 11b displays the single scattering albedo $\omega_o$ of ice particles (GHM) with $r_{eff}$ of 10 μm and 15 μm. The $\omega_o$ strongly depends on $r_{eff}$ and describes the degree of absorption at each individual wavelength by cloud particles. The $\omega_o$ is smaller for larger particles, and therefore the absorption is higher. The spectral $w_m$ at each individual cloud layer clearly shows a wavelength dependence.





At a wavelength where $\omega_o$ is small, and therefore the absorption is high, the peak of $w_m$ lies closer to the cloud top. In contrast, for a wavelength with $\omega_o \approx 1$ (small absorption), $w_m$ in the lower layers significantly increases while the maximum $w_m$ is reduced. The spectral $w_m$ also shows that spectral measurements in the near-infrared wavelengths affords more information on the particle sizes located in different cloud altitudes.

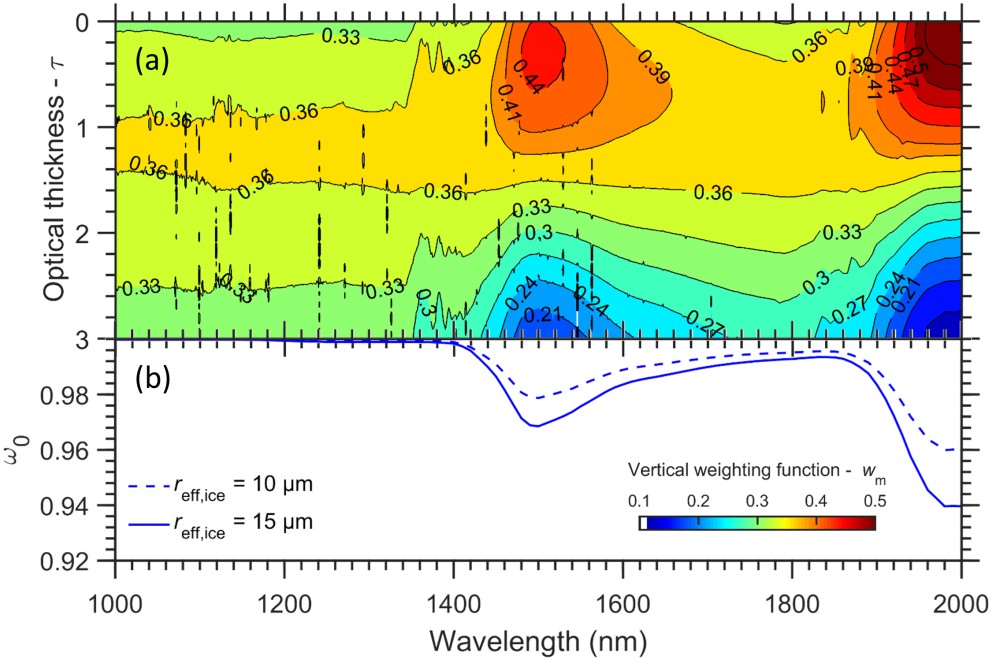

**Figure 11.** (a) Spectral vertical weighting function calculated for cloud A (cirrus) in Fig. 10a. (b) Single scattering albedo $\omega_o$ of ice particles with an effective radius of 10 μm (dashed line) and 15 μm (solid line) using the general habit mixture particle (GHM) by Baum et al. (2007).

It is found, that $w_m$ is a function of the cloud profile itself. Assuming a vertically homogeneous profile in the forward simulation will result in different $w_m$ compared to assuming a realistic profile. Consequently, this may lead to discrepancies in the retrieved cloud properties between both assumptions. With help of $w_m$, possible impacts are investigated by comparing the weighting-estimate $r^*_{eff,w}$ and the retrieved effective radius $r_{eff,ret}$ using $\lambda = 1240$ nm and 1640 nm. Radiances above cloud A and B calculated for the entire cloud layer $I^{\uparrow}_{\lambda,\tau_c}$, as described in Section 5.2, serve as synthetic measurements for the radiance ratio

retrieval. Two combinations, C1(1240 nm) and C2(1640 nm), are employed. The resulting $r^*_{eff,w}$ and $r_{eff,ret}$ are summarized in Table 3. The results from both approaches show, that the $r_{eff}$ derived using $\lambda = 1640$ nm is consistently smaller than using $\lambda = 1240$ nm, which agree with the expectation in such conditions where the particle size decreases toward the cloud top. The absolute deviation between $r_{eff,ret,1240}$ and $r^*_{eff,w,1240}$ is 0.5 μm for both cloud A and B. While between $r_{eff,ret,1640}$ and $r^*_{eff,w,1640}$, the absolute deviation is 0.4 μm for cloud A and 0.1 μm for cloud B.

The comparisons between $r^*_{eff,w}$ and $r_{eff,ret}$ for cloud A and B yield a systematic deviation. It is found, that retrievals using a vertically homogeneous assumption result in an underestimation of $r_{eff,ret}$ compared to $r^*_{eff,w}$ which assumes a realistic cloud





profile with decreasing particle size towards cloud top. For the two realistic profiles (cloud A and B), large particles which have higher absorption are located in the lower layers. Consequently, $w_m$ in the lower cloud layers becomes higher, while $w_m$ for the upper cloud layers is slightly smaller compared to a vertically homogeneous cloud profile (not shown here). However, the impact of vertical profile assumption will decrease for retrievals using wavelengths with higher absorption by cloud particles

such as $\lambda = 1640$ nm.

### 5.4   Heterogeneity of the surface albedo

The accuracy of the surface albedo assumed in the forward simulations influences the uncertainty of the retrieved cloud properties (Rolland and Liou, 2001; Fricke et al., 2014; Ehrlich et al., 2017). For vertically homogenous clouds, these studies found uncertainties of up to $83\,\%$ and $62\,\%$ in the retrieved values of $\tau$ and $r_{eff}$, respectively, when an inaccurate surface albedo is

assumed in the forward simulation. In the tropical rainforest, such as observed during the ACRIDICON-CHUVA campaign, the heterogeneity of the surface albedo can be high where forested and deforested areas are located close to each other. Fig. 12a shows a photo of Amazonian surface taken from HALO during the campaign. Several surface types are classified in a small scale area such as (1) forest, (2) dry-land, (3) water body, and (4) wet-land associated with different surface albedos. For airborne measurements over this area, this can lead to sudden changes of the surface albedo along the flight path. Therefore, a

representative assumption of homogeneous surface for the whole flight leg is not appropriate.

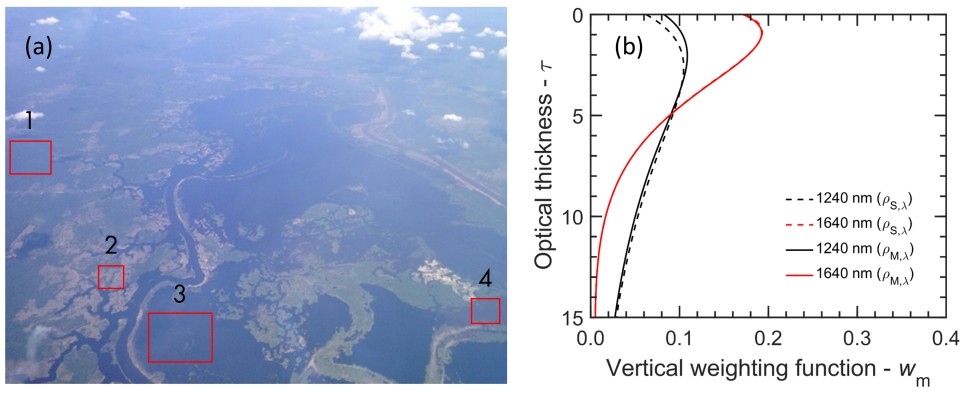

**Figure 12.** (a) Picture of Amazonian surface taken during the ARIDICON-CHUVA campaign. Four surface types are classified such as (1) forest, (2) dry-land, (3) water body, and (4) wet-land. (b) $w_m$ at $\lambda = 1240$ nm (black) and 1640 nm (red) calculated using two spectral surface albedos $\rho_\lambda$ assumed in the forward simulation. $\rho_{S,\lambda}$ is measured by SMART (dashed line), while $\rho_{M,\lambda}$ is derived from the MODIS BRDF/Albedo product (solid line).

Satellite remote sensing is the most practical way to consistently map the surface albedo (Peng et al., 2017). Therefore, for retrievals of DCC over Amazon, the MODIS BRDF/Albedo product is used. Both MODIS-Terra and MODIS-Aqua are used to generate this product in 500 meter resolution. It combines registered, multi-date, multi-band, atmospherically corrected surface reflectance data from the MODIS and MISR instruments to fit a Bidirectional Reflectance Distribution Function (BRDF) in





seven spectral bands consisting of three visible bands centered at $\lambda = 460$ nm, 555 nm, and 645 nm, and four near-infrared bands centered at $\lambda = 865$ nm, 1240 nm, 1640 nm, and 2130 nm (Strahler et al., 1999). The spectral surface albedo derived from the MODIS BRDF/albedo product $\rho_{M,\lambda}$ centered at $\lambda = 645$ nm (a), 1240 nm (b), and 1640 nm (b) are shown in Fig. 13. The values of $\rho_{M,\lambda}$ indicate that the observed DCC was situated above a heterogeneous vegetation surface.

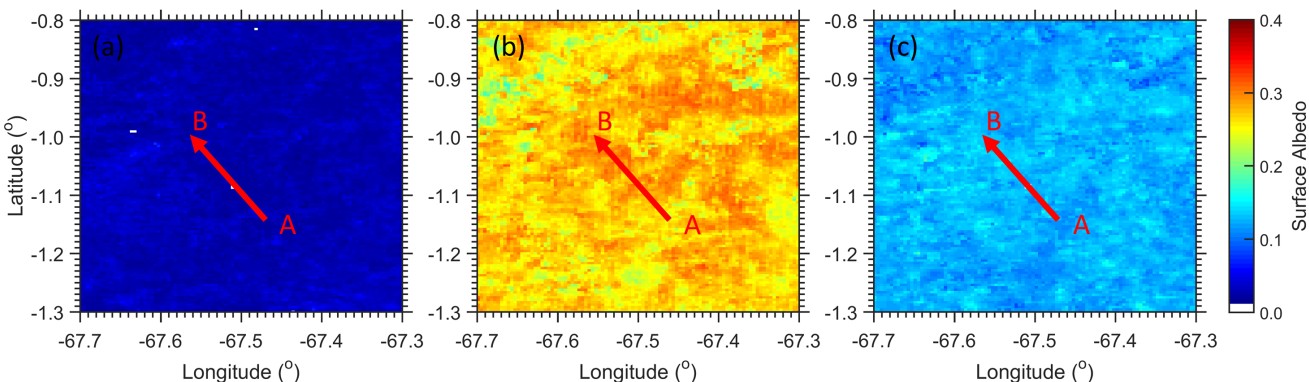

**Figure 13.** Spectral surface albedo derived from the MODIS BRDF/Albedo product $\rho_{M,\lambda}$ centered at $\lambda = 645$ nm (a), 1240 nm (b), and 1640 nm (c). The red arrow indicates HALO flight legs from point A to B during the DCC measurements.

The impact of different surface albedo assumptions on the vertical weighting function $w_m$ is investigated. Cloud B specified in Table 3 is chosen for the calculations to represent an anvil of DCC situated over a heterogeneous rainforest surface. $w_m$ are calculated for two spectral surface albedos $\rho_\lambda$ assumed in the forward simulation. First, a spectral surface albedo of forest was measured by SMART $\rho_{S,\lambda}$ during the ACRIDICON-CHUVA campaign, which results in $\rho_{S,645} = 0.04$, $\rho_{S,1240} = 0.30$, and $\rho_{S,1640} = 0.12$. Second, a spectral surface albedo is derived from the MODIS BRDF/Albedo product $\rho_{M,\lambda}$. For this purpose,

$\rho_{M,\lambda}$ is averaged along HALO flight legs during the DCC measurements, which results in $\rho_{M,645} = 0.06$, $\rho_{M,1240} = 0.21$, and $\rho_{M,1640} = 0.15$. Fig. 12b shows $w_m$ at $\lambda = 1240$ nm (black) and 1640 nm (red) calculated for both $\rho_\lambda$. The dashed line describes $w_m$ calculated for $\rho_{S,\lambda}$, while the solid line is for $\rho_{M,\lambda}$. The result shows, that the impact for $\lambda = 1640$ nm is negligible because radiation is stronger absorbed by cloud particles and not transmitted to the surface and back. Whereas, changes in $\rho_\lambda$ slightly shift $w_m$ at $\lambda = 1240$ nm where sufficient radiation is transmitted through the cloud and can interact with the surface.

In general, the maximum weighting at cloud top is reduced and shifted to lower altitude when $\rho_\lambda$ is increased. Furthermore, the lower cloud layers are now weighted higher due to the enhanced reflection of transmitted radiation back to the cloud base eventually reaching the sensor above cloud top. For higher $\rho_\lambda$, even if the correct $\rho_\lambda$ has been considered, the retrieved $r_{eff}$ is located lower because the $w_m$ is smaller towards cloud top. Therefore, for cloud B with decreasing particle size towards cloud top, assuming a higher $\rho_\lambda$ will result in a larger retrieved $r_{eff}$ than for assuming a smaller $\rho_\lambda$. The opposite result is expected

for clouds where the particle size decreases toward the cloud top, e.g. adiabatic liquid water clouds.



## 5.5 Impact of underlying liquid water cloud

The changes of the vertical weighting function $w_\mathrm{m}$ due to the presence of liquid water cloud below cloud A and B are investigated. Therefore, the calculations of $w_\mathrm{m}$ for cloud A and B presented in Section 5.3 are repeated by adding a liquid layer cloud in the radiative transfer simulations. For cloud A, the liquid water cloud is located between 1.5 - 2 km with $\tau = 8$ and

5    $r_\mathrm{eff} = 10$ µm according to the cloud properties observed during the cirrus case. For cloud B, the liquid water cloud is located between 5 - 6 km with $\tau = 15$ and $r_\mathrm{eff} = 15$ µm, which represents a DCC topped by an anvil of ice particles, while the lower core of DCC is assumed to be liquid water particles only. For simplification, the profiles of liquid water cloud are assumed to be vertically homogeneous. For comparison, $w_\mathrm{m}$ are calculated and normalized for the ice cloud only. Fig. 14a and 14b show $w_\mathrm{m}$ at $\lambda = 1240$ nm (black) and 1640 (red) nm calculated for cloud A and cloud B in a condition with (solid line) and without

10   (dashed line) the presence of liquid water cloud. Additionally, the single scattering albedo $\omega_0$ of ice (blue) and liquid water (red) particles with $r_\mathrm{eff}$ of 10 µm (dashed line) and 15 µm (solid line) is displayed in Fig. 14c.

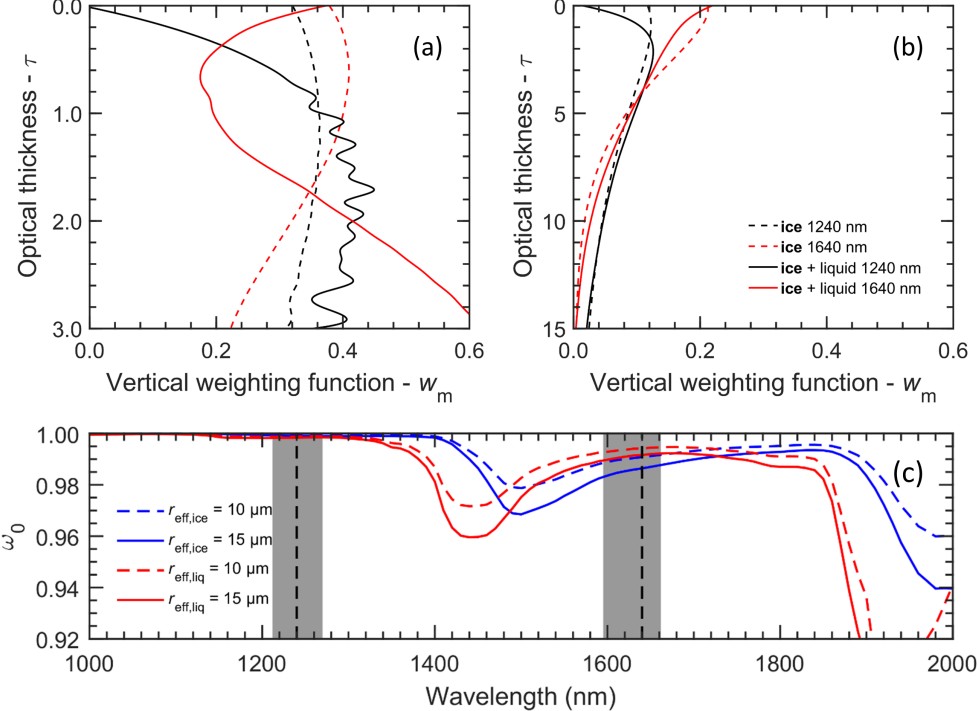

**Figure 14.** (a) is $w_\mathrm{m}$ at $\lambda = 1240$ nm and 1640 nm calculated for cloud A, while (b) is for cloud B. Solid line and dashed line describe $w_\mathrm{m}$ calculated with and without the presence of underlying liquid water cloud, respectively. (c) is single scattering albedo $\omega_0$ of ice (GHM) and liquid water particle with $r_\mathrm{eff}$ of 10 µm and 15 µm.

In general, it is expected that a liquid water cloud changes $w_\mathrm{m}$ similar to a bright surface as described in Section 5.4, where it reflects solar radiation stronger than a dark surface such as water and forest. Fig. 14a and 14b show, that this indeed holds for




the $w_\mathrm{m}$ at $\lambda = 1240$ nm where scattering by cloud particles dominates. In both clouds A and B, the maximum of $w_\mathrm{m}$ is shifted to lower altitudes due to multiple reflection of radiation between surface and cloud base. As a consequence, this will result in an increase of the retrieved $r_\mathrm{eff}$. However, for $\lambda = 1640$ nm $w_\mathrm{m}$ changes differently when adding a liquid cloud below the ice cloud. The changes of $w_\mathrm{m}$ for cloud A are significantly larger compared to cloud B. This pattern results from the stronger

absorption by the ice particles at $\lambda = 1640$ nm. The ice cloud does not transmit sufficient radiation to have a strong interaction with the low level cloud, which leads to almost similar $w_\mathrm{m}$ for the optically thick cloud B with $\tau_\mathrm{c} = 15$. However, for optically thin cloud A with $\tau_\mathrm{c} = 3$, $w_\mathrm{m}$ is modified at the cloud top due to the underlying liquid water cloud. Here the different particle phase and size of the liquid water cloud layer lead to a reduction of the upward radiance $I_\lambda^\uparrow$ when an ice cloud is added to the simulations. Due to the small liquid water particles with high $\omega_0$ at $\lambda = 1640$ nm, the liquid water cloud alone reflects stronger

than together with the ice cloud which adds large ice crystals characterized by a lower $\omega_0$ reducing the total reflectivity. This decrease of $I_\lambda^\uparrow$ strongly contributes to the $w_\mathrm{m}$ close to cloud top, while at about $\tau = 1$ a minimum of $w_\mathrm{m}$ is observed where $I_\lambda^\uparrow$ changes only slightly. Below $\tau = 1$ (lower altitudes), the impact of the liquid water cloud vanishes and scattering by the ice particles increases $I_\lambda^\uparrow$ again corresponding to higher $w_\mathrm{m}$ at cloud base. In general, a similar pattern is imprinted in the $w_\mathrm{m}$ of the optically thick cloud B but not relevant for the entire $w_\mathrm{m}$ due to the higher $\tau_\mathrm{c}$ of the ice cloud. This demonstrates, that

for optically thick clouds, like the DCC case investigated in this study, a retrieval assuming ice cloud only can be applied to retrieve $r_\mathrm{eff}$ of the upper most cloud layer, even if liquid water clouds are present below the ice layer at cloud top.

### 5.6 Optical thickness and effective radius retrieved by SMART and MODIS

Time series of $\tau$ and $r_\mathrm{eff}$ retrieved from SMART and MODIS radiance measurements, as from the MODIS cloud product are compared for the two cloud cases, cirrus and DCC. The MODIS cloud product MYD06_L2 (Platnick et al., 2003, 2017)

provides three different $r_\mathrm{eff}$ which are retrieved using different near-infrared wavelengths centered at $\lambda = 1640$ nm, 2130 nm, and 3700 nm (so-called $r_\mathrm{eff,1640}$, $r_\mathrm{eff,2130}$, and $r_\mathrm{eff,3700}$). However, the quality of $r_\mathrm{eff,1640}$ is limited due to problems of the detectors of MODIS-Aqua band 6. Therefore, $r_\mathrm{eff,1640}$ is not considered in the comparison. The spectral radiance at $\lambda = 2130$ nm and 3700 nm which is used for the MODIS cloud product, $r_\mathrm{eff,2130}$ and $r_\mathrm{eff,3700}$, are not covered by the SMART measurements for both cloud cases. However, the $w_\mathrm{m}$ at $\lambda = 2130$ nm is very similar to $w_\mathrm{m}$ of $\lambda = 1640$ nm (not shown here).

For typical cloud profiles as analyzed in Section 5.3, the differences of retrieved $r_\mathrm{eff,2130}$ and $r_\mathrm{eff,1640}$ are less than 0.5 µm. Therefore, $r_\mathrm{eff,2130}$ can be employed to compare the MODIS cloud product and the radiance ratio retrieval using C2 ($r_\mathrm{eff,1640}$). For observations over land, the MODIS algorithm combines $\lambda = 645$ nm and 2130 nm (C3 - combination 3) to retrieve $\tau$ and $r_\mathrm{eff,2130}$, respectively. While over ocean, it combines $\lambda = 858$ nm and 2130 nm (C4 - combination 4) to retrieve the respective cloud properties.

Time series of cirrus optical thickness and effective radius retrieved using C1, $\tau_\mathrm{ci,C1}$ and $r_\mathrm{eff,ci,C1}$, are presented in Fig. 15a and 15b, respectively. Note that the corrected $I_\mathrm{M,1240}^\uparrow$ by Eq. 3 have applied in the MODIS radiance ratio retrieval. $\eta$ describes the mean $\pm$ standard deviation of the corresponding cloud properties along the selected time series with a subscript of "S" for SMART and "M" for MODIS. To quantify the agreement of the retrieved cirrus properties based on SMART and MODIS, the normalized mean absolute deviation $\zeta$ is calculated. A $\zeta_{\tau_\mathrm{ci,C1}}$ of $0.7\,\%$ and a $\zeta_{r_\mathrm{eff,ci,C1}}$ of $2.5\,\%$ is obtained. Fig. 15c and





15d show time series of cirrus optical thickness and effective radius retrieved using C2, $\tau_{ci,C2}$ and $r_{eff,ci,C2}$, respectively. A $\zeta_{\tau_{ci,C2}}$ of $0.5\%$ and a $\zeta_{r_{eff,ci,C2}}$ of $2.3\%$ are obtained. The analysis shows, that deviations between SMART and MODIS in the retrieved cloud properties are only slightly enhanced by the non-linearity in the retrieval algorithm. Cloud properties derived from MYD06_L2 (blue) are also shown in Fig. 15c and 15d, where $\eta$ with a subscript of "L" describes the respective mean $\pm$

5    standard deviation along the selected time series.

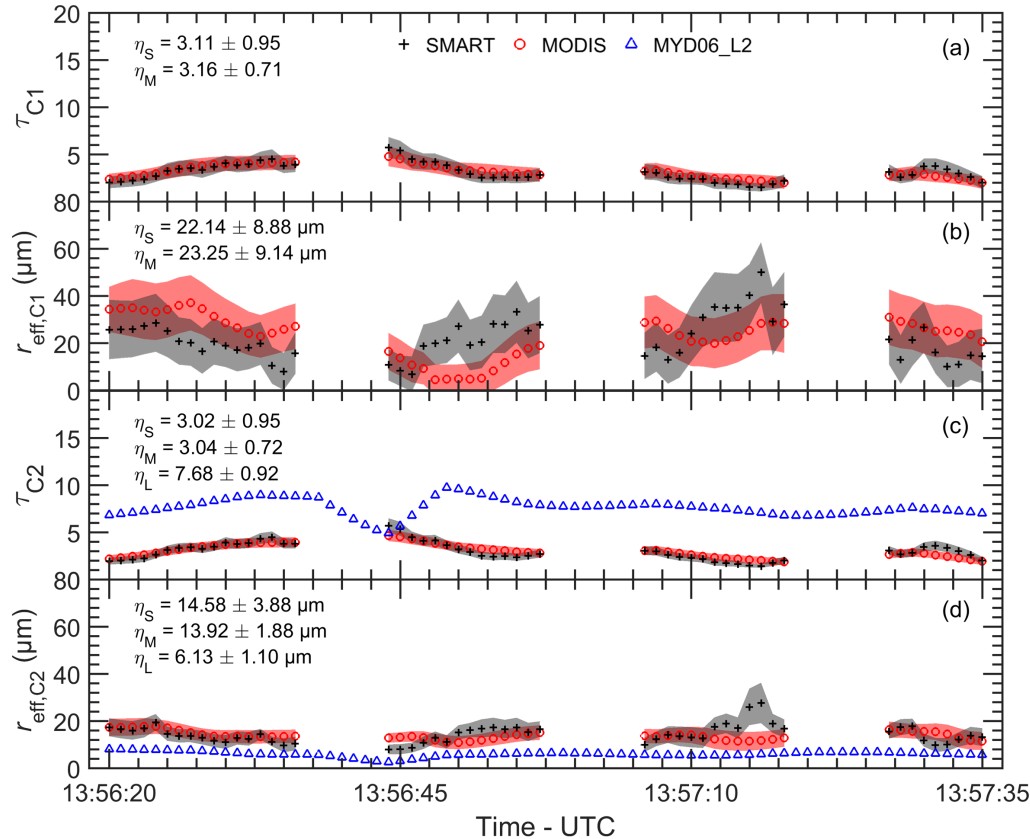

**Figure 15.** Time series of cirrus $\tau$ (a) and $r_{eff}$ (b) retrieved from SMART (black) and MODIS (red) using combination 1 (C1). Dark shaded area describes retrieval uncertainties. $\eta_S$ (SMART) and $\eta_M$ (MODIS) represent the mean $\pm$ standard deviation along time series. (c) and (d) are the respective properties retrieved using combination 2 (C2). Cloud products derived from MYD06_L2, $\tau$ and $r_{eff,2130}$, are shown in blue (only in panel c and d). $\eta_L$ represents the mean $\pm$ standard deviation of the derived properties from MYD06_L2 along time series.

Retrieved cirrus properties using combinations C1 and C2 are compared to C4. Along the selected time series, all combinations show that $\tau_{ci}$ is homogeneous in the observed area, which is indicated by the small values of standard deviation $\sigma_{\tau_{ci}}$ only of up to 1. However, it is found that $\tau_{ci,C4}$ derived from the MODIS cloud product yields a significant overestimation (see Fig. 15c). The absolute deviation between the mean $\overline{\tau}_{ci,C4}$ and $\overline{\tau}_{ci,C2}$ is up to 4.7 or nearly $160\%$. In the MODIS cloud product, the

10   retrieval is performed with an assumption of single layer cloud even if multilayer clouds are reported (Platnick et al., 2017).




However, the liquid water cloud underlying cirrus increases the reflected upward radiances in the visible wavelengths (Finger et al., 2016). Assuming a single layer cloud in the retrieval consequently results in a large overestimation on the retrieved $\tau_{ci}$ because the increase of reflectivity is considered to result from the cirrus alone. Including a low level liquid water cloud as in the radiance ratio retrieval applied to SMART and MODIS, more realistic $\tau_{ci}$ are obtained. Furthermore, it is found that the

5    dependency between retrievals of $\tau$ and $r_{eff}$ leads to a small difference between $\tau_{ci,C1}$ and $\tau_{ci,C2}$. Fig. 9 shows, that the lookup tables of $\tau$ tilt to the right. Consequently, for a larger value of $r_{eff}$, it will result in a larger value of $\tau$. While for a smaller value of $r_{eff}$, it will result in a smaller value of $\tau$. For a cirrus cloud where the particle size decreases toward the cloud top, it is expected that $r_{eff,C1} > r_{eff,C2}$. This finally leads to higher optical thickness retrieved by the combination C1, $\overline{\tau}_{ci,C1} > \overline{\tau}_{ci,C2}$.

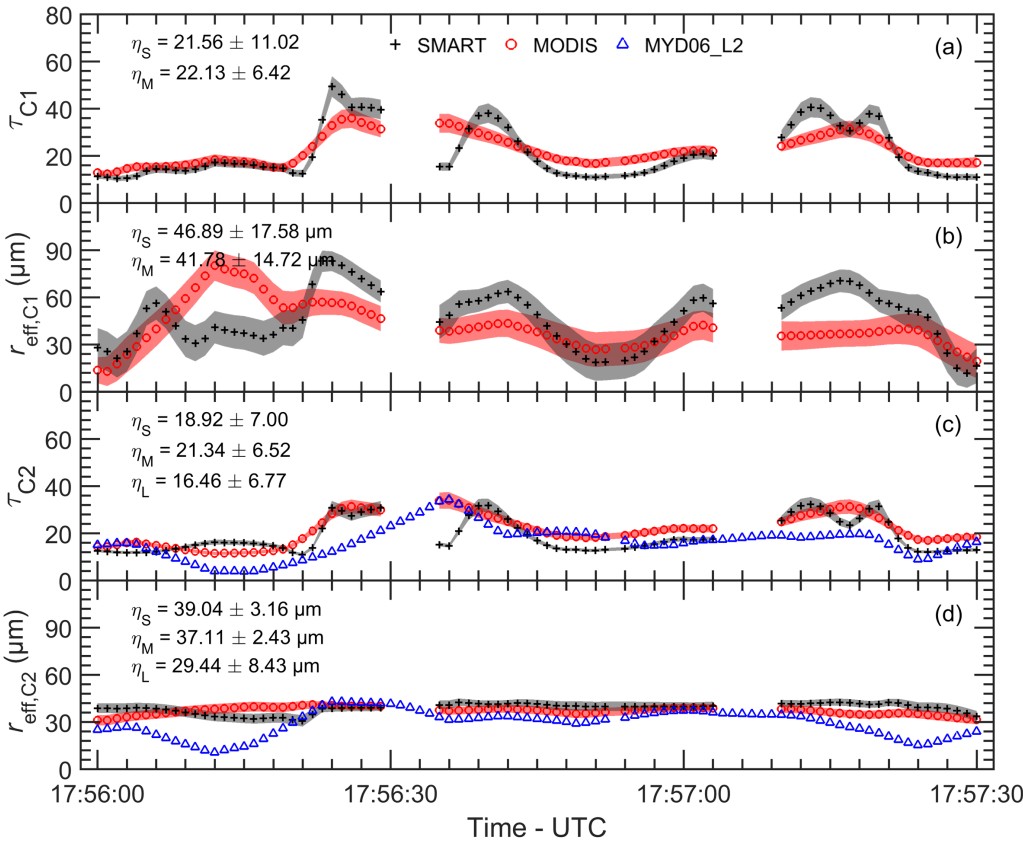

**Figure 16.** Same as Fig. 15 but for the DCC.

The results from all approaches show that $\overline{r}_{eff,ci,C1} > \overline{r}_{eff,ci,C2} > \overline{r}_{eff,ci,C4}$. By neglecting the underlying liquid water cloud

10    which is characterized by small liquid water particles, $\overline{r}_{eff,ci,C4}$ underestimates the particle size of the cirrus cloud. The difference between $r_{eff,C1}$ and $r_{eff,C2}$ results from the different $w_m$ as discussed in Section 5.3, the $w_m$ of $\lambda = 1640$ nm is shifted towards cloud top compared to $\lambda = 1240$ nm and this causes $r_{eff,C1} > r_{eff,C2}$ for cirrus with decreasing particle size towards cloud top. Additionally, the results show that the standard deviation $\sigma_{r_{eff,ci,C1}} > \sigma_{r_{eff,ci,C2}} > \sigma_{r_{eff,ci,C4}}$. This phenomena indi-





cates that more homogeneous particle sizes lie in the higher cloud layers, while mixture particle sizes are located in the lower cloud layers due to size sorting and the increased fall speeds of larger particles in ice clouds (van Diedenhoven et al., 2016). Smaller particles with low sedimentation velocity remain in the higher altitudes, while larger particles with faster sedimentation velocity drop into the layers below. The results also show, that $\Delta r_{\mathrm{eff,ci,C1}} > \Delta r_{\mathrm{eff,ci,C2}}$, which confirms that retrievals of $r_{\mathrm{eff}}$ using a wavelength with a smaller absorption by cloud particles will result in a higher uncertainty. Additionally, it is found that increasing $\tau$ and $r_{\mathrm{eff}}$ has a positive correlation with increasing $\Delta\tau$ and $\Delta r_{\mathrm{eff}}$. This is due to decreasing sensitivity in the lookup tables for larger $\tau$ and $r_{\mathrm{eff}}$.

Time series of DCC optical thickness and effective radius retrieved using C1, $\tau_{\mathrm{dcc,C1}}$ and $r_{\mathrm{eff,dcc,C1}}$, are shown in Fig. 16a and 16b, respectively. The resulting $\zeta_{\tau_{\mathrm{dcc,C1}}}$ of $5.1\,\%$ and $\zeta_{r_{\mathrm{eff,dcc,C1}}}$ of $17.5\,\%$ are obtained. Compared to the cirrus case, the larger value of $\zeta_{r_{\mathrm{eff,dcc,C1}}}$ indicates a strong microphysical properties evolution in the deeper layer of DCC. Fig. 16c and Fig. 16d show time series of DCC optical thickness and effective radius retrieved using C2, $\tau_{\mathrm{dcc,C2}}$ and $r_{\mathrm{eff,dcc,C2}}$. A $\zeta_{\tau_{\mathrm{dcc,C2}}}$ of $6.1\,\%$ and a $\zeta_{r_{\mathrm{eff,dcc,C2}}}$ of $2.6\,\%$ are obtained. In addition of the fast cloud evolution, larger 3-D radiative effects are likely influencing the observations, which can enhance the deviations of the retrieved cloud properties between SMART and MODIS. The cloud properties derived from MYD06_L2 (blue) are also presented in Fig. 16c and 16d. For this case, the MYD06_L2 algorithm uses C3 (over land). The standard deviation values $\sigma_{\tau_{\mathrm{dcc}}}$ from approach C1, C2, and C3, which are up to 11.1, indicate that $\tau_{\mathrm{dcc}}$ is heterogeneous except in the anvil region. The DCC anvil (noticed as cirrus) is detected between 17:56:00 - 17:56:20 UTC and is characterized by relatively smaller $\tau$ between 8 - 20. Later, $\tau_{\mathrm{dcc}}$ increases sharply corresponding to the DCC core and decreases again toward the cloud edge. Looking at each instrument, it is found that $\overline{r}_{\mathrm{eff,dcc,C1}} > \overline{r}_{\mathrm{eff,dcc,C2}} > \overline{r}_{\mathrm{eff,dcc,C4}}$. This indicates, that the particle size in the DCC also decreases toward the cloud top. Additionally, it is found that $\sigma_{r_{\mathrm{eff,dcc,C1}}} > \sigma_{r_{\mathrm{eff,dcc,C2}}}$ and $\sigma_{r_{\mathrm{eff,dcc,C2}}} < \sigma_{r_{\mathrm{eff,dcc,C2}}}$. This condition yields, that the particle sizes are more homogeneous in the level of $r_{\mathrm{eff,dcc,C2}}$ compared to the level of $r_{\mathrm{eff,dcc,C1}}$ (lower cloud layer) and $r_{\mathrm{eff,dcc,C3}}$ (higher cloud layer).

## 6   Comparison with in situ measurement

The retrieved effective radius is compared to the in situ effective radius for the cirrus case. Here, the terminology of $r_{\mathrm{eff,z}}$ is used to describe the in situ effective radius sampled at a specific vertical layer $z$, while the effective radius retrieved using a remote sensing technique $r_{\mathrm{eff}}$ represents a bulk value. The number size distribution measured by the CCP has been converted into the particle effective radius with 1 Hz temporal resolution (Weigel et al., 2016). A binning method is applied to gain the profile of cirrus effective radius with a 20 m vertical resolution. Fig. 17a shows, that the CCP detected a cirrus cloud between 10.7 and 12 km, where the mean values (solid line) range between 3 - 30 µm. The shaded area illustrates measurement uncertainties of the in situ observations. The smallest particle with $r_{\mathrm{eff}} = 3.1$ µm is found at the cloud base $z_{\mathrm{b}} = 10.7$ km and increases rapidly up to 30.2 µm at $z = 10.8$ km. Later, $r_{\mathrm{eff}}$ decreases reaching a value of 8.4 µm at the cloud top $z_{\mathrm{b}} = 11.97$ km.

To compare in situ and retrieved effective radius, the vertical weighting function $w_{\mathrm{m}}$ has to be considered. A comparison at a single layer is not appropriate because each individual cloud layer contributes to the absorption, as discussed in Section 5.3. The $w_{\mathrm{m}}$ is calculated using the profile of effective radius $r_{\mathrm{eff,z}}$ and ice water content (IWC) measured by in situ instru-





ments. Furthermore, the profile of effective radius $r_{\mathrm{eff,z}}$ is convoluted with the $w_{\mathrm{m}}$ to calculate the in situ weighting-estimate $r_{\mathrm{eff,w}}^{*}$ using Eq. 7 to allow a comparison with the retrieved effective radius $r_{\mathrm{eff}}$. Similarly, the weighting-altitude $z_{\mathrm{w}}^{*}$, which characterizes the altitude of weighting-estimate and retrieved effective radius can be calculated by:

$$z_{\mathrm{w}}^{*}(\lambda,\tau_c,\mu_0,r_{\mathrm{eff}}) = \int\limits_{0}^{\tau_{\mathrm{c}}} w_{\mathrm{m}}(\lambda,\tau,\tau_c,\mu_0,r_{\mathrm{eff}})\, z\, \mathrm{d}\tau, \qquad (8)$$

where $z$ is the cloud altitudes. Due to different absorption characteristics in the wavelength, it is expected that $z_{\mathrm{w}}^{*}$ varies for different near-infrared wavelengths used in the retrieval. The stronger the absorption by cloud particles in the wavelength, the higher the $z_{\mathrm{w}}^{*}$.

**Table 4.** The mean $\pm$ standard deviation $\eta$ of cirrus effective radius determined by in situ weighting-estimate (CCP) and retrievals (SMART, MODIS, and MYD06_L2) using near-infrared wavelengths between 1240 nm - 3700 nm. The wavelengths have been sorted in order that the degree of absorption by cloud particles increases to the right. $z_{\mathrm{w}}^{*}$ is the weighting-altitude.

| $\lambda$ | 1240 nm | 1700 nm | 1640 nm | 2130 nm | 1550 nm | 1500 nm | 3700 nm |
|---|---|---|---|---|---|---|---|
| $\eta_{\mathrm{CCP}}$ (µm) | $19.0 \pm 9.8$ | $18.3 \pm 9.6$ | $18.0 \pm 9.5$ | $17.5 \pm 9.4$ | $17.0 \pm 9.3$ | $16.7 \pm 9.3$ | $7.0 \pm 5.0$ |
| $\eta_{\mathrm{SMART}}$ (µm) | $22.1 \pm 8.9$ | $19.7 \pm 7.1$ | $14.6 \pm 3.9$ | - | $13.9 \pm 2.7$ | $14.8 \pm 2.7$ | - |
| $\eta_{\mathrm{MODIS}}$ (µm) | $23.3 \pm 9.1$ | - | $13.9 \pm 1.9$ | - | - | - | - |
| $\eta_{\mathrm{MYD06}}$ (µm) | - | - | - | $6.3 \pm 1.2$ | - | - | $4.8 \pm 3.7$ |
| $z_{\mathrm{w}}^{*}$ (km) | 11.39 | 11.42 | 11.44 | 11.46 | 11.48 | 11.49 | 11.89 |

The comparison of in situ weighting-estimate $r_{\mathrm{eff,w}}^{*}$ and the mean value of retrieved effective radius $r_{\mathrm{eff}}$ are presented in Fig. 17b by symbols. Horizontal error bars represent the standard deviation, while vertical error bars are the estimated uncertainty of

weighting-altitude $\Delta z_{\mathrm{w}}^{*}$. $\Delta z_{\mathrm{w}}^{*}$ is estimated by the standard deviation of $z_{\mathrm{w}}^{*}$ calculated with different ice crystal shapes assumed in the forward simulations, which results in a value of 40 m. Additional $r_{\mathrm{eff}}$ retrieved by use of additional wavelengths of SMART at $\lambda$ = 1500 nm, 1550 nm, and 1700 nm are applied in the comparison. By use of these additional wavelengths of SMART allows to enhance the vertical resolution of retrieved $r_{\mathrm{eff}}$. In addition, the MODIS cloud product (MYD06_L2), $r_{\mathrm{eff,2130}}$ and $r_{\mathrm{eff,3700}}$, are also employed in the comparison. The $w_{\mathrm{m}}$ for $\lambda$ = 3700 nm is calculated using solar radiation only

and does not account thermal emissivity.

In general, results from in situ weighting-estimates, SMART and MODIS retrievals, and MODIS cloud products (MYD06_L2) show that the particle size in the observed cirrus decreases toward the cloud top. Additionally, the results also confirm that although retrievals of effective radius using multi near-infrared wavelengths result in particle sizes from different cloud altitudes, this conventional retrieval technique only provides information on the cloud-top effective radius. This is due to the fact, that

the retrieved $r_{\mathrm{eff}}$ represents a vertically weighted value where cloud top layers are weighted at most, which is in agreement





with analyses reported by Chang and Li (2002), Chang and Li (2003), Zhang et al. (2010), King and Vaughan (2012), , King et al. (2013), and van Diedenhoven et al. (2016).

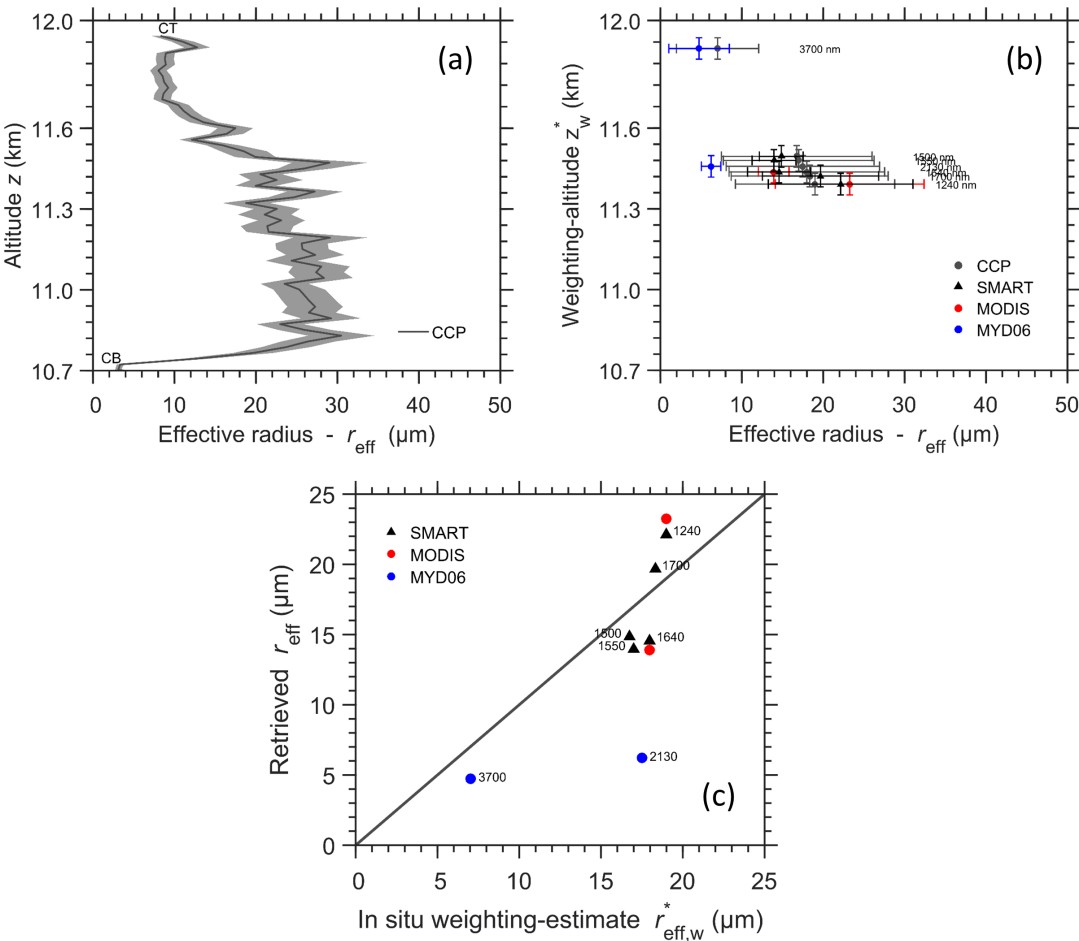

**Figure 17.** (a) Profile of effective radius $r_{\mathrm{eff,z}}$ determined from in situ measurements (solid line), while the grey area represents the uncertainties of in situ observations. (b) Comparison of in situ weighting-estimate (CCP) and the mean value of retrieved effective radius (SMART, MODIS, and MYD06) for $\lambda$ between 1240 nm - 3700 nm. Horizontal error bars represent the standard deviation, while vertical error bars are $\Delta z_{\mathrm{w}}^{*}$. (c) Scatter plot between in situ weighting-estimate and the mean value of retrieved effective radius. The grey line is the one-to-one line, while the labels at each data point describe the wavelength used in the retrievals.

Table 4 summarizes the mean $\pm$ standard deviation $\eta$ of effective radius from $r_{\mathrm{eff}}$ and $r_{\mathrm{eff,w}}^{*}$, and the weighting-altitude $z_{\mathrm{w}}^{*}$ for multi near-infrared wavelengths between 1240 nm - 3700 nm. To quantify the agreement between in situ weighting-estimate and retrieved effective radius, the normalized mean absolute deviation $\zeta$ is calculated. The resulting $\zeta$ between the $r_{\mathrm{eff,w}}^{*}$ and the mean value of SMART retrievals is 7.6 %, 3.6 %, 10.5 %, 9.9 %, and 6.1 % for $r_{\mathrm{eff,1240}}$, $r_{\mathrm{eff,1700}}$, $r_{\mathrm{eff,1640}}$, $r_{\mathrm{eff,1550}}$, and $r_{\mathrm{eff,1500}}$, respectively. The resulting $\zeta$ between the $r_{\mathrm{eff,w}}^{*}$ and the mean value of MODIS retrievals is 10.1 % and 12.7 % for





$r_{\mathrm{eff},1240}$ and $r_{\mathrm{eff},1640}$, respectively. While between the $r^{*}_{\mathrm{eff,w}}$ and the mean value of MODIS cloud products (MYD06_L2), the resulting $\zeta$ is $47.5\,\%$ and $19.3\,\%$ for $r_{\mathrm{eff},2130}$ and $r_{\mathrm{eff},3700}$, respectively. The large deviation between $r^{*}_{\mathrm{eff,w},2130}$ and the MODIS cloud product $r_{\mathrm{eff},2130}$ is due to the presence of liquid water cloud, where the MODIS cloud product does not consider it. However, the influence is almost negligible for $r_{\mathrm{eff},3700}$ due to strong absorption by cloud particles at $\lambda = 3700$ nm. Except the

MODIS cloud product $r_{\mathrm{eff},2130}$, the values of $\zeta$ range between $3.6 - 19.3\,\%$, which agree within the standard deviation.

Fig. 17c shows a scatter plot of in situ weighting-estimate and retrieved effective radius. The symbols represent which data is compared to the in situ. The grey line represents the one-to-one line. The result shows, that there is only a small correlation between the variation of in situ and retrieved effective radius, which is in agreement with analyses reported by King et al. (2013). The deviation between in situ and retrieved effective radius depends on the choice of near-infrared wavelength used

in the retrieval algorithm. The variability of particle size distributions and the presence of liquid water cloud underlying cirrus are considered as potential error contributors. In addition, the simplification in the retrieval forward simulations which assume a vertically homogeneous cloud is also considered to cause the discrepancies between in situ and retrieved effective radius, which is in agreement with the finding discussed in Zhang et al. (2010) and Nagao et al. (2013). This argument is confirmed by the profile of cirrus effective radius measured by in situ, which clearly show in-cloud vertical inhomogeneity, as shown in

Fig. 17a. Therefore, a vertically homogeneous assumption in the retrieval forward simulation is not appropriate.

## 7    Conclusions

Accurate solar radiation measurements are necessary to obtain a high-quality cloud products, e.g., cloud optical thickness $\tau$ and particle effective radius $r_{\mathrm{eff}}$, from satellite remote sensing. Small measurement uncertainties propagate and may potentially amplify in the retrieval algorithm. Airborne-satellite validation is one option to access the retrieval uncertainties. The cases for

a comparison of airborne and satellite derived cloud products have to be selected carefully, e.g., cloud top altitudes, time delays should be minimized, cloud shadows should be discarded, and identical observation geometries of aircraft and satellite should be guaranteed.

A validation of upward (cloud-reflected) radiance and cloud products of MODIS-Aqua was performed for the case of cirrus and DDC using airborne SMART measurements during the two HALO campaigns, ML-CIRRUS and ACRIDICON-CHUVA.

For cirrus measurements, it is found that MODIS radiances centered at $\lambda = 1240$ nm are systematically overestimated to those measured by SMART. The slope of linear regression between SMART and MODIS radiances centered at $\lambda = 1240$ is calculated, which results in a value of 0.86. This value is used to reduce MODIS radiance measurements centered at $\lambda = 1240$ nm. Comparisons of the mean value of upward radiance centered at $\lambda = 645$ nm, 1240 nm, and 1640 nm yield a normalized mean absolute deviation with a maximum value of $1.36\,\%$ for the cirrus case and $6.7\,\%$ for the DCC case. The higher deviations

in case of DCC are related to the fast cloud evolution, which already significantly changed the cloud properties within the 1 min time delay between SMART and MODIS. In addition, larger 3-D radiative effects are considered.

The MODIS cloud products were evaluated by airborne remote sensing and in situ measurements using different approaches. A radiance ratio retrieval is applied to SMART and MODIS nadir upward radiances to retrieve $\tau$ and $r_{\mathrm{eff}}$, as well as to investigate





the propagation of measurement uncertainties into the retrieval. Two combinations, C1 ($I^{\uparrow}_{645}$ and $\Re_1$) and C2 ($I^{\uparrow}_{645}$ and $\Re_2$) are applied. Using the ratios, measurement uncertainties due to the radiometric calibration of the sensor cancel, and therefore the retrieval uncertainty is smaller compared to the usual bi-spectral method. Using different near-infrared wavelengths with different absorption characteristics by cloud particles in the retrieval algorithm provides $r_{\text{eff}}$ from different cloud altitudes. A

retrieval using C1 represents $r_{\text{eff}}$ from a deeper cloud layer, while using C2 yields $r_{\text{eff}}$ from a higher cloud layer. Therefore, those different combinations can be used to investigate the vertical variability of particle sizes in the cloud. However, using C1 in the retrieval algorithm results in larger uncertainties in the retrieved cloud properties due to the small absorption by cloud particles at $\lambda = 1240$ nm.

  The impact of assuming vertically homogeneous cloud in the retrieval is investigated. For ice clouds where the particle size

typically decreases toward the cloud top, retrievals assuming a vertically homogeneous cloud lead to an underestimation in the retrieved $r_{\text{eff}}$. In a sensitivity study for an exemplary ice cloud with cloud base effective radius $r_{\text{eff,b}}$ of 40 µm, cloud top effective radius $r_{\text{eff,t}}$ of 10 µm and total optical thickness $\tau_c$ of 3, an absolute deviation of 0.5 µm and 0.4 µm is obtained when using $\lambda = 1240$ nm and 1640 nm, respectively. While for an ice cloud with $r_{\text{eff,b}} = 50$ µm, $r_{\text{eff,t}} = 20$ µm, and $\tau_c = 15$, an absolute deviation of 0.5 µm and 0.1 µm is obtained when using $\lambda = 1240$ nm and 1640 nm, respectively. The results

show, that the impact is larger for a retrieval using a wavelength where cloud particles absorb less radiation. In this case, more radiations are transmitted in the lower cloud layers increasing the contribution of the lower cloud layers. The vertical weighting function has shown, that each individual cloud layer has a contribution to the absorption imprinted in the radiance reflected above cloud top. Furthermore, it is found that the profile of weighting function depends on the cloud profile itself and the chosen wavelength. Therefore, it should be noted that $r_{\text{eff}}$ retrieved using this remote sensing technique do not represent $r_{\text{eff}}$ at

a single cloud layer only. The retrieved $r_{\text{eff}}$ describes a single bulk value, which represents the entire cloud layer.

  It is found that a higher surface albedo does change the vertical weighting function by increasing the weighting of the lower cloud layers. An enhanced surface reflections increases the interaction of radiation with the lower cloud layers and shifts the vertical weighting towards lower altitudes. Consequently, the retrieved $r_{\text{eff}}$ will change for different surface albedos assumed in the forward simulation. For ice clouds where the particle size increases toward the cloud top, the retrieved value of $r_{\text{eff}}$ will

increase above a high reflecting surface. As observed during the ACRIDICON-CHUVA campaign, the surface heterogeneity in the Amazonian rainforest is high, where forested and deforested areas are located close to each other. In this condition, the surface albedo can change suddenly along the flight path. Therefore, in this study the MODIS BRDF/Albedo product is used, which consistently maps the spectral surface albedo over land surfaces.

  The presence of liquid water cloud underlying cirrus leads to significant discrepancies on the retrieved cirrus properties. In

general, the liquid water cloud acts similar to a bright surface, however the impact depends on the cloud properties and the wavelengths used in the retrieval. The liquid water cloud enhances the radiance at visible wavelengths, which results to an overestimation of retrieved cirrus $\tau$. While, the impact on the retrieved cirrus $r_{\text{eff}}$ is shown by changes of the vertical weighting function. When the cirrus $\tau$ is optically thin, radiation is transmitted through the cirrus and reflected by the low liquid water cloud back to the cirrus. Consequently, the contribution of the lower cloud layers to the absorption and the vertical weighting

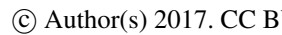



function at cloud base is enhanced. For typical cirrus where the particle size decreases toward the cloud top, the retrieved cirrus $r_{\mathrm{eff}}$ becomes larger when a liquid water cloud is present below the cirrus.

For the cirrus case, the normalized mean absolute deviation of retrieved cloud properties from SMART and MODIS using combination C1 and C2 is found in the range of up to $0.5\,\%$ for $\tau$ and $2.5\,\%$ for $r_{\mathrm{eff}}$. These deviations are only slightly larger

than the deviations found for the upward radiance comparisons. This indicates, that the errors are only slightly enhanced by the non-linearity in the retrieval algorithm. The cirrus $\tau$ derived from the MODIS cloud product results to a significant overestimation of up to $160\,\%$ compared to the retrievals using C1 and C2. This is due to the presence of liquid water cloud, which is not considered by the MODIS cloud product. For the DCC case, it results of up to $6.1\,\%$ for $\tau$ and $17.5\,\%$ for $r_{\mathrm{eff}}$. In this case, the fast cloud evolution is the major issue, as well as larger 3-D radiative effects. The dependency between the

retrieval of $\tau$ and $r_{\mathrm{eff}}$ is analyzed, which leads to small discrepancies in the retrieved $\tau$ between all approaches. For both cloud cases, it is found that the particle size decreases toward the cloud top. Mixture particle sizes are observed in the lower cloud layers, while more homogeneous particle sizes are located in the higher cloud layers.

For the cirrus case, the effective radius retrieved by remote sensing technique (SMART, MODIS, and MYD06_L2) is compared to the effective radius measured by the in situ instrument (CCP). The terminology of $r_{\mathrm{eff,z}}$ is introduced to describe the profile

of effective radius measured by in situ sampled at a specific altitudes $z$, while the remote sensing effective radius $r_{\mathrm{eff}}$ represents a single bulk value. To compare in situ and retrieved effective radius, the vertical weighting function has to be considered. The vertical weighting function is calculated using the profiles of particle effective radius and ice water content measured by in situ. To calculate a particle size comparable to the retrieved effective radius, the in situ weighting-estimate $r^{*}_{\mathrm{eff,w}}$ is calculated by convoluting the profile of effective radius with the vertical weighting function. Additional near-infrared wavelengths of

SMART at $\lambda$ = 1700 nm, 1550 nm, and 1550 nm are employed in the radiance ratio retrieval to increase the information extracted from the spectral measurements of SMART and the vertical resolution of the retrieved $r_{\mathrm{eff}}$. Except the $r_{\mathrm{eff,2130}}$, the resulting normalized mean absolute deviation between in situ and retrieved effective radius varies between $3.6 - 19.3\,\%$, depending on the chosen wavelength and agree within the standard deviation value. The large deviation on the $r_{\mathrm{eff,2130}}$ which is up to $48\,\%$ is due to the presence of liquid water cloud, which is not considered by the MODIS cloud product. The variability of

particle size distributions, the presence of liquid water cloud below the cirrus, and the simplification in the retrieval algorithm by assuming a vertically homogeneous cloud in the forward radiative transfer simulation are considered as the potential error contributors.

Additionally, the weighting-altitude $z^{*}_{\mathrm{w}}$ which characterizes the altitude of retrieved $r_{\mathrm{eff}}$ was calculated by convoluting the altitudes with the vertical weighting function. For wavelengths characterized by a high absorption by cloud particles, $z^{*}_{\mathrm{w}}$ are

located in higher altitudes compared to wavelengths dominated by scattering. However, this conventional retrieval technique provides information only on the cloud-top effective radius because the retrieved $r_{\mathrm{eff}}$ represents a vertically weighted value where cloud top layers are weighted at most. Further studies have to be performed to develop an advanced method combined with spectral measurements of SMART, which has the potential to reconstruct the vertical profile of cloud microphysical properties. Simultaneous airborne and satellite remote sensing, and airborne in situ observations analyzed in this study for





the two cases illustrate the need of well calibrated and carefully collocated measurements to develop, test, and validate cloud remote sensing methods.

*Acknowledgements.* This work was supported by the Max Planck Society (MPG), the German Science Foundation (DFG) funding the SPP HALO 1294 and the grants of WE 1900/35-1 and VO 1504/4-1, the German Aerospace Center (DLR), and the FAPESP (Sao Paulo Research Foundation) grants 2009/15235-8 and 2013/05014-0. The HGF is acknowledged for supports under the contract number W2/W3-60). The author acknowledges the Ministry of Research, Technology and Higher Education of the Republic of Indonesia (RISTEKDIKTI) and the German Academic Exchange Service (DAAD) for the research grant under the scheme of Indonesia-German Scholarship Programme (IGSP). The entire ML-CIRRUS and ACRIDICON-CHUVA project team is gratefully acknowledged for collaboration and support.



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
