# Peer review of "Comparing airborne and satellite retrievals of cloud optical thickness and particle effective radius using a spectral radiance ratio technique: Two case studies for cirrus and deep convective clouds"

_Atmospheric Chemistry and Physics, 2017_

## Referee Comment (RC1) · Anonymous Referee #1 · 26 Oct 2017

This is a review of the paper "Comparing Airborne and Satellite Retrievals of Optical and Microphysical Properties of Cirrus and Deep Convective Clouds using a Radiance Ratio Technique" submitted to ACPD by Krisna et al.

The paper describes a study on remote sensing of ice cloud optical thickness and ice particle size. It aims to compare airborne and satellite remote sensing measurements with each other and with in situ measurements. Much attention is given to the sensitivity of the particle size retrievals to the vertical variation of ice sizes.

[Figure]

While the paper contains some interesting parts, I am struggling to see the general motivation of the study. The introduction mentions the validation of satellite remote sensing measurements and retrievals. These are indeed very important, but the main case study selected in this paper seems to be one of the worst situations for this, namely a thin cirrus over a liquid cloud. Operational retrievals using MODIS or other instruments (including SMART) will indeed not be able to account for the liquid clouds and will be biased. Accounting for a liquid cloud using additional information as is attempted in the paper is expected to add considerable uncertainty to the cirrus retrievals, making the comparison between in situ and remote sensing measurements not very informative. Any reader would wonder why this particular case is selected.

In addition, the use of MODIS measurements in this study is questionable. Measurements in the 2130nm band are used to 'reconstruct' the 1640nm band measurements using a scaling method that was certainly not design for cloud properties retrievals. The other MODIS band used is the 1240 nm band, but this is scaled in a somewhat ad hoc manner by a factor 0.86, which is rather large, because the data does not agree with the SMART measurements. Regardless where this factor originates from, I find it rather bold to assume without discussion that the MODIS values need to be corrected instead of the SMART measurements. Also, the influence of this scaling on the retrieved effective radius should also have been discussed. Finally, the operational MODIS retrievals of effective radius at 2130 nm are included, but these are known to be affected by the lower liquid cloud, so I do not see the relevance of including these.

Parts of the study on the vertical weighting function are interesting. Also, the comparisons between remotely sensed ice effective radius and the in situ measurements are remarkably good despite the lower liquid clouds and all the other caveats discussed above. This means that either the lower lying liquid cloud properties happen to be chosen well in this case or the properties of the liquid cloud (in particular droplet size) do not affect the effective radius retrievals of the upper layer that much. The latter explanation may be interesting and should then be further investigated in the paper.

In its current form, the paper is not suited for publication, mainly because of the reasons listed above. I aimed to suggest changes to the paper to make it suitable for publication, but ended up with a long list. If all of these issues are addressed the paper might be suitable for publication by ACP.

Below my major comments on this paper are listed followed by some detailed minor comments.

1) The introduction rightfully states that validation of remote sensing retrievals of cloud properties is important and that accounting for the vertical variation of ice sizes is also important. However, the introduction fails to motivate the present study using the selected cases. The authors should argue convincingly why the two discussed cases are selected. The presence of the liquid cloud under the cirrus should be mentioned in the introduction and it should be argued why this and the DCC case are interesting cases for the evaluation of satellite remote sensing results.

2) Many of the references discussed in the introduction (page 3, lines 7-31) are about liquid clouds, while this study focusses on ice clouds. The influence of vertical variation on remote sensing of drop and ice sizes are very different. Please focus the discussion on ice clouds and remove references that focus on liquid clouds.

3) MODIS data is introduced in section 3.2. I assume the latest collection 6 data (level 1 and 2) is used? If so, please state that in the paper. If not, then please use collection 6 for the study.

4) Although the wavelength range of SMART is said to extend to 2200, the 2130 MODIS bands is not considered to be in its range. (This is stated rather late in the paper and should be brought forward.) The 1.64 MODIS band is selected instead, but this band has many unreliable detectors. Therefor a scaling function is used to scale 2.13 micron measurements to mimic 1.64 micron measurements. This scaling function was developed to apply a snow detection algorithm, and was never intended to be applied to cloud measurements and microphysical retrievals. One could argue that the method

may work for ice clouds, because of the similarity of snow and ice surfaces, but this is not shown anywhere. I suggest to use the remaining detectors of the 1.64 band to verify the applicability of this method. Alternatively, would the remaining 1.64 micron detectors not be enough for you study?

5) The data filter described in section 4.2 is based on the cirrus case, while it is stated that the DCC case is more variable in time. Would a separate data filter for the DCC case be not more appropriate? Please include the DCC points in figure 2, or add two additional panels to this figure for the DCC case. Is a better agreement for the DCC case obtained if a stricter time difference is used? Please revise the paper to address these points.

6) The SMART and MODIS radiances are directly compared in section 4.3. The measurements at 1.24 micron are different by a factor 0.86, which is rather large. As stated earlier, this scaling should be discussed more and not directly be assumed to be owing to MODIS calibration errors without a proper reference. I do not know of any record about the 1240 band being biased by such an amount, although the 1240 nm band is used for several products. SMART is on an aircraft with atmosphere above it, causing possible biases in the derived reflectances. This is actually the reason why the radiance ratio method is used. So, I would think SMART is more uncertain than MODIS. Also, these biases may be very different between the two cases. In addition, please discuss (and investigate) the influence of this scaling on the resulting effective radius retrievals.

7) The general habit mixture of Baum et al. is used for the retrievals. Please add the level of surface roughness that is applied (is it severely rough?). Also, discuss the sensitivity of the ice size and optical thickness retrievals to the choice of optical model. Refer to, e.g., Holz et al. (2016, https://doi.org/10.5194/acp-16-5075-2016) and/or Van Diedenhoven et al. (2014; J. Geophys. Res. Atmos., 119, 11,809–11,825, doi:10.1002/2014JD022385.)

8) To account for the liquid layer, in section 5.1 it is stated that "the properties of liquid water cloud are estimated by comparing simulated and measured spectral radiance averaged over the selected time series, where the reff of liquid water cloud agrees with values of in situ climatological data reported in e.g., Miles et al. (2000)." Firstly, please give some more information on the technique to obtain the optical thickness using the measured spectral radiances. Should you not have knowledge on the ice cloud optical thickness for that? Also, either here or in section 5.5, please discuss the influence of the estimated optical depth and effective radius of the ice cloud layer on the ice cloud retrievals. I am sure the cloud properties would be variable over the investigated flight leg. How are the ice cloud size retrievals affected when instead the liquid cloud is assumed to consist of, e.g., 5 or 15 micron drops? What is the uncertainty on the optical thickness estimate and how does that affect the ice cloud retrievals? The influence of these assumptions on the weighting functions are discussed in section 5.5, but please also show the influence on the retrieved ice cloud properties.

9) In section 5.3, a rather interesting investigation on the weighting functions is shown. At the end, it is stated that the assumption of a homogeneous layer in the retrievals leads to a systematic deviation. This is reiterated in the conclusions and section 6. However, this deviation is found to be smaller that 1 micron for the investigated cases. That can be considered quite small. Please stress this in section 5.3 and in the conclusions, as it strikes me as a good validation for the use of homogeneous layers.

10) The comparison with in situ measurements is interesting and an important part of the paper. However, it is unclear how effective radius is derived from the in situ measurements. Effective radius is proportional to the volume (or mass) over the projected area of the ice crystals. The CCP probes do not measure mass/volume per particle (there exists no probe that does that). I believe crystal area could be derived from the probe. Is there a separate IWC measurement? Is there an area-mass relationship used? Please explain how effective radius is derived and what the uncertainty might be.

11) In addition to the previous point, it is not clear how the weighting function is applied to the in situ measurements. The weighting function is in terms of optical depth from cloud top, while the in situ measurements are derived at various physical depths within the cloud. How is physical depth converted to optical depth? Is there an extinction measurement made? Please explain in the paper.

12) I find it rather pointless and confusing to include the operational MODIS 2130 nm results in the analysis of section 6. It is clear that the lower liquid cloud is causing a bias in the ice effective radius retrievals. It is interesting though that the 3.7 retrievals are not much affected by the liquid layer. Please remove the 2130 nm results here.

13) In section 6, it is stated that "there is only a small correlation between the variation of in situ and retrieved effective radius which is in agreement with analyses reported by King et al. (2013).". I do not agree really. When the 2130 point is removed (which should be done), the correlation seems pretty good, especially considering the difference between 3.7 and the rest of the point, as well as all the uncertainties discussed above. What is the correlation coefficient? Also, the ranges shown on the in situ measurements are rather large, and all retrievals fall within them, which could be considered a good comparison. Please discuss this in more detail. Also, the King et al. reference is about liquid clouds, which have much greater extinction, minimizing the information on vertical structure in the various bands. This reference is not relevant for ice clouds. Please remove this reference here.

14) Section 6 ends with the statement that "a vertically homogeneous assumption in the retrieval forward simulation is not appropriate", which is also not backed up by the simulations shown, which show a <1 micron biases caused by the homogeneous layer. Please change or remove this sentence and refer to the simulations instead.

15) The conclusions section is pretty long and detailed. I suggest to summarize the general conclusions without going into too many details. Also, rewrite the conclusions according to all the changes made related to the above points.

Minor comments:

Somewhere in the paper, give a definition of effective radius of cloud ice.

Page 3, line 32: Please define the SMART acronym on first use in the text.

Section 5: how high was HALO flying and how high were the clouds. Was is clear above the HALO aircraft?

Page 5, line 13: Irradiance is misspelled.

Page 9, line 22: I believe you mean 1640 instead of 2130 here.

Page 13, line 12: Please give a definition of Ip for completeness.

Page 18, line 3: I believe you mean "offers" instead of "affords".

Page 18, Line 13: Do not start a new sentence at "while". (Same on page 29)

Page 19, line 8: The Platnick et al. (2017) paper is also a good reference for the influence of surface albedo.

Page 21, figure 14: Can the oscillations for the 1240 ice + liquid case be explained?

Page 22, line 22: Refer to the Zhang et al (2010) paper when talking about the differences in ice absorption at 1.6 and 2.13 micron.

Page 25, line 4: What does the Delta symbol represent?

Page 25, line 8 and further. Note the good agreement between SMART and MODIS for the DCC case and give the mean differences, etc. in the same way as the cirrus case was discussed.

Page 28, line 10: In the list of possible uncertainties also note the uncertainties of deriving effective radius from the in situ measurements and the uncertainties caused by unconstrained choice of ice optical model for the retrievals.

---

## Referee Comment (RC2) · Anonymous Referee #2 · 13 Nov 2017

The authors utilize an airborne radiometer to evaluate MODIS ice cloud retrievals and determine the impact of photons vertical penetration in remotely-sensed cloud effective radius. In addition, in-situ aircraft observations appear to partially validate their hypothesis on the value of different wavelengths for assessing the cloud microphysical vertical structure. The manuscript is interesting and the idea that different near-infrared wavelengths provide information about the cloud vertical structure is interesting. However, the sampling is very small and the in-situ observations matched with MODIS and the radiometer SMART are limited to a few points, so any solid statistical inference or

validation of the authors' algorithm (and assumptions) are difficult. Moreover, I do not think it is well-justified the pre-processing of MODIS reflectances as the derivation of a new MODIS 1640 nm using only the 2130 nm reflectance is unphysical. The authors cite a number of papers for justifying the MODIS corrections, but this is a misinterpretation of the literature results. I am struggling with my recommendations because even though the authors show some interesting results, the analysis is not rigorous, and the comparison between MODIS science team retrievals and their own MODIS cloud retrievals is flawed.

Specific comments:

- The derivation of MODIS band 6 using band 7 is unphysical and the justification based on the results in Wang et al (2006) is misleading as Wang et al. shows that the correction is useful for estimating NDVI, which is a completely different problem. The 2130 nm and 1640 nm have distinctive photon vertical penetration (e.g. your figure 11) so the conversion is unphysical. I do not think the paper can be accepted until they authors use the standard MODIS channels without any correction. I agree that the 1640 nm MODIS channel has issues, so I would suggest the direct use of the 2130 nm channel instead.

- The authors correct the MODIS 1240 nm channels using as a justification Lyapustin et al (2014) but, again, the use of this reference is misleading. This paper only corrects MODIS in order to remove a spurious trend. Moreover, the new collection 6 radiances should have incorporated the modifications described in Lyapustin et al. Overall, the use of SMART for correcting MODIS is unjustified. It is much more rigorous to list the differences between SMART and MODIS and then compare the retrievals, keeping in mind the instruments differences.

- A central assumption is the liquid optical depth and effective radius of the layer below the cirrus clouds. The authors choose constant values for each case but these uncertain values can substantially bias the retrievals, and moreover there is no way to know

if the constant values are correct or not. So, any comparison with MODIS standard retrievals will have the huge uncertainty due to the liquid optical depth utilized for creating the lookup tables (the use of climatological values is suspicious). This is very problematic because it is unclear whether the new retrievals are better than MODIS. Overall, to prove the point that accurate ice retrievals depend on the ability of accounting for the cloud layer below the cirrus clouds, you only need synthetic observations, which is why I do not see the point of comparing satellite and airborne remote sensing retrievals.

- I find it surprising that the authors did not retrieve effective radius using MODIS 3.78 um channel. If you want to demonstrate that the liquid cloud layer can bias the retrievals, then you have to calculate the effective radius for all the available MODIS channels.

- In my opinion, the main conclusion of the paper is that accounting for liquid clouds is far more important than accounting for the vertical inhomogeneity.

- I believe collection 6 utilizes a new habit for the lookup table calculations. This new habit appears to produce different results compared to the ice crystal habit of collection 5.

- I do not see the value of section 5.4. If you deem it necessary, please include the section as a part of the appendix.

---

## Author Comment (AC1) · 5 Feb 2018

==================================================================

We thank the reviewer for the time and efforts reading our manuscript and providing valuable comments and suggestions. We have made revisions according to your comments and suggestions, as described below. The reviewer comments are written in roman and while the author responses are in *italic*. The diff file indicating changes in the manuscript is enclosed in the end of this document.

General comments :

This is a review of the paper "Comparing Airborne and Satellite Retrievals of Optical and Microphysical Properties of Cirrus and Deep Convective Clouds using a Radiance Ratio Technique" submitted to ACPD by Krisna et al. The paper describes a study on remote sensing of ice cloud optical thickness and ice particle size. It aims to compare airborne and satellite remote sensing measurements with each other and with in situ measurements. Much attention is given to the sensitivity of the particle size retrievals to the vertical variation of ice sizes.

While the paper contains some interesting parts, I am struggling to see the general motivation of the study. The introduction mentions the validation of satellite remote sensing measurements and retrievals. These are indeed very important, but the main case study selected in this paper seems to be one of the worst situations for this, namely a thin cirrus over a liquid cloud. Operational retrievals using MODIS or other instruments (including SMART) will indeed not be able to account for the liquid clouds and will be biased. Accounting for a liquid cloud using additional information as is attempted in the paper is expected to add considerable uncertainty to the cirrus retrievals, making the comparison between in situ and remote sensing measurements not very informative. Any reader would wonder why this particular case is selected.

In addition, the use of MODIS measurements in this study is questionable. Measurements in the 2130nm band are used to 'reconstruct' the 1640nm band measurements using a scaling method that was certainly not design for cloud properties retrievals. The other MODIS band used is the 1240 nm band, but this is scaled in a somewhat ad hoc manner by a factor 0.86, which is rather large, because the data does not agree with the SMART measurements. Regardless where this factor originates from, I find it rather bold to assume without discussion

that the MODIS values need to be corrected instead of the SMART measurements. Also, the influence of this scaling on the retrieved effective radius should also have been discussed. Finally, the operational MODIS retrievals of effective radius at 2130 nm are included, but these are known to be affected by the lower liquid cloud, so I do not see the relevance of including these.

Parts of the study on the vertical weighting function are interesting. Also, the comparisons between remotely sensed ice effective radius and the in situ measurements are remarkably good despite the lower liquid clouds and all the other caveats discussed above. This means that either the lower lying liquid cloud properties happen to be chosen well in this case or the properties of the liquid cloud (in particular droplet size) do not affect the effective radius retrievals of the upper layer that much. The latter explanation may be interesting and should then be further investigated in the paper.

In its current form, the paper is not suited for publication, mainly because of the reasons listed above. I aimed to suggest changes to the paper to make it suitable for publication, but ended up with a long list. If all of these issues are addressed the paper might be suitable for publication by ACP. Below my major comments on this paper are listed followed by some detailed minor comments.

Response of general comments :

*The reviewer is correct, the limited and not well suited cases investigated in the study are not sufficient to draw general conclusions from the comparison of airborne and satellite cloud retrieval. The limited number of cases results from the careful selection of measurements which allows to evaluate the direct measurement (radiance) and retrieved cloud products. The direct comparison of radiance requires almost perfectly collocated measurement of satellite and aircraft which is given only for few flights of the two investigated campaigns. In addition, inappropriate cloud situations had to be rejected. Specified descriptions about complexities in the case selection will be addressed in the detailed reviewer comments below. In order to avoid the impression, that the comparison is valid for general cirrus and deep convective clouds, we strengthen throughout the revised manuscript, that only a case study is presented. The title is changed to:*

> *"Comparing airborne and satellite retrievals of cloud optical thickness and particle effective radius using a spectral radiance ratio technique: Two case studies for multilayer cirrus and deep convective clouds"*

*The reviewer is right, that the treatment of the MODIS measurements is questionable and not well justified in the manuscript. The motivation is to use identical wavelength for both SMART and MODIS retrievals. Unfortunately, the SMART measurements in the near-infrared only cover two MODIS bands centered at \lambda = 1240 nm and 1640 nm. At \lambda = 2130 nm the uncertainty of*

*SMART is large. Therefore, it could not be included in the study. It is known that MODIS band 6 ($\lambda$ = 1640 nm) has problems with the detector. Using remaining detectors of MODIS band 6 is not possible due to the very limited number of pixel in our cloud cases. Therefore we used the approach by Wang et al. (2006), which indeed was developed for snow surfaces to 'retrieve' MODIS band 6. We think that this approach is justified also for measurements above ice clouds because the optical properties ice clouds are very similar to a snow surface (similar refractive index). To some degree this is confirmed by the agreement between restored MODIS band 6 and SMART as shown the manuscript (Fig. 4c, 5c, and 6b). In the revised manuscript we added the motivation and the method of MODIS band 6 retrieval more clearly.*

*It is true that the correction of the 1240 nm MODIS band can not only be justified by the disagreement with the SMART measurements since SMART might be wrong as well. But we found other indications, that MODIS radiance at 1240 nm is biased in this case. Using the original MODIS radiance band 5 (1240 nm), the cloud retrieval fails because the measurements fall far from the range provided by the forward simulations. Therefore, we used the SMART data as reference in order to apply the correction scheme based on Lyapustin et al. (2014). In the revised manuscript we extended the discussion on this critical issue. The radiances are now included in the radiance lookup tables (Fig. 8) indicating clearly that they do not match the forward simulations.*

*The properties of the liquid water cloud below the cirrus is estimated carefully by varying the properties of both cloud layers and searching the best fit in the spectral, particularly in the water vapor absorption bands and O2 A-band. The detailed technique to estimate the liquid cloud properties will be answered through the major comment (point 8) below. Additionally, we added Sec. 5.2 to discuss the possible uncertainties on the retrieved cirrus properties which can raise from the assumption of liquid properties.*

Major comments:

1) The introduction rightfully states that validation of remote sensing retrievals of cloud properties is important and that accounting for the vertical variation of ice sizes is also important. However, the introduction fails to motivate the present study using the selected cases. The authors should argue convincingly why the two discussed cases are selected. The presence of the liquid cloud under the cirrus should be mentioned in the introduction and it should be argued why this and the DCC case are interesting cases for the evaluation of satellite remote sensing results.

Response:

*The reviewer is correct, the motivation to use both specific cases for a general validation of remote sensing failed in the original manuscript. Due to the multi-layer structure these cases only allow to investigate how satellite retrieval deal with such complex situations. How strong the retrieved properties are influenced by the lower clouds, how the vertical weighting functions differ in multilayer clouds. We therefore shifted the focus of the manuscript into this direction. The title was changed to:*

> *"Comparing airborne and satellite retrievals of cloud optical thickness and particle effective radius using a spectral radiance ratio technique: Two case studies for multilayer cirrus and deep convective clouds"*

*In the introduction we added a discussion on the current approach of satellite retrieval to deal with multi-layer clouds. The two general cases: a cirrus above a liquid cloud and deep convective clouds where a liquid/mixed cloud is topped by an anvil (cirrus cloud) are introduced. Climatology of the occurrence of multi-layer clouds are presented.*

> *"Standard retrieval methods such as MODIS operational retrievals commonly assume a priori, that there is one homogenenous cloud layer with a specific thermodynamic phase, either liquid water or ice (Platnick et al., 2017). However, studies by Hahn et al. (1984) and Warren et al. ( 1985) analyzing ground-based observations reported, that the coexistence of multilayer clouds (e.g., cirrus above liquid water clouds) are found in about 50% of the data, and therefore do not fulfill the assumptions of the retrieval algorithm. Chang and Li (2005) and Sourdeval et al. (2015) have demonstrated, that omitting the low liquid water cloud in the retrieval algorithm will introduce significant uncertainties in the retrieved cirrus properties."*

*In this context, the two cases are well suited to investigate what information satellite retrieval provide for multi-layer clouds.*

2) Many of the references discussed in the introduction (page 3, lines 7-31) are about liquid clouds, while this study focusses on ice clouds. The influence of vertical variation on remote sensing of drop and ice sizes are very different. Please focus the discussion on ice clouds and remove references that focus on liquid clouds.

Response:

*We have removed unnecessary references about liquid clouds. However, we keep King et al. (2013) and Painemal and Zuidema (2011) as the reference for the comparison of in situ and retrieved reff since not many papers did such comparisons. In the recent form, studies by Zhang et al. (2010) and Wang et al. (2009) are cited for the study of cirrus vertical structure.*

> *"For cirrus cloud, Zhang et al. (2010) and Wang et al. (2009) demonstrated that the discrepancy between passive remote sensing and in situ measurements is influenced by the simplification in the retrieval algorithm which assumes in-cloud vertical homogeneity."*

3) MODIS data is introduced in section 3.2. I assume the latest collection 6 data (level 1 and 2) is used? If so, please state that in the paper. If not, then please use collection 6 for the study.

Response:

*All MODIS data, MODIS level 1B calibrated radiance and cloud products, used in this study are collection 6.*

> *"Satellite data used in this study stem from the Level 1B Moderate Resolution Imaging Spectroradiometer (MODIS) - Aqua collection 6."*

> *"The MODIS cloud product collection 6, namely MYD06_L2, provides three different reff …………"*

4) Although the wavelength range of SMART is said to extend to 2200, the 2130 MODIS bands is not considered to be in its range. (This is stated rather late in the paper and should be brought forward.) The 1.64 MODIS band is selected instead, but this band has many unreliable detectors. Therefor a scaling function is used to scale 2.13 micron measurements to mimic 1.64 micron measurements. This scaling function was developed to apply a snow detection algorithm, and was never intended to be applied to cloud measurements and microphysical retrievals. One could argue that the method may work for ice clouds, because of the similarity of snow and ice surfaces, but this is not shown anywhere. I suggest to use the remaining detectors of the 1.64 band to verify the applicability of this

method. Alternatively, would the remaining 1.64 micron detectors not be enough for you study?

*Response:*

*The spectral range covered by SMART is between 300 - 2200 nm. However, the sensitivity of the spectrometers decrease for small and large wavelengths depending on the magnitude of radiation. For measurements used in this study, only the wavelengths range between 400 - 1800 nm provides measurements with reasonable uncertainty. In this way, a direct comparison of the 2130 nm MODIS band is not possible.*

*Using the remaining MODIS band 6 detectors in this study would be not sufficient because only 3 pixels are left and the spatial coverage of the investigated cloud be too coarse. The motivation and technique to retrieve MODIS band 6 is presented in the revised manuscript.*

> *"According to Wang et al. (2006), the MODIS radiance band 6 (IM,B6) can be retrieved using band 7 IM,B7 ($\lambda = 2130$ nm). This technique was originally developed and tested on the basis of snow surface, assuming that the spectral characteristics of the snow reflectivity between MODIS band 6 and 7 does not change significantly for different snow types. Assuming that ice clouds and snow have similar optical properties, the same approach can be applied. Similar to Wang et al. (2006), a parameterization of IM,B6 is developed on the basis of radiative transfer simulations of upward radiance performed for cirrus with different $\tau$ and reff. A polynomial fit is applied to quantify the relation between IM,B6 and IM,B7 which result the parameterization:*
>
> *IM,B6 = -81.033 $IM,B7^2$ + 3.257 IM,B7 + 0.002 "*

> *"The validity of the parameterization is tested using the remaining detectors of MODIS band 6 for observations above cirrus (not shown here). The linear regression between original and retrieved IM,B6 showed differences below 5% (slope of 0.95 and zero bias) with a correlation coefficient of 0.94."*

*To develop the parameterization as shown by the Equation above, the simulations are run for different values of $\tau$ (2-7) and reff (10-45 µm). Fig. a below is the scatter plot between radiance band 6 (1640 nm) and band 7 (2130 nm). The dashed line is the linear regression line. The parameterization is developed by making use of the relation between the two bands. Fig. b is the scatter plot of radiance band 6 original vs. 'retrieved' using the equation above. Here, the retrieval of MODIS band 6 shows a good performance with a slope of 1, no bias, and $R^2 = 1$.*

*Finally, as suggested by the reviewer we compared the radiance from the remaining detectors of MODIS band 6 with the retrieved values to test the validity of MODIS band 6 retrieval in real measurements, as shown in Fig. c. Here we compared the remaining detectors of MODIS band 6 and the retrieved values for measurements above clouds. Again, the result confirms the performance and validity of MODIS band 6 retrieval for cloud measurements.*

[Figure]

5) The data filter described in section 4.2 is based on the cirrus case, while it is stated that the DCC case is more variable in time. Would a separate data filter for the DCC case be not more appropriate? Please include the DCC points in figure 2, or add two additional panels to this figure for the DCC case. Is a better agreement for the DCC case obtained if a stricter time difference is used? Please revise the paper to address these points.

Response:

*In the revised manuscript, we applied the data filter separately as shown in Fig. 2. Indeed, for an equally given threshold, the scatter is larger for the DCC case which is caused by the fast cloud evolution. Therefore, the time difference for the DCC case was reduced to 300 s.*

*"For the DCC (Fig. 2b), the scatter is significantly larger compared to the cirrus for the given threshold of |\Delta t| < 300 s and even worse for the threshold of |\Delta t| > 300 s with \ R^2 = 0.79 and -0.09, respectively. In this case, the horizontal wind speed is smaller with an average of 9 ms^-1, but the fast cloud evolution is the major issue. Luo et al. (2014) and Schumacher et al., (2015) reported, that tropical DCCs located at altitude between 6 - 8 km typically have an updraft velocity about 2 - 4 ms^-1. According to this analysis, the comparisons are restricted to |\Delta t| < 500 s for the cirrus case, while for the DCC case the threshold is tightened to |\Delta t| < 300 s."*

[Figure]

6) The SMART and MODIS radiances are directly compared in section 4.3. The measurements at 1.24 micron are different by a factor 0.86, which is rather large. As stated earlier, this scaling should be discussed more and not directly be assumed to be owing to MODIS calibration errors without a proper reference. I do not know of any record about the 1240 band being biased by such an amount, although the 1240 nm band is used for several products. SMART is on an aircraft with atmosphere above it, causing possible biases in the derived reflectances. This is actually the reason why the radiance ratio method is used. So, I would think SMART is more uncertain than MODIS. Also, these biases may be very different between the two cases. In addition, please discuss (and investigate) the influence of this scaling on the resulting effective radius retrievals.

Response:

*It is correct, that without independent standard, we cannot judge, if MODIS or SMART do measure right or wrong. However, the atmospheric influence on SMART is rather low because the HALO aircraft did fly at altitudes of 12.3 km during the cirrus and 8.3 km during the DCC observations. Only little atmospheric scattering is expected at these altitudes especially at the large wavelength of 1240 nm. The major justification why we corrected MODIS and used SMART as the reference shows up in the cloud retrieval. In the revised manuscript, we added all measured*

*radiances into the LUTs in Fig. 8. Here it is obvious, that the MODIS data does not fall into the solution space of the forward simulations. While the range of radiance I,645 still matches the simulations, the ratio R_1240 = I,1240/ I,645 nm does not. We also could not find any comment about such a bias in literature and also cannot exclude that the forward simulations are biased (effect of ice crystal shape or scattering library). To allow meaningful retrievals with MODIS data, we finally decided to correct the MODIS band 5 (1240 nm). For both cloud cases, we found that the bias is nearly consistent about 12% for the cirrus and 10% for the DCC. Increasing retrieval failure in the cirrus case is related to the larger solar zenith angle, which makes the reff LUTs way denser.*

[Figure]

*A detailed discussion on the scalling is added to the revised manuscript:*

*"The measurements of SMART (black crosses) and MODIS (blue circles) are included for both scenes in Fig. 8. For the C1 which is based on I_1240, the MODIS data does not match the lookup table solution space. The results in Section 3.3 show clearly, that I_M,1240 are higher than I_S,1240 by about 15%. Using the original I_M,1240 for the cirrus case, all the retrievals of reff are fail because the measurements lie far outside the lookup table solution space (see Fig. 8a), while for the DCC case the retrieval failure is smaller (see Fig. 8c). Enhancing retrieval failure in the cirrus case is due to the larger \theta_0. At a larger \theta_0, the upward radiance becomes more insensitive to the changes of reff and consequently the lookup tables are denser. To gain meaningful retrieved cloud properties, a correction of I_M,1240 is applied. Following Lyapustin et al. (2014), a correction factor g is calculated by the slope of linear regression between I_M,1240 and I_S,1240, which results in g = 0.88 for the cirrus case and g = 0.90 for the DCC case. The corrected I_M,1240 (red circles) are added in Fig. 8 and now match the lookup table solution space. Therefore, all following radiance ratio retrievals for the two cloud cases use these corrected I_M,1240."*

7) The general habit mixture of Baum et al. is used for the retrievals. Please add the level of surface roughness that is applied (is it severely rough?). Also, discuss the sensitivity of the ice size and optical thickness retrievals to the choice of optical

model. Refer to, e.g., Holz et al. (2016, https://doi.org/10.5194/acp-16-5075-2016) and/or Van Diedenhoven et al. (2014; J. Geophys. Res. Atmos., 119, 11,809–11,825, doi:10.1002/2014JD022385.)

Response:

*For the retrievals of the cirrus case, we use GHM based on severely roughened aggregates composed of nine habits (Baum et al., 2014), while the ice crystal habit of plate with high surface roughness (Yang et al., 2013) is applied for the retrievals of the DCC case. The assumption of ice crystal habit considers the measurements by in situ probes. The In the revised manuscript, we added a discussion on the impact of using GHM instead of aggregated columns which is based on the suggested literature.*

> *"These particle habits differ from the MODIS collection 6 retrievals which use severely-roughened compact aggregates of solid columns (so-called aggregated columns) by Yang et al. (2013). A sensitivity study infers that the retrievals assuming GHM and plate generally will result in a larger \tau and smaller reff (not shown here), which is in agreement with findings by van Diedenhoven et al. (2014) and Holz et al. (2016)."*

8) To account for the liquid layer, in section 5.1 it is stated that "the properties of liquid water cloud are estimated by comparing simulated and measured spectral radiance averaged over the selected time series, where the reff of liquid water cloud agrees with values of in situ climatological data reported in e.g., Miles et al. (2000)." Firstly, please give some more information on the technique to obtain the optical thickness using the measured spectral radiances. Should you not have knowledge on the ice cloud optical thickness for that? Also, either here or in section 5.5, please discuss the influence of the estimated optical depth and effective radius of the ice cloud layer on the ice cloud retrievals. I am sure the cloud properties would be variable over the investigated flight leg. How are the ice cloud size retrievals affected when instead the liquid cloud is assumed to consist of, e.g., 5 or 15 micron drops? What is the uncertainty on the optical thickness estimate and how does that affect the ice cloud retrievals? The influence of these assumptions on the weighting functions are discussed in section 5.5, but please also show the influence on the retrieved ice cloud properties.

Response:

*As stated by the reviewer, the characterization of the liquid cloud layer is crucial for the retrieval of the cirrus properties and the manuscript did not present this issue properly. Disentangling the contribution of both cloud layers to the total measured radiance is challenging. We used simulations for different combinations of liquid water and cirrus cloud properties and compared the simulated radiance with SMART measurements of the entire spectral range covered by SMART (see*

*figures below). The absorption bands of water vapor (940 and 1135 nm) and the O2 A-band (761 nm) provide some information on the multi-layer structure. Depending on the \tau of the high (cirrus) and low (liquid) cloud layer, absorption features by atmospheric trace gases are stronger or weaker imprinted in the spectral radiance.*

*Fig. a-c indicate, that the best fit in the spectral, especially in the absorption bands of water vapor and O2 A-band, was found for \tau_li = 8 and reff_li = 10 µm combined with \tau_ci = 3 and reff_ci = 15 µm. The reff of the liquid cloud is less relevant when the \tau of the cirrus is sufficiently high. In this way, the spectral range used to derive the reff of the cirrus dominated by scattering/absorption in the cirrus layer only. In Fig. d, we show the impact of the combination between \tau_ci and \tau_li, which can also give some insights by observing changes in the spectral. For this purpose, we hold \tau_ci + \tau_li = constant with fixed reff_ci = 15 µm and reff_li = 10 µm. Here, again we found the best fit in the spectral is given by \tau_ci = 3 and \tau_li = 8. Underestimation / overestimation of \tau_li will produce gaps, particularly in water vapor absorption bands and O2-A band.*

[Figure]

[Figure]

In the revised manuscript, we considered the variability of the liquid cloud properties along the flight paths. To describe this problem more clearly, we added a sensitivity study in Sec. 5.2 with respect to the assumed properties of the liquid cloud which provides an estimation of the retrieval uncertainties due to uncertainties in the assumption of the liquid cloud properties.

*"For the cirrus case, the properties of the low liquid water cloud is assumed to be constant along the flight leg. This assumption might not hold in reality and affect the retrieved cirrus properties. Therefore, the sensitivity of the cirrus retrieval on the assumed properties of the liquid water cloud is quantified using radiative transfer simulations. Spectral radiance are simulated for different combinations of liquid water cloud and cirrus properties. The liquid water cloud is varied for \tau_li = 6 - 10 and reff_li = 6 - 14 µm, while the cirrus is changed for \tau_ci = 2 - 8 and reff_ci = 10 - 40 µm. These simulated radiances are used as synthetic measurements and analyzed with the retrieval algorithm using C2 (I_645 and \Re_1640), which assumes a liquid water cloud with \tau_li = 8 and reff_li = 10 µm. The comparison of synthetically retrieved and original \tau_ci and r_eff,ci is shown in Fig. 9. The annotation of "overestimation" (below one-to-one line) and "underestimation" (above one-to-one line) corresponds to when the retrieval is run with an overestimation and underestimation of the properties of liquid water cloud. The retrieved \tau_ci are analyzed in Fig. 9a for different \tau_li, while r_eff,ci and r_eff,li are fixed to 20 µm and 10 µm, respectively. Similarly, the retrieved r_eff,ci are analyzed in Fig. 9b for different r_eff,li but for a fixed combination of \tau_ci = 3 and \tau_li = 8. In general, the simulations show that an overestimation of \tau_li leads to an underestimation of \tau_ci because in this case, the liquid water cloud contribute stronger to the reflected radiation than in reality. Therefore, a smaller \tau_ci is required to match the measurement, and vice versa. For the range of \tau_ci analyzed here, the retrieved \tau_ci is found be over- or underestimated by 1.3 when in reality \tau_li is 6 or 10, while the retrieval assumes \tau_li = 8. These biases of $\tau_ci show, that \tau_li needs to be estimated accurately because a wrong assumption of tau_li almost directly propagates in uncertainties of \tau_ci.*

*A similar behavior is found for the retrieval of reff_ci, where an overestimation of reff_li leads to an underestimation of reff_ci, and vice versa. Assuming larger*

*liquid droplets than in reality implies that these droplets contribute stronger to the measured absorption at \lambda = 1640 nm, and therefore the ice crystals only contribute less (smaller reff_ci). Fig. 9b illustrates, that the impact of reff_li is strongest when small liquid droplets (reff_li <= 8 µm) are present. For larger liquid droplets (reff_li > 10 µm), the impact is smaller. The maximum uncertainties of reff_ci found for the range of reff_ci and reff_li considered here are about 8 µm for the underestimation of µm which show a tendency of higher uncertainties for higher reff_ci. The retrieval of reff_ci is less affected by reff_li, when the cirrus layer is sufficiently thick (\tau_ci > 5) since the cirrus layer will dominate the reflected radiation in the absorption bands."*

9) In section 5.3, a rather interesting investigation on the weighting functions is shown. At the end, it is stated that the assumption of a homogeneous layer in the retrievals leads to a systematic deviation. This is reiterated in the conclusions and section 6. However, this deviation is found to be smaller that 1 micron for the investigated cases. That can be considered quite small. Please stress this in section 5.3 and in the conclusions, as it strikes me as a good validation for the use of homogeneous layers.

Response:

*If a bias of 1 micron is small or not might depend on the related question. We agree, compared to other retrieval uncertainties this is not the major issue and the vertically homogeneous assumption might be sufficient. In the revised manuscript, we modified the conclusion on the biases introduced by cloud vertical inhomogeneity to:*

> *"The assumption of vertically homogeneous cloud in the retrieval algorithm has only a small impact on the retrieval results."*

10) The comparison with in situ measurements is interesting and an important part of the paper. However, it is unclear how effective radius is derived from the in situ measurements. Effective radius is proportional to the volume (or mass) over the projected area of the ice crystals. The CCP probes do not measure mass/volume per particle (there exists no probe that does that). I believe crystal area could be derived from the probe. Is there a separate IWC measurement? Is there an area-mass relationship used? Please explain how effective radius is derived and what the uncertainty might be.

Response:

*We added more details on the data analysis of the CCP addressing the open questions identified by the reviewer:*

*"The reff from CCP data is derived from the geometrical properties and number of detected particles. Many definitions of reff exist as summarized in McFarquhar and Heymsfield (1998). In general, reff as a measure for the cloud radiative properties is defined as the ratio of the third to the second moment of a size distribution, implying spheres of equivalent cross-sectional area for any cloud particle shape (Frey et al., 2011; Heymsfield and McFarquhar, 1996). The accuracy of the cloud particle sizing is conservatively estimated to be about 10% for spherical particles (Molleker et al., 2014). The sizing uncertainty increases as a function of particles shape complexity (i.e., when dendrites or particles with elevated aspect ratio were predominating)."*

*Indeed the projection area is the basis for the diameter extraction with OAPs, no matter if from crystals, bullets, dentrites or droplets. IWC is not measured by CCP, but there is a separate instrument namely WARAN (Kaufmann et al., 2014; Voigt et al., 2014). The IWC data from WARAN were used to obtain the profile of extinction, which will be discussed in more specifically in point 11 below.*

11) In addition to the previous point, it is not clear how the weighting function is applied to the in situ measurements. The weighting function is in terms of optical depth from cloud top, while the in situ measurements are derived at various physical depths within the cloud. How is physical depth converted to optical depth? Is there an extinction measurement made? Please explain in the paper.

Response:

*In the previous manuscript, the explanation is missing. Therefore, we add the discussion of "How is physical depth converted to optical depth?" in the revised manuscript.*

*We did not measure the extinction directly. Following the method by Wang et al. (2009) and Fu and Liou (1993), we combined in situ IWC measured by the WARAN with in situ reff from the CCP to calculate the extinction. Then, the profile of \tau(z) is obtained by the vertical integration of the extinction from cloud top to the cloud level z.*

> *"Note that the wm in this study is calculated in terms of \tau from cloud top toward cloud base. Therefore, the conversion of geometrical altitude and optical thickness \tau(z) has to be specified and considered in the analysis. For this purpose, IWC(z) measured by WARAN and reff(z) derived from CCP are converted into a profile of the extinction coefficient \beta(z) following the scheme introduced by Fu and Liou (1993) and Wang et al. (2009):*

$$\beta_e(z) \approx \mathrm{IWC}(z) \cdot \left[ a + \frac{b}{r_{\mathrm{eff}}(z)} \right],$$

*where a = -6.656x10$^{-3}$, b = 3.686. \beta(z) is in the unit of m$^{-1}$, IWC(z) in g m$^{-3}$, and reff(z) in μm. Further, the extinction profile is used to calculate \tau(z) by integrating \beta(z) from cloud top to the altitude level z:*

$$\tau(z) = \int\limits_{z}^{z_t} \beta_{\mathrm{e}}(z)\,\mathrm{d}z$$

*Using \tau(z), re(z) can be translated into reff(\tau)……….."*

12) I find it rather pointless and confusing to include the operational MODIS 2130 nm results in the analysis of section 6. It is clear that the lower liquid cloud is causing a bias in the ice effective radius retrievals. It is interesting though that the 3.7 retrievals are not much affected by the liquid layer. Please remove the 2130 nm results here.

Response:

*Reff,2130 from the MODIS cloud product has been removed. We now summarize the MODIS cloud products in Table 4 of the revised manuscript. Therefore, the difference between the original MODIS cloud products which does not distinct between liquid cloud and cirrus and the results of radiance ratio retrieval considering the liquid water cloud below cirrus is still mentioned. More specified descriptions are available in the point 13 below.*

13) In section 6, it is stated that "there is only a small correlation between the variation of in situ and retrieved effective radius which is in agreement with analyses reported by King et al. (2013)."I do not agree really. When the 2130 point is removed (which should be done), the correlation seems pretty good, especially considering the difference between 3.7 and the rest of the point, as well as all the uncertainties discussed above. What is the correlation coefficient? Also, the ranges shown on the in situ measurements are rather large, and all retrievals fall within them, which could be considered a good comparison. Please discuss this in more detail. Also, the King et al. reference is about liquid clouds, which have much greater extinction, minimizing the information on vertical structure in the various bands. This reference is not relevant for ice clouds. Please remove this reference here.

Response:

*It is correct, that including the original MODIS cloud product was not a good idea as this data does not account for the liquid cloud below the cirrus and thus is strongly biased (especially at 2130 nm).*

*In the revised manuscript, we removed the original MODIS cloud product and additionally performed retrievals using MODIS band 7 (2130 nm) and band 20 (3700 nm) using radio retrieval and also considering the liquid water cloud below the cirrus. Now the data is consistent with the retrieval using the shorter wavelength bands.*

*By doing this, significantly improved the correlation of the retrievals results with the in situ weighting-estimate reff,w\*. A normalized mean absolute deviation of \zeta= 8.3% and 1.5% for retrieval using 2130 nm and 3700 nm was obtained. By removing original MODIS cloud product in this analysis, overall, the \zeta between in situ reff,w\* and retrieved reff lies between 1.5 - 10.3% which falls within the standard deviation (variability of horizontal reff) and considerably as a good agreement. The resulting correlation coefficient $R^2$ is 0.82 which shows a robust agreement. We changed the discussion in the manuscript accordingly.*

> *"Additionally, the reff retrieved by using additional SMART measurements at \lambda = 1500 nm, 1550 nm, and 1700 nm, and also MODIS radiances centered at \lambda = 2130 nm and 3700 nm (band 20) are applied in this comparison. The retrieval and the calculation of wm for \lambda = 3700 nm are performed by considering both solar and thermal radiation….."*

> *"The deviations of in situ reff,w$^*$ and SMART reff range between 3.2% (\lambda = 1500 nm) and 10.3% (\lambda = 1550 nm). Between reff,w$^*$and MODIS reff, the \zeta results in a value between 1.5% for \lambda = 3700 nm and 9.1% for \lambda = 1640 nm. Overall, the values of \zeta are in the range between 1.5 - 10.3% and agree within the horizontal standard deviation, as shown in Fig. 15b."*

> *"The reff derived from the MODIS cloud product are obviously affected by the low liquid water cloud, which is not included in the algorithm of MODIS operational retrieval. Therefore, a \zeta of 47.5% and 19.3% are obtained for reff,L,2130 and reff,L,3700, respectively. The absorption by ice crystals at \lambda = 3700 nm is very strong. Consequently, the first top layers will dominate the absorption and significantly reduce the effect of the underlying liquid water cloud. Fig. 15c shows a scatter plot of in situ reff,w$^*$ and reff retrieved from SMART (black triangles) and MODIS (red dots), while the dashed line represents the one-to-one line. There is a robust agreement between in situ reff,w$^*$ and retrieved reff with a correlation coefficient $R^2$ of 0.82."*

*We have removed King et al. (2013).*

14) Section 6 ends with the statement that "a vertically homogeneous assumption in the retrieval forward simulation is not appropriate", which is also not backed up by the simulations shown, which show a <1 micron biases caused by the

homogeneous layer. Please change or remove this sentence and refer to the simulations instead.

Response:

*We have removed this sentence and changed the conclusion to:*

*"The variability of particle size distributions, the uncertainties of deriving reff from the in situ measurements, the presence of liquid water cloud below cirrus, and the uncertainties caused by unconstrained choice of ice crystal shapes for the retrievals, are considered as the main contributor which can reveal the discrepancies between in situ and retrieved reff. The assumption of vertically homogeneous cloud in the retrieval algorithm has only a small impact on the retrieval results."*

15) The conclusions section is pretty long and detailed. I suggest to summarize the general conclusions without going into too many details. Also, rewrite the conclusions according to all the changes made related to the above points.

Response:

*We have reduced and revised the conclusion according to the suggestion by the reviewer and changes which have been made during the revision process.*

**Minor comments:**

Somewhere in the paper, give a definition of effective radius of cloud ice.

*The reff definition has been given in Section 2.1*

*"In general, reff as a measure for the cloud radiative properties is defined as the ratio of the third to the second moment of a size distribution implying spheres of equivalent cross-sectional area for any cloud particle shape (Heymsfield and McFarquhar, 1996; Frey et al., 2011)."*

Page 3, line 32: Please define the SMART acronym on first use in the text.

*The acronym has been given in P.4 L.3*

*"Measurements of spectral solar radiation using the Spectral Modular Airborne Radiation Measurement System (SMART) installed on board of HALO during the Mid-Latitude Cirrus (ML-CIRRUS)……."*

Section 5: how high was HALO flying and how high were the clouds. Was is clear above the HALO aircraft?

*The description about HALO and cloud altitudes are given in Sec. 3.2*

*"The first case, a cirrus cloud located above low liquid water clouds (stratocumulus) is selected from ML-15 between 13:56:20 - 13:57:35 UTC as shown in Fig. 3a. The cloud top altitude zt of cirrus was about 12 km while HALO flew at about 12.3 km altitude. The second case, a DCC topped by an anvil cirrus is selected from AC-18 between 17:56:00 - 17:57:30 UTC as presented in Fig. 3b. The zt of the selected DCC was about 8 km while HALO flew at 8.3 km altitude. Flight descriptions and atmospheric conditions during cloud measurements are summarized in Table 1."*

Page 5, line 13: Irradiance is misspelled.

*It has been changed from "irradiace" to irradiance.*

Page 9, line 22: I believe you mean 1640 instead of 2130 here.

*Yes, exactly. However, the original sentences here have been removed because the correction of MODIS band 5 (1240 nm) is now discussed in Sec. 4.1 (also refer to the major comment point 6)*

Page 13, line 12: Please give a definition of Ip for completeness.

*The definition of Ip have been given in Sec. 4.1.*

*"In this study, Ip is defined from the spectral slope of SMART radiance measurements at \lambda =1550 nm and 1700 nm, where the value is typically larger than zero for ice clouds."*

Page 18, line 3: I believe you mean "offers" instead of "affords".

*"Affords" has been changed to offers.*

*"The spectral wm also shows that spectral measurements in the near-infrared wavelengths offers more information on the particle sizes located in different cloud altitudes."*

Page 18, Line 13: Do not start a new sentence at "while". (Same on page 29)

*The uses of 'while' when start a new sentence have been removed .*

Page 19, line 8: The Platnick et al. (2017) paper is also a good reference for the influence of surface albedo.

*Considering the comment from the second reviewer, we merged and tightened the discussion of surface albedo into Section 4.1.*

*"For the cirrus case, the spectral surface albedo \rho of ocean implemented in the forward simulations was measured by SMART. For the DCC case, which is above Amazonian rainforest, no corresponding SMART albedo measurements at low altitude covering exactly the same flight path are available. In this area, the heterogeneity of the surface albedo is very high because where forested and deforested areas are located close to each other. This implies, that a representative assumption of homogeneous surface for the whole flight legs is not appropriate. Therefore, in the DCC case \rho derived from the MODIS BRDF/Albedo product (Strahler et al., 1999) is used to include the horizontal variability of the surface albedo of tropical rainforest."*

Page 21, figure 14: Can the oscillations for the 1240 ice + liquid case be explained?

*We assume, that the oscillation results from numerical uncertainties of optically thin layer. At \lambda = 1240 nm, the upward radiance is dominated by the reflection of the low liquid cloud. Adding thin increments of the cirrus layer does not change the upward radiance significantly. Therefore, numerical uncertainties are visible when calculating the derivate for the weighting function.*

*The plot below is the result when we split the cloud into 20 layers instead of 30 layers. As the result, the optical thickness of each layer is thicker. For cloud A with 20 layers, the optical thickness of each layer is 0.15. In this way, there is no oscillation anymore. Splitting the cloud into 30 layers will result in an optical thickness of 0.10 for each layer. Using this setup, the oscillation occurs as shown in the previous manuscript. To avoid this issue, in the revised manuscript we changed the setup from 30 to 20 layers.*

[Figure]

Page 22, line 22: Refer to the Zhang et al (2010) paper when talking about the differences in ice absorption at 1.6 and 2.13 micron.

*The suggested reference has been implemented in Sec. 4.6.*

*"Due to the similar ice crystal absorption at \lambda = 1640 nm and 2130 nm, both wavelengths have almost identical wm (Wang et al., 2009; Zhang et al., 2010)"*

Page 25, line 4: What does the Delta symbol represent?

*The delta symbol represents the uncertainty. The definition is given in the revised manuscript P.24 L.9.*

*"The results also show, that the uncertainty \delta reff,ci,C1 > \delta reff,ci,C2."*

Page 25, line 8 and further. Note the good agreement between SMART and MODIS for the DCC case and give the mean differences, etc. in the same way as the cirrus case was discussed.

*The agreement between SMART and MODIS and the description have been given the revised manuscript Sec. 4.6 P.24 L.13 and further:*

*"Time series of DCC optical thickness and effective radius retrieved using C1, $\tau_{dcc,C1}$ and $reff_{dcc,C1}$, are shown in Fig. 14a and 14b, respectively. A $\zeta_{\tau,dcc,C1}$ of 1.1% and a $\zeta_{reff,dcc,C1}$ of 6.5% is obtained between SMART and MODIS retrievals. Compared to the cirrus case, the larger horizontal variability indicates a strong evolution of microphysical properties in the deeper layer of DCC. Fig. 14c and Fig. 14d show time series of DCC optical thickness and effective radius retrieved using C2, $\tau_{dcc,C2}$ and $reff_{dcc,C2}$. A $\zeta_{\tau,dcc,C2}$ of 3.5% and a $\zeta_{reff,dcc,C2}$ of 4.1% are obtained in this case. In addition of the fast cloud evolution, larger 3-D radiative effects are likely influencing the observations, which can enhance the deviations of retrieved cloud properties."*

Page 28, line 10: In the list of possible uncertainties also note the uncertainties of deriving effective radius from the in situ measurements and the uncertainties caused by unconstrained choice of ice optical model for the retrievals.

*Thank you. Those really improve the conclusion addressing the discrepancies between in situ and retrieved reff. In the conclusion of the revised manuscript, we wrote:*

*"The variability of particle size distributions, the uncertainties of deriving reff from the in situ measurements, the presence of liquid water cloud below cirrus, and the uncertainties caused by unconstrained choice of ice crystal shapes for the retrievals are identified as the major contributor which can reveal the discrepancies between in situ and retrieved reff. The assumption of vertically homogeneous cloud in the retrieval algorithm has only a small impact on the retrieval results."*

[revised manuscript text omitted]
_{liq}$ = 6 or 10, while the retrieval assumes $\tau_{liq}$ = 8. These biases of $\tau_{ci}$ show, that $\tau_{liq}$ needs to be estimated accurately because a wrong assumption of $\tau_{liq}$ almost directly propagates in the uncertainties of $\tau_{ci}$.
A similar behavior is found for the retrieval of $r_{eff,ci}$, where an overestimation of $r_{eff,liq}$ leads to an underestimation of $r_{eff,ci}$, and vice versa. Assuming larger liquid droplets than in reality implies that these droplets contribute more strongly to the measured absorption at $\lambda$ = 1640 nm, and therefore the ice crystals only contribute less (smaller $r_{eff,ci}$). Fig. 9b illustrates, that

15   the impact of $r_{eff,liq}$ is strongest when small liquid droplets ($r_{eff,liq} \lesssim 8$ μm) are present. For larger liquid droplets ($r_{eff,liq} > 10$ μm), the impact is reduced. The maximum uncertainties of $r_{eff,ci}$ found for the range of $r_{eff,ci}$ and $r_{eff,liq}$ considered here are about 8 μm for the underestimation of $r_{eff,liq}$, which show a tendency of higher uncertainties for higher $r_{eff,ci}$. The retrieval of $r_{eff,ci}$ is less affected by $r_{eff,liq}$, when the cirrus layer is sufficiently thick ($\tau_{ci} > 5$) since then the cirrus layer will dominate the reflected radiation in the absorption bands.

**4.3 Forward simulation of vertically inhomogeneous clouds**

5    It is known from measurements, that the cloud particle sizes can significantly vary with altitudes. For  nonprecipitating ice clouds,  the ice crystal size typically decreases as a function of altitude (Heymsfield et al., 2017, e.g.,). However, to simplify the retrieval algorithm vertically homogeneous clouds are commonly assumed in the forward radiative transfer simulations. To quantify the effects of such simplifications, simulations with vertically inhomogeneous ice clouds are performed. Analytical profiles of

10   effective radius as a function of geometrical height are developed  using a modified parameterization that was originally proposed by Platnick (2000):

$$r_{eff}(z, h) = a_0 - \left( a_1 - a_2 \cdot \frac{z}{h} \right)^{1/k}, \tag{4}$$

where the altitude $z$ ranges from 0 at the cloud base to $h$ at the cloud top.  The parameters $a_0 = r_{eff,t} + r_{eff,b}$, $a_1 = r^k_{eff,t}$, and $a_2 = r^k_{eff,t} - r^k_{
[revised manuscript text omitted]
_{\rm dcc,C2}$ and $r_{\rm eff,dcc,C2}$. A $\zeta_{\tau_{\rm dcc,C2}}$ of 3.5 % and a $\zeta_{r_{\rm eff,dcc,C2}}$ of 4.1 % are obtained in this case. In addition to the fast cloud evolution, larger 3-D radiative effects are likely influencing the observations, which can enhance the deviations of retrieved cloud properties. The cloud properties derived from the MODIS cloud product (blue) are also presented in Fig. 14c and 14d. In this case (over land), the MODIS cloud product algorithm uses C3. The high values of standard deviation $\sigma_{\tau_{\rm dcc}}$ from approach C1, C2, and C3, which are up to 10.3, indicate that $\tau_{\rm dcc}$ is heterogeneous except in the anvil region. The DCC anvil is observed between 17:56:00 - 17:56:20 UTC, which is characterized by relatively smaller $\tau$ between 8 - 15. Later, $\tau_{\rm dcc}$ increases sharply corresponding to the DCC core and decreases again towards the cloud edge. The mean value $\overline{r}_{\rm eff,dcc,C1} > \overline{r}_{\rm eff,dcc,C2} > \overline{r}_{\rm eff,dcc,C4}$ indicates decreasing particle size towards cloud top. It is found, that $\overline{r}_{\rm eff,dcc,C3}$ is larger than $\overline{r}_{\rm eff,dcc,C2}$ corresponding to the different assumptions of the ice crystal habit of plate (SMART and MODIS retrievals) and aggregated columns (MODIS cloud product). Given that $\sigma_{r_{\rm eff,dcc,C1}} > \sigma_{r_{\rm eff,dcc,C2}}$ and $\sigma_{r_{\rm eff,dcc,C2}} < \sigma_{r_{\rm 
[revised manuscript text omitted]

---

## Author Comment (AC2) · 5 Feb 2018

==================================================================

We thank the reviewer for the time and efforts reading our manuscript and providing valuable comments and suggestions. We have made revisions according to your comments and suggestions, as described below. The reviewer comments are written in roman and while the author responses are in *italic*. The diff file indicating changes in the manuscript is enclosed in the end of this document.

General comments:

The authors utilize an airborne radiometer to evaluate MODIS ice cloud retrievals and determine the impact of photons vertical penetration in remotely-sensed cloud effective radius. In addition, in-situ aircraft observations appear to partially validate their hypothesis on the value of different wavelengths for assessing the cloud microphysical vertical structure. The manuscript is interesting and the idea that different near-infrared wavelengths provide information about the cloud vertical structure is interesting. However, the sampling is very small and the in-situ observations matched with MODIS and the radiometer SMART are limited to a few points, so any solid statistical inference or validation of the authors' algorithm (and assumptions) are difficult. Moreover, I do not think it is well-justified the pre-processing of MODIS reflectances as the derivation of a new MODIS 1640 nm using only the 2130 nm reflectance is unphysical. The authors cite a number of papers for justifying the MODIS corrections, but this is a misinterpretation of the literature results. I am struggling with my recommendations because even though the authors show some interesting results, the analysis is not rigorous, and the comparison between MODIS science team retrievals and their own MODIS cloud retrievals is flawed.

Response of general comments:

*The reviewer is rightfully spotted that one scientific question is using multi near-infrared wavelengths to investigate the cloud vertical structure based on the theory of vertical photon transport. We agree, that the limited and not well suited cases investigated in the study are not sufficient to draw general conclusions about the MODIS performance using airborne measurements. The limited number of cases results from the careful selection of measurements which allows to evaluate the radiance measurements and retrieved cloud products. Such comparisons*

*require almost perfectly collocated measurement between airborne and satellite, which is given by only few flights during the two investigated campaigns (ML-CIRRUS and ACRIDICON-CHUVA). In addition, inappropriate cloud situations had to be rejected. In order to avoid the impression, that the comparison is valid for cirrus and deep convective clouds in general, we strengthen throughout the revised manuscript, that only a case study is presented. The title is changed to:*

> *"Comparing airborne and satellite retrievals of cloud optical thickness and particle effective radius using a spectral radiance ratio technique: Two case studies for multilayer cirrus and deep convective clouds"*

*It is right, that the treatment of the MODIS measurements is questionable and not well justified in the previous manuscript. The motivation is to use identical wavelength for both SMART and MODIS retrievals. Unfortunately, the SMART measurements in the near-infrared do only cover the MODIS bands centered at \lambda = 1240 nm and 1640 nm. At 2130 nm the uncertainty of SMART is large. Therefore, it could not be included in the study. It is known that MODIS band 6 (\lambda = 1640 nm) has problems with the detector. Using remaining detectors of MODIS band 6 is not possible due to the very limited number of pixel in our cloud cases. Therefore we used the approach by Wang et al. (2006), which indeed was developed for snow surfaces to 'retrieve' MODIS band 6. We think that this approach is justified also for measurements above ice clouds because, the optical properties of ice clouds are very similar to a snow surface (similar refractive index). To some degree this is confirmed by the agreement between restored MODIS band 6 and SMART as shown the revised manuscript (Fig. 4c, 5c, and 6b). In the revised manuscript we also added the motivation and the method of MODIS band 6 more clearly.*

=================================================================

Specific comments:

1) The derivation of MODIS band 6 using band 7 is unphysical and the justification based on the results in Wang et al (2006) is misleading as Wang et al. shows that the correction is useful for estimating NDVI, which is a completely different problem. The 2130 nm and 1640 nm have distinctive photon vertical penetration (e.g. your figure 11) so the conversion is unphysical. I do not think the paper can be accepted until they authors use the standard MODIS channels without any correction. I agree that the 1640 nm MODIS channel has issues, so I would suggest the direct use of the 2130 nm channel instead.

Response:

*The reviewer is correct that the original approach by Wang et al. (2006) to retrieve MODIS band 6 from band 7 was applied on the basis of NDSI (normalized difference snow index), while in this study we intended to apply comparisons of radiance and retrieval results using the collocation wavelengths between SMART and MODIS. The spectral range covered by SMART is indeed between 300 - 2200 nm. However, the sensitivity decreases for small and large wavelengths depending on the magnitude of radiation. Only the wavelengths range between 400 - 1800 nm provides measurements with reasonable uncertainty. Therefore, a direct comparison with the 2130 nm MODIS band is not possible.*

[Figure]

*Wang et al. (2006) demonstrated that that snow have similar reflectance characteristics in both wavelengths. Due to similarities on the optical properties of snow and ice clouds, we think that both would also have similar absorptions. Fig. 11 in the manuscript only shows wm up to 2000 nm. To provide more relevant information, we calculated wm at \lambda centered at 1640 nm and 2130 nm as shown in the plots above. (a) is for \tau_ci = 3, while (b) is for (\tau_ci = 15). Our findings illustrate that both wavelengths have almost identical wm for ice clouds, in accordance with findings by Wang et al. (2009) and Zhang et al. (2010). For the two idealized clouds presented in our study, the reff retrieved using 1640 nm and 2130 nm only differs about 0.2-0.4 µm.*

*Following Wang et al. (2006), we performed a study to investigate and further to parameterize the correlation between band 6 and 7 for measurements of ice clouds. The motivation and technique to retrieve MODIS band 6 is presented in the revised manuscript:*

> *"According to Wang et al. (2006), the MODIS radiance band 6 (IM,B6) can be retrieved using band 7 IM,B7 (\lambda = 2130 nm). This technique was originally developed and tested on the basis of snow surface, assuming that the spectral characteristics of the snow reflectivity between MODIS band 6 and 7 does not change significantly for different snow types. Assuming that ice clouds and snow have similar optical properties, the same approach can be applied. Similar to Wang et al. (2006), a parameterization of IM,B6 is*

*developed on the basis of radiative transfer simulations of upward radiance performed for cirrus with different \tau and reff. A polynomial fit is applied to quantify the relation between IM,B6 and IM,B7 which result the parameterization:*

*IM,B6 = -81.033  IM,B7$^2$ + 3.257 IM,B7 + 0.002 "*

*"The validity of the parameterization is tested using the remaining detectors of MODIS band 6 for observations above cirrus (not shown here). The linear regression between original and retrieved IM,B6 showed differences below 5% (slope of 0.95 and zero bias) with a correlation coefficient of 0.94."*

*To develop the parameterization as shown by the Equation above, the simulations are run for different values of \tau (2-7) and reff (10-45 μm). Fig. a below is the scatter plot between radiance band 6 (1640 nm) and band 7 (2130 nm). The dashed line is the linear regression line. The equation is developed by making use of the relation between the two bands. Fig. b is the scatter plot of radiance band 6 original vs. 'retrieved' using the Equation above. Here we see that, the retrieval of MODIS band 6 shows a good performance with a slope of 1, no bias, and R$^2$= 1.*

*Finally, as suggested by the first reviewer we compared the radiance from the remaining detectors of MODIS band 6 with the retrieved values to test the validity of this band retrieval technique in real measurements (Fig. c below). Here we compared the remaining detectors of MODIS band 6 and the retrieved values for measurements above clouds. The result confirms the performance and validity of the 'retrieved' MODIS band 6 for cloud measurements.*

*We do not omit the reviewer suggestion to use 2130 nm in the retrieval. Therefore in Sec. 5 we also run MODIS retrievals using 2130 nm for the comparison with the in situ data. The results is presented in Fig. 14 and Table 4.*

*"Additionally, the reff retrieved using additional SMART measurements at \lambda = 1500 nm, 1550 nm, and 1700 nm, and also MODIS radiances centered at \lambda = 2130 nm and 3700 nm (band 20) are applied in this comparison."*

*Here we also found the similarities between the result of \lambda = 1640 nm and 2130 nm showed by the mean value of reff which only differs by 0.3 μm (Table 4).*

[Figure]

2) The authors correct the MODIS 1240 nm channels using as a justification Lyapustin et al (2014) but, again, the use of this reference is misleading. This paper only corrects MODIS in order to remove a spurious trend. Moreover, the new collection 6 radiances should have incorporated the modifications described in Lyapustin et al. Overall, the use of SMART for correcting MODIS is unjustified. It is much more rigorous to list the differences between SMART and MODIS and then compare the retrievals, keeping in mind the instruments differences.

Response:

*It is correct, that without independent standard, we cannot judge, if MODIS or SMART do measure right or wrong. The major justification why we corrected MODIS and used SMART as reference shows up in the cloud retrieval. In the revised manuscript, we added all measured radiances into the LUTs in Fig. 8. Here it is obvious, that the MODIS data does not fall into the parameter space of the forward simulations. While the range of radiance I,645 still matches the simulations, the ratio R_1240 = I,1240/I,645 nm does not. We also could not find any comment about such a bias in literature and also cannot exclude that the forward simulations are biased (effect of ice crystal shape or scattering library), while in this study we already use MODIS collection 6. In order to allow a reasonable retrieval with MODIS data, we finally decided to scale the MODIS band 5 (1240 nm). For both cloud cases, the bias is nearly consistent about 10%. Increasing*

*retrieval failure in the cirrus case is related to the larger solar zenith angle, which makes the reff LUTs more denser. Therefore small changes in the measurements can lead to a significant bias.*

[Figure]

*A detailed discussion on the scaling is added to the revised manuscript:*

*"The measurements of SMART (black crosses) and MODIS (blue circles) are included for both scenes in Fig. 8. For the C1 which is based on I_1240, the MODIS data does not match the lookup table solution space. The results in Section 3.3 show clearly, that I_M,1240 are higher than I_S,1240 by about 15%. Using the original I_M,1240 for the cirrus case, all the retrievals of reff are fail because the measurements lie far outside the lookup table solution space (see Fig. 8a), while for the DCC case the retrieval failure is smaller (see Fig. 8c). Enhancing retrieval failure in the cirrus case is due to the larger \theta_0. At a larger \theta_0, the upward radiance becomes more insensitive to the changes of reff and consequently the lookup tables are denser. To gain meaningful retrieved cloud properties, a correction of I_M,1240 is applied. Following Lyapustin et al. (2014), a correction factor g is calculated by the slope of linear regression between I_M,1240 and I_S,1240, which results in g = 0.88 for the cirrus case and g = 0.90 for the DCC case. The corrected I_M,1240 (red circles) are added in Fig. 8 and now match the lookup table solution space. Therefore, all following radiance ratio retrievals for the two cloud cases use these corrected I_M,1240."*

3) A central assumption is the liquid optical depth and effective radius of the layer below the cirrus clouds. The authors choose constant values for each case but these uncertain values can substantially bias the retrievals, and moreover there is no way to know if the constant values are correct or not. So, any comparison with MODIS standard retrievals will have the huge uncertainty due to the liquid optical depth utilized for creating the lookup tables (the use of climatological values is suspicious). This is very problematic because it is unclear whether the new retrievals are better than MODIS. Overall, to prove the point that accurate ice retrievals depend on the ability of accounting for the cloud layer below the cirrus

clouds, you only need synthetic observations, which is why I do not see the point of comparing satellite and airborne remote sensing retrievals.

Response:

*As stated by the reviewer, the characterization of the liquid cloud layer is crucial for the retrieval of the cirrus properties and the manuscript did not present this issue properly. Disentangling the contribution of both cloud layers to the total measured radiance is challenging. We used simulations for different combinations of liquid water and cirrus cloud properties and compared the simulated radiance with SMART measurements of the entire spectral range covered by SMART (see figure below). The absorption bands of water vapor (940 and 1135 nm) and the O2 A-band (761 nm) provide some information on the multi-layer structure. Depending on the \tau of the high (cirrus) and low (liquid) cloud layer, absorption features by atmospheric trace gases are stronger or weaker imprinted in the spectral radiance.*

*Fig. a-c below indicate, that the best fit in the spectral, especially in the absorption bands of water vapor and O2 A-band, was found for \tau_li = 8 and reff_li = 10 µm combined with \tau_ci = 3 and reff_ci = 15 µm. The reff of the liquid cloud is of less importance. If the \tau of the cirrus is sufficiently high, the spectral range used to derive the reff of the cirrus dominated by scattering in the cirrus layer only. In Fig. d, we show the impact of the combination between \tau_ci and \tau_li, which can also give some insights by observing changes in the spectral. For this purpose, we hold \tau_ci + \tau_li = constant with fixed reff_ci = 15 µm and reff_li = 10 µm. Here, again we found the best fit in the spectral is given by \tau_ci = 3 and \tau_li = 8. Underestimation / overestimation of \tau_li will produce gaps, particularly in water vapor absorption bands and O2-A band.*

[Figure]

[Figure]

In the revised manuscript, we also considered the variability of the liquid cloud properties along the flight path. To illustrate this problem more clearly, we added a sensitivity study in Sec. 4.2 with respect to the assumed properties of the liquid cloud which provides an estimation of the retrieval uncertainties due to uncertainties in the assumption of the liquid cloud properties.

*"For the cirrus case, the properties of the low liquid water cloud is assumed to be constant along the flight leg. This assumption might not hold in reality and affect the retrieved cirrus properties. Therefore, the sensitivity of the cirrus retrieval on the assumed properties of the liquid water cloud is quantified using radiative transfer simulations. Spectral radiance are simulated for different combinations of liquid water cloud and cirrus properties. The liquid water cloud is varied for \tau_li = 6 - 10 and reff_li = 6 - 14 µm, while the cirrus is changed for \tau_ci = 2 - 8 and reff_ci = 10 - 40 µm. These simulated radiances are used as synthetic measurements and analyzed with the retrieval algorithm using C2 (I_645 and \Re_1640), which assumes a liquid water cloud with \tau_li = 8 and reff_li = 10 µm. The comparison of synthetically retrieved and original \tau_ci*

*and r_eff,ci is shown in Fig. 9. The annotation of "overestimation" (below one-to-one line) and "underestimation" (above one-to-one line) corresponds to when the retrieval is run with an overestimation and underestimation of the properties of liquid water cloud. The retrieved \tau_ci are analyzed in Fig. 9a for different \tau_li, while r_eff,ci and r_eff,li are fixed to 20 μm and 10 μm, respectively. Similarly, the retrieved r_eff,ci are analyzed in Fig. 9b for different r_eff,li but for a fixed combination of \tau_ci = 3 and \tau_li = 8. In general, the simulations show that an overestimation of \tau_li leads to an underestimation of \tau_ci because in this case, the liquid water cloud contribute stronger to the reflected radiation than in reality. Therefore, a smaller \tau_ci is required to match the measurement, and vice versa. For the range of \tau_ci analyzed here, the retrieved \tau_ci is found be over- or underestimated by 1.3 when in reality \tau_li is 6 or 10, while the retrieval assumes \tau_li = 8. These biases of $\tau_ci show, that \tau_li needs to be estimated accurately because a wrong assumption of tau_li almost directly propagates in uncertainties of \tau_ci.*

*A similar behavior is found for the retrieval of reff_ci, where an overestimation of reff_li leads to an underestimation of reff_ci, and vice versa. Assuming larger liquid droplets than in reality implies that these droplets contribute stronger to the measured absorption at \lambda = 1640 nm, and therefore the ice crystals only contribute less (smaller reff_ci). Fig. 9b illustrates, that the impact of reff_li is strongest when small liquid droplets (reff_li <= 8 μm) are present. For larger liquid droplets (reff_li > 10 μm), the impact is smaller. The maximum uncertainties of reff_ci found for the range of reff_ci and reff_li considered here are about 8 μm for the underestimation of μm which show a tendency of higher uncertainties for higher reff_ci. The retrieval of reff_ci is less affected by reff_li, when the cirrus layer is sufficiently thick (\tau_ci > 5) since the cirrus layer will dominate the reflected radiation in the absorption bands."*

4) I find it surprising that the authors did not retrieve effective radius using MODIS 3.78 um channel. If you want to demonstrate that the liquid cloud layer can bias the retrievals, then you have to calculate the effective radius for all the available MODIS channels. In my opinion, the main conclusion of the paper is that accounting for liquid clouds is far more important than accounting for the vertical inhomogeneity.

Response:

*It is correct, that to demonstrate the bias due of low liquid cloud, we should use MODIS radiance and run the radiance ratio retrieval considering the low liquid cloud. Therefore in the revised manuscript, we introduced retrievals using MODIS band 7 and band 20 (3700 nm). By doing this, the correlation between the retrieval results with the in situ weighting-estimate reff,w\* is significantly improved, which*

*also can be used to estimate the impact of low liquid cloud. A normalized mean absolute deviation of \zeta= 8.3% and 1.5% for retrieval using 2130 nm and 3700 nm was obtained, where using the reff derived from the MODIS cloud product the \zeta is up to 47.5%. By removing MODIS cloud product in this analysis, overall, the \zeta between in situ reff,w\* and retrieved reff ranges between 1.5 - 10.3% which falls within the standard deviation (variability of horizontal reff) and considerably as a good agreement. The resulting correlation coefficient $R^2$ is 0.82 which shows a robust agreement. We changed the discussion in the manuscript accordingly.*

> *"Additionally, the reff retrieved by using additional SMART measurements at \lambda = 1500 nm, 1550 nm, and 1700 nm, and also MODIS radiances centered at \lambda = 2130 nm and 3700 nm (band 20) are applied in this comparison. The retrieval and the calculation of wm for \lambda = 3700 nm are performed by considering both solar and thermal radiation....."*

> *"The deviations of in situ reff,w$^*$ and SMART reff range between 3.2% (\lambda = 1500 nm) and 10.3% (\lambda = 1550 nm). Between reff,w$^*$and MODIS reff, the \zeta results in a value between 1.5% for \lambda = 3700 nm and 9.1% for \lambda = 1640 nm. Overall, the values of \zeta are in the range between 1.5 - 10.3% and agree within the horizontal standard deviation, as shown in Fig. 15b."*

> *"The reff derived from the MODIS cloud product are obviously affected by the low liquid water cloud, which is not included in the algorithm of MODIS operational retrieval. Therefore, a \zeta of 47.5% and 19.3% are obtained for reff,L,2130 and reff,L,3700, respectively. The absorption by ice crystals at \lambda = 3700 nm is very strong. Consequently, the first top layers will dominate the absorption and significantly reduce the effect of the underlying liquid water cloud. Fig. 15c shows a scatter plot of in situ reff,w$^*$ and reff retrieved from SMART (black triangles) and MODIS (red dots), while the dashed line represents the one-to-one line. There is a robust agreement between in situ reff,w$^*$ and retrieved reff with a correlation coefficient $R^2$ of 0.82."*

5) In my opinion, the main conclusion of the paper is that accounting for liquid clouds is far more important than accounting for the vertical inhomogeneity.

Response:

*A bias of 1 micron is small or not might depend on the related question. We agree, compared to other retrieval uncertainties this is not the major issue and the vertically homogeneous assumption might be sufficient. Also with considering the first reviewer comment, we modified the conclusion in the revised manuscript as follows:*

*"The variability of particle size distributions, the uncertainties of deriving reff from the in situ measurements, the presence of liquid water cloud below cirrus, and the uncertainties caused by unconstrained choice of ice crystal shapes for the retrievals, are considered as the main contributor which can reveal the discrepancies between in situ and retrieved reff. The assumption of vertically homogeneous cloud in the retrieval algorithm has only a small impact on the retrieval results."*

6) I believe collection 6 utilizes a new habit for the lookup table calculations. This new habit appears to produce different results compared to the ice crystal habit of collection 5.

Response:

*For the retrievals of the cirrus case, we use GHM based on severely roughened aggregates composed of nine habits (Baum et al., 2014), while the ice crystal habit of plate with high surface roughness (Yang et al., 2013) is applied for the retrievals of the DCC case. The assumption of ice crystal habit considers the measurements by in situ probes. The In the revised manuscript, we added a discussion on the impact of using GHM instead of aggregated columns which is based on the suggested literature.*

*"These particle habits differ from the MODIS collection 6 retrievals which use severely-roughened compact aggregates of solid columns (so-called aggregated columns) by Yang et al. (2013). A sensitivity study infers that the retrievals assuming GHM and plate generally will result in a larger \tau and smaller reff (not shown here), which is in agreement with findings by van Diedenhoven et al. (2014) and Holz et al. (2016)."*

7) I do not see the value of section 5.4. If you deem it necessary, please include the section as a part of the appendix.

*We merged and tightened the discussion of the surface albedo in Section 4.1.*

[revised manuscript text omitted]

---

## Author Response (AR2)

Response to Co-Editor and Reviewers

Comparing airborne and satellite retrievals of cloud optical thickness and particle effective radius using a spectral radiance ratio technique: Two case studies for multilayer cirrus and deep convective clouds

acp-2017-758 : Krisna et al.
Email to: trismono_candra.krisna@uni-leipzig.de

We thank the reviewer and editor for the time and efforts reading our manuscript and providing valuable comments. We have made revisions according to your comments below. The changes in the manuscript are written in *italic* inside quotation marks (""). The diff file indicating changes in the manuscript is enclosed in the end of this document.

Comments to the Author:
I got one formal review and one informal review.

Both reviewers recommend acceptance, one asked for minor revisions:

The second reviewer wrote:

The authors have done a remarkable work addressing my questions and concerns. There are only a few lingering (minor) issues:

- Please, revise the writing style of the manuscript. There are several places in which the sentences lack style, i.e. they are grammatically correct but need to be reworded (e.g. first two sentences of section 4.6). I do acknowledge that the latest manuscript is in better shape, though.

**Response:**

We have changed the manuscript according to the language style along with the grammar checking. Regarding the specific comment given by the reviewer, we revised the first two sentences of Section 4.6.

*"Time series of $\tau$ and reff retrieved from SMART and MODIS radiance measurements, along with the MODIS cloud product, are compared for the two cloud cases. The MODIS cloud product, namely MYD06_L2, provides three different reff (so-called reff_L,1640, reff_L,2130, and reff_L,3700), which are retrieved using respective near-infrared wavelengths centered at $\lambda = 1640$ nm,2130 nm, and 3700 nm (Platnick et al., 2017)."*

- Regarding the parameterization to derive the new MODIS band 6 radiances. It says that the validity of the parameterization was tested with observations, but how are you going to test it if the actual MODIS band 6 is not functional. Did you use synthetic measurements to test the parameterization?

**Response:**

The reviewer is correct that the actual MODIS band 6 is noisy or nonfunctional. For more precise, 15 of 20 detectors are in trouble. The striping due to this problem is shown by Fig. 1a below. To check the parameterization, we are not only using synthetic measurement. If we do so, of course it will result a perfect agreement (Fig. 1b). The validity of band 6 retrieval is examined using remaining pixels of MODIS band 6 (along the flight legs and neighbouring pixels). The analysis of a linear regression results in a slope of 0.95 (5% bias) and a correlation coefficient of 0.94 (Fig. 1c).

[Figure]

Fig. 1. Actual MODIS band 6 (a).  Comparison of actual and retrieved band 6 using synthetic measurements for different prescribed cloud properties (b). Comparison or actual and retrieved MODIS radiance band 6.

- Page 16, line 3: Rather tan citing Zhang and King, I would recommend to mention Liang et al. (2009) because, to the best of my knowledge, Liang et al is the first study that applies the heterogeneity index to MODIS reflectances. In addition inhomogeneities is also producing a spatial heterogeneity effect, which is unrelated to the 3D radiative effects.

  Liang, L., L. Di Girolamo, and S. Platnick (2009), View-angle consistency in reflectance, optical thickness and spherical albedo of marine water-clouds over the northeastern

Pacific through MISR-MODIS fusion, Geophys. Res. Lett., 36, L09811, doi:10.1029/2008GL037124.

**Response:**

We thank for the reference suggested by the reviewer. In the revised manuscript, we added Liang et al. (2009) in the following form:

*"Liang et al. (2009), Zhang and Platnick (2011), and King et al. (2013) estimated the influence of 3-D radiative effects using the cloud heterogeneity index (\sigma_sub)...."*

King et al. (2013) investigated specifically the influence of cloud heterogeneity (both sub-pixel and surrounding-pixel) on the retrieval of reff which highlighted the potentially large influence of 3-D effects when comparing reff from different instruments. They found that the differences between retrieved and in situ values are getting higher with increasing cloud heterogeneity.

- Figure 9: This is a nice figure. My only criticism is on the use of over/underestimation of tau and reff, as It is a little bit confusing. It sounds better for instance: "prescribed tau_liq is overestimated (underestimated).

We have changed the figure as follows:

[Figure]

- Table 4 is interesting. It is confusing the use of the eta letter. Just use r_e plus/minus sigma. I am not aware of many studies that show such a nice comparison between the three different MODIS r_eff and in-situ observations. Kokhanovsky et al. (2013) attempted to retrieve the cloud vertical structure of stratiform clouds with three MODIS channels, and then compare their results with in-situ observations. However, their attempt was applied to warm clouds only.

A. A. Kokhanovsky, D. Painemal and V. V. Rozanov, "The Intercomparison of Satellite-Derived and In Situ Profiles of Droplet Effective Radii in Marine Stratocumulus Clouds," in IEEE Geoscience and Remote Sensing Letters, vol. 10, no. 5, pp. 1147-1151, Sept. 2013. doi: 10.1109/LGRS.2012.2233710

**Response :**

The reviewer is correct that Kokhanovsky et al. (2013) aimed to compare remote sensing and in situ reff of liquid water clouds. However, what they did specifically is to retrieve the profile of reff by assuming a linear increase of reff. A similar work has been done by Wang et al. (2009), but for ice clouds (has been cited in our paper). Nevertheless, we think that such comparisons are rather different with our work because we did not retrieve the full profile.

Instead, we have cited Painemal and Zuidema (2011) and King et al. (2013), who performed rather similar studies with us but for liquid water clouds. Painemal and Zuidema (2011) compared MODIS and in situ reff at cloud top (averaged between geometrical depth of 25 m). King et al. (2013) re-implemented the approach by Painemal and Zuidema (2011), and additionally calculated the weighting reff for comparing with MODIS reff. Both references have been cited in our paper. We also encountered similar results with them, that the retrieved and in situ reff differ of up to 20%.

We have revised Table 4 as follows:

Table 4. The mean $\pm$ standard deviation of $r_{\text{eff}}$ from in situ (CCP), retrievals (SMART and MODIS), and MODIS cloud product (MYD06_L2) for different near-infrared wavelengths between 1240 nm - 3700 nm. The wavelengths have been sorted in order that the degree of absorption by cloud particles increases to the right. The last line is the the weighting-altitude $z_w^*$.

| $r_{\text{eff}} \pm \sigma$ | 1240 nm | 1700 nm | 1640 nm | 2130 nm | 1550 nm | 1500 nm | 3700 nm |
|---|---|---|---|---|---|---|---|
| CCP (µm) | $19.0 \pm 9.8$ | $18.3 \pm 9.6$ | $18.0 \pm 9.5$ | $17.5 \pm 9.4$ | $17.0 \pm 9.3$ | $16.7 \pm 9.3$ | $7.0 \pm 5.0$ |
| SMART (µm) | $22.7 \pm 8.8$ | $16.5 \pm 6.7$ | $15.6 \pm 3.9$ | - | $13.9 \pm 3.7$ | $15.7 \pm 2.1$ | - |
| MODIS (µm) | $22.4 \pm 8.6$ | - | $15.0 \pm 1.9$ | $14.8 \pm 4.9$ | - | - | $7.2 \pm 5.1$ |
| MYD06_L2 (µm) | - | - | - | $6.2 \pm 1.2$ | - | - | $4.8 \pm 3.7$ |
| $z_w^*$ (km) | 11.39 | 11.42 | 11.44 | 11.46 | 11.48 | 11.49 | 11.89 |

Based on the comments by the two (experienced) reviewers, this paper is accepted subject to the above recommended minor revisions.

Non-public comments to the Author:
One of the reviewers recommended confidentially (in my words) careful revision by the prime author in close cooperation with his senior co-authors.

[revised manuscript text omitted]

$$I^{\uparrow}_{S,\lambda} = \frac{\int_{\lambda_1}^{\lambda_2} I^{\uparrow}_{s,\lambda} \cdot R(\lambda) \, d\lambda}{\int_{\lambda_1}^{\lambda_2} R(\lambda) \, d\lambda} \tag{1}$$

In this study, upward radiances centered at the MODIS band 1 ($\lambda$ = 645 nm), band 5 ($\lambda$ = 1240 nm), and band 6 ($\lambda$ = 1640 nm) will be primarily used to retrieve $\tau$ and $r_{eff}$. It is known that 15 of the 20 detectors in the MODIS-Aqua band 6 are either

5 nonfunctional or noisy. However, according to Wang et al. (2006), the MODIS radiance band 6 $I_{M,B6}$ can be retrieved using band 7 $I_{M,B7}$ ($\lambda$ = 2130 nm). This technique was originally developed and tested on the basis of snow surfaces assuming that

the spectral characteristics of the snow reflectivity between MODIS band 6 and 7 do not change significantly for different snow types. Assuming that ice clouds and snow have similar optical properties, the same approach can be applied. Similar to Wang et al. (2006), a parameterization of $I_{M,B6}$ is developed on the basis of radiative transfer simulations of upward radiance

10 performed for cirrus with different $\tau$ and $r_{eff}$. A polynomial fit is applied to quantify the relation between $I_{M,B6}$ and $I_{M,B7}$ which result the parameterization:

[revised manuscript text omitted]

---

## Author Response (AR3)

Response to the Co-Editor

Comparing airborne and satellite retrievals of cloud optical thickness and particle effective radius using a spectral radiance ratio technique: Two case studies for multilayer cirrus and deep convective clouds

acp-2017-758 : Krisna et al.
Email to: trismono_candra.krisna@uni-leipzig.de

General comments:

The paper has been well revised taking all comments by the reviewers into account, as far as I could see. So it is acceptable. Still, I suggest a few minor editorial changes to the text. Please see my comments in the annotated pdf manuscript, attached.

We thank the co-editor for the time and efforts reading our manuscript and providing valuable comments. We have edited the manuscript according to your comments. The diff file, which indicates changes in the manuscript is enclosed in the end of this document.

Specific comments:

- I suggest omittng the \pm sign at all other places after the meaning has been clarified at one place, e.g., here.

Response:

As suggested by the co-editor, we have changed the respective term in the revised manuscript:

"*The values of mean standard deviation \eta at each spectral wavelength are summarized in Table 2. Note, that all standard deviation values in this paper refer to the \pm values.*"

- I assume you adjust the temperature and perhaps the humidity, not the trace gas profiles. Please specify what you adjusted.

Response:

The co-editor is correct that we adjusted the temperature and humidity profile. Therefore, we clarified this issue in the revised manuscript:

"*The atmospheric profiles of gases and constituents are adapted from the standard profile (Anderson, 1986) "mid-latitude" for ML-CIRRUS and "tropical" for ACRIDICON-CHUVA, and are adjusted to the radio sounding data (temperature and humidity) close to the measurement area.*"

- I am a bit confused by your vertical and spectral weighting functions, since $w_m$ depends on both and on even more parameters (Eq. 5). Perhaps you can make this clearer to avoid misunderstandings.

Response:

The co-editor is correct that the vertical weighting function w_m is calculated by Eq. 5. What we mean about the spectral vertical weighting function is exactly same as the w_m in Fig. 10c and 10d, but it is calculated at \lambda = 1000 - 2000 nm, as shown in Fig. 11a. The color in Fig. 11a represents the weighting. To avoid misunderstandings, we do not use the term of "spectral weighting function" anymore. Instead, we specifically wrote:

[revised manuscript text omitted]